# Independence Testing-Based Approach to Causal Discovery under Measurement Error and Linear Non-Gaussian Models

**Haoyue Dai**[1,2]  **Peter Spirtes**[1]  **Kun Zhang**[1,2]

[1]Department of Philosophy, Carnegie Mellon University
[2]Machine Learning Department, Mohamed bin Zayed University of Artificial Intelligence
hyda@cmu.edu    ps7z@andrew.cmu.edu    kunz1@cmu.edu

## Abstract

Causal discovery aims to recover causal structures generating the observational data. Despite its success in certain problems, in many real-world scenarios the observed variables are not the target variables of interest, but the imperfect measures of the target variables. Causal discovery under measurement error aims to recover the causal graph among unobserved target variables from observations made with measurement error. We consider a specific formulation of the problem, where the unobserved target variables follow a linear non-Gaussian acyclic model, and the measurement process follows the random measurement error model. Existing methods on this formulation rely on non-scalable over-complete independent component analysis (OICA). In this work, we propose the Transformed Independent Noise (TIN) condition, which checks for independence between a specific linear transformation of some measured variables and certain other measured variables. By leveraging the non-Gaussianity and higher-order statistics of data, TIN is informative about the graph structure among the unobserved target variables. By utilizing TIN, the ordered group decomposition of the causal model is identifiable. In other words, we could achieve what once required OICA to achieve by only conducting independence tests. Experimental results on both synthetic and real-world data demonstrate the effectiveness and reliability of our method.[1]

## 1 Introduction

Discovery of causal relations is a fundamental goal of science. To identify causal relations from observational data, known as causal discovery, has thus drawn much attention in various scientific fields, including economics, biology, and social science [45, 27, 11]. Methods for causal discovery can be roughly categorized into constraint-based ones (e.g., PC [43]), score-based ones (e.g., Greedy Equivalence Search (GES) [8]), and ones based on structural equation models (SEM) [38, 17, 51]. Almost all these methods assume that the recorded values are values of the variables of interest, which however, is usually not the case in real-world scenarios. Some variables may be impossible to observe or quantify, so recorded values are actually a proxy of them (e.g., measure one's mental status by survey questionnaire), and some variables, though quantifiable, may subject to error introduced by instruments (e.g., measure brain signals using functional magnetic resonance (fMRI)). The difference between quantities of interest and their measured value is termed as *measurement error* [12].



Figure 1: Example of measurement error. Gray nodes are latent underlying variables and white nodes are observed ones.

---

[1]An online demo and codes are available at https://cmu.edu/dietrich/causality/tin.

36th Conference on Neural Information Processing Systems (NeurIPS 2022).

Measurement error adversely impairs causal discovery [28, 24, 36]. The measuring process can be viewed as directed edges from underlying variables of interest (unobservable) to measured values (observable), and the d-separation patterns on underlying variables typically do no hold on measured ones. Consider the causal effects from factory emissions 🏭 to air quality 🌫 then to residents' lung health 🫁, as shown in Figure 1, while we only have corresponding measured quantities: chimney statistics 📊, $PM_{2.5}$, and hospital reported cases 🏥. Though 🏭 and 🫁 are independent given 🌫, 📊 and 🏥 are however dependent given $PM_{2.5}$. If measurement error is severe, 📊 and 🏥 even tend to be marginally independent [53, 36], which makes $PM_{2.5}$ look like a collider (common child). One might thus incorrectly infer that, lung cancer causes air pollution. In fact, such measurement error is always a serious threat in environmental epidemiologic studies [30, 10].

Denote by $\tilde{\mathbf{X}} = \{\tilde{X}_i\}_{i=1}^n$ the latent measurement-error-free variables and $\mathbf{X} = \{X_i\}_{i=1}^n$ observed ones. While there are different models for measuring process [12, 6, 49], in this paper, we consider the random measurement error model [36], where the observed variables are generated from the latent measurement-error-free variables $\tilde{X}_i$ with additive random measurement errors $\mathbf{E} = \{E_i\}_{i=1}^n$:

$$X_i = \tilde{X}_i + E_i. \tag{1}$$

Measurement errors $\mathbf{E}$ are assumed to be mutually independent and independent of $\tilde{\mathbf{X}}$. We assume causal sufficiency relative to $\tilde{\mathbf{X}}$ (i.e., no confounder of $\tilde{\mathbf{X}}$ who does not have a respective measurement), and focus on the case where $\tilde{\mathbf{X}}$ is generated by a linear, non-Gaussian, acyclic model (LiNGAM [38], see §3.1). Note that here w.l.o.g., the linear weights of $\{\tilde{X}_i \rightarrow X_i\}_{i=1}^n$ are assumed to be one (since we do not care about scaling). Generally, if observations are measured by $X_i = c_i\tilde{X}_i + E_i$ with weights $\{c_i\}_{i=1}^n$ not necessarily being one, all results in this paper still hold.

The objective of causal discovery under measurement error is to recover causal structure among latent variables $\tilde{\mathbf{X}}$, denoted as $\tilde{G}$, a directed acyclic graph (DAG), from contaminated observations $\mathbf{X}$. As illustrated by Figure 1, causal discovery methods that utilize (conditional) independence produce biased estimation (see Proposition 1 for details). SEM-based methods also typically fail to find correct directions, since the SEM for $\tilde{\mathbf{X}}$ usually do not hold on $\mathbf{X}$. Unobserved $\tilde{\mathbf{X}}$ are actually confounders of $\mathbf{X}$, and there exists approaches to deal with confounders, such as Fast Causal Inference (FCI [44]). However, they focus on structure among observed variables instead of the unobserved ones, which is what we aim to recover here. With the interest for the latter, another line of research called *causal discovery with latent variables* is developed, which this paper is also categorized to. However, existing methods [41, 42, 47, 46, 1, 23, 50, 35] cannot be adopted either, since they typically require at least two measurements (indicators) for each latent variable, while we only have one for each here (and is thus a more difficult task). Specifically on the measurement error problem, [16] proposes anchored causal inference in the binary setting. In the linear Gaussian setting, [53] presents identifiability conditions by factor analysis. A main difficulty here is the unknown variances of the measurement errors $\mathbf{E}$, otherwise the covariance matrix of $\tilde{\mathbf{X}}$ can be obtained and readily used. To this end, [2] provides an upper-bound of $\mathbf{E}$ and [34] develops a consistent partial correlations estimator. In linear non-Gaussian settings (i.e., the setting of this paper), [54] shows that the *ordered group decomposition* of $\tilde{G}$, which contains major causal information, is identifiable. However, the corresponding method relies on over-complete independent component analysis (OICA [19]), which is notorious for suffering from local optimal and high computational complexity [39, 18]. Hence, the identifiability results in [54], despite the theoretical correctness, is far from practical achievability.

The main contributions of this paper are as follows: **1)** We define the **T**ransformed **I**ndependent **N**oise (TIN) condition, which finds and checks for independence between a specific linear transformation (combination) of some variables and others. The existing Independent Noise (IN [40]) and Generalized Independent Noise (GIN [50]) conditions are special cases of TIN. **2)** We provide graphical criteria of TIN, which might further improve identifiability of causal discovery with latent variables. **3)** We exploit TIN on a specific task, causal discovery under measurement error and LiNGAM, and identify the *ordered group decomposition*. This identifiability result once required computationally and statistically ineffective OICA to achieve, while we achieve it merely by conducting independence tests. Evaluation on both synthetic and real-world data demonstrate the effectiveness of our method.

## 2    Motivation: Independence Condition and Structural Information

The example in Figure 1 illustrates how the (conditional) (in)dependence relations differ between observed $\mathbf{X}$ and latent $\tilde{\mathbf{X}}$, and thus lead to biased discovery results. To put it generally, we have,

**Proposition 1** (*rare* d-separation). *Suppose variables follow random measurement error model defined in Equation* (1). *For disjoint sets of observed variables* $\mathbf{Z}, \mathbf{Y}, \mathbf{S}$ *and their respective latent ones* $\tilde{\mathbf{Z}}, \tilde{\mathbf{Y}}, \tilde{\mathbf{S}}$, *d-separation* $\mathbf{Z} \perp\!\!\!\perp_d \mathbf{Y}|\mathbf{S}$ *holds, only when marginally* $\tilde{\mathbf{Z}} \perp\!\!\!\perp_d \tilde{\mathbf{Y}}$, *and* $\tilde{\mathbf{Z}} \perp\!\!\!\perp_d \tilde{\mathbf{Y}}|\tilde{\mathbf{S}}$ *hold.*

By *'rare'* we mean that the d-separation patterns among $\tilde{\mathbf{X}}$ usually do not hold among $\mathbf{X}$ (except for *rare* marginal ones), since the observed variables are not causes of any other (though the latent variables they intend to measure might be). For example, consider the underlying $\tilde{G}$ to be chain structure (Figure 2a) and fully connected DAG (Figure 2c). There exists no (conditional) independence on either, and PC algorithm will output just a fully connected skeleton on both cases. Then, without (conditional) independence (which is non-parametric) to be directly used, can we *create independence*, by leveraging the parametric assumption (LiNGAM) and benefit from non-Gaussianity?

Naturally we recall the Independent Noise (IN) condition proposed in Direct-LiNGAM [40]:

**Definition 1** (IN condition). Let $Y_i$ be a single variable and $\mathbf{Z}$ be a set of variables. Suppose variables follow LiNGAM. We say $(\mathbf{Z}, Y_i)$ satisfies IN condition, denoted by $\text{IN}(\mathbf{Z}, Y_i)$, if and only if the residual of regressing $Y_i$ on $\mathbf{Z}$ is statistically independent to $\mathbf{Z}$. Mathematically, let $\tilde{\omega}$ be the vector of regression coefficients, i.e., $\tilde{\omega} := \text{cov}(Y_i, \mathbf{Z}) \text{cov}(\mathbf{Z}, \mathbf{Z})^{-1}$; $\text{IN}(\mathbf{Z}, Y_i)$ holds iff $Y_i - \tilde{\omega}^\mathsf{T}\mathbf{Z} \perp\!\!\!\perp \mathbf{Z}$.

Here "cov" denotes the variance-covariance matrix. IN identifies exogenous (root) variables, based on which the causal ordering of variables can be determined (Lemma 1 in [40]). However, IN cannot be applied to measurement error model. With hidden confounders ($\tilde{\mathbf{X}}$) behind observed $\mathbf{X}$, independence between regressor and residual typically does not exist on any regression among $\mathbf{X}$. In fact, $\mathbf{X}$ follows errors-in-variables models [15, 7], for which the identifiability w.r.t. $\tilde{G}$ is not clear.

However, we might still benefit from this idea to leverage non-Gaussianity of exogenous noises. Consider the Figure 1 example and abstract it to $\tilde{X}_1 \xrightarrow{a} \tilde{X}_2 \xrightarrow{b} \tilde{X}_3$ with $\{\tilde{X}_i \xrightarrow{1} X_i\}_{i=1}^3$. Although IN does not hold on any of $\mathbf{X}$, interestingly, there exists a linear transformation of observations $bX_2 - X_3$, which contains only $\{\tilde{E}_3, E_2, E_3\}$ ($\tilde{E}_1$ and $\tilde{E}_2$ are cancelled out) and shares no common non-Gaussian noise term with $X_1$. Hence, by the Darmois–Skitovich theorem [21], $bX_2 - X_3$ is independent of $X_1$. This finding is echoed in Generalized Independent Noise (GIN [50]) condition:

**Definition 2** (GIN condition). Let $\mathbf{Z}$ and $\mathbf{Y}$ be two sets of random variables that follow LiNGAM. We say $(\mathbf{Z}, \mathbf{Y})$ satisfies the GIN condition, denoted by $\text{GIN}(\mathbf{Z}, \mathbf{Y})$, if and only if the following two conditions are satisfied: **1)** There exists nonzero solution vectors $\omega \in \mathbb{R}^{|\mathbf{Y}|}$ to equation $\text{cov}(\mathbf{Z}, \mathbf{Y})\omega = \mathbf{0}$, and **2)** Any such solution $\omega$ makes the linear transformation $\omega^\mathsf{T}\mathbf{Y}$ independent of $\mathbf{Z}$.

Here $|\mathbf{Y}|$ denotes the dimensionality of $\mathbf{Y}$. The intuition of GIN is that, despite no independent residual by normal regression, it is possible to realize independent *"pseudo-residuals"* [5] by regressing with *"reference variables"*. [50] shows IN as a special case of GIN (Proposition 2), and further gives graphical criteria (Theorem 2), based on which a recursive learning algorithm is developed to solve the latent-variable problem. Each latent variable is required to have at least two observations. Interestingly we find that, in measurement error models, if each latent variable $\tilde{X}_i$ has two measurements $X_{i_1}, X_{i_2}$, then the GIN condition can be readily used to *fully identify* the structure of $\tilde{G}$, which is already a breakthrough over existing methods [42, 47, 46, 23]. See Appendix B for the whole procedure. With regard to our more challenging task where each $\tilde{X}_i$ has only one measurement $X_i$, a natural question is that, can GIN also help? Given the example in Figure 1 illustrated above, the answer seems to be affirmative: $\text{GIN}(\{X_i\}, \mathbf{X}\backslash\{X_i\})$ only holds for $i = 1$, so the root can be identified. More generally:

**Example 1** (GIN on chain structure). Consider cases where the underlying graph $\tilde{G}$ is a chain structure with $n$ ($n \geq 3$) vertices and directed edges $\{\tilde{X}_i \to \tilde{X}_{i+1}\}_{i=1}^{n-1}$. Figure 2a is an example with $n = 5$. We find that $\text{GIN}(\mathbf{Z} = X_1, \mathbf{Y} = X_{2,3,4,5})$ holds (where $X_{2,3,4,5}$ denotes $\{X_2, X_3, X_4, X_5\}$; same below), with solution $\omega = [-bx - bcy - bcdz, x, y, z]^\mathsf{T}$, $x, y, z \in \mathbb{R}$. $\omega^\mathsf{T}\mathbf{Y}$ cancels noise components in $\mathbf{Y}$ that are also shared by $\mathbf{Z}$, and thus $\omega^\mathsf{T}\mathbf{Y} \perp\!\!\!\perp \mathbf{Z}$. However, $\text{GIN}(X_i, \mathbf{X}\backslash\{X_i\})$ is violated for any other $i \neq 1$ (see Example 12 for a detailed derivation). With this asymmetry, the latent root $\tilde{X}_1$ can be identified. Furthermore, $\text{GIN}(X_i, X_{i+1,\cdots,n})$ holds for any $i = 1, \cdots, n - 2$.

Example 1 might give us an intuition that by recursively testing GIN (over the newly found subroot and the remaining vari-

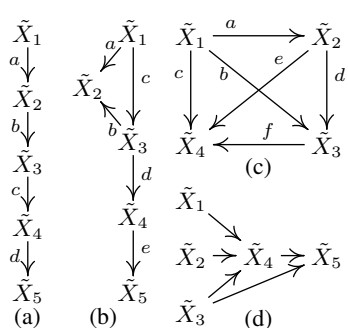

Figure 2: Graph structure $\tilde{G}$ examples. For simplicity, observed $\mathbf{X}$, measurement edges are omitted.

ables), we could identify the causal ordering of first $n-2$ variables for any DAG. However, this is over-optimistic thanks to the sparsity of chain structure. Consider a denser structure:

**Example 2** (GIN on fully connected DAG). Consider cases where $\tilde{G}$ is a fully connected DAG with $n$ ($n \geq 3$) vertices and directed edges $\{\tilde{X}_i \to \tilde{X}_j\}$ for every $i < j$. Figure 2c is an example with $n = 4$. We first find that $\mathrm{GIN}(X_1, X_{2,3,4})$ holds. However, in contrast to the chain structure, $\mathrm{GIN}(X_2, X_{3,4})$ does not hold - there is no way to cancel both $\tilde{E}_{1,2}$ from $X_{3,4}$. More generally we have: $\mathrm{GIN}(X_1, X_{2,\cdots,n})$, $\mathrm{GIN}(X_{1,2}, X_{3,\cdots,n})$, $\cdots$, $\mathrm{GIN}(X_{1,\cdots,k}, X_{k+1,\cdots,n})$, $k = \lfloor (n-1)/2 \rfloor$.

Since a fully connected DAG is the densest extreme, Example 2 might give us an intuition that GIN could identify at least the causal ordering of the first half of variables. Unfortunately, this is still over-optimistic, since we could not know beforehand the structure type of $\tilde{G}$.

**Example 3** (GIN on chain structure with *triangular head*). Figure 2b shows a variation of chain structure, with edges $\tilde{X}_1 \to \tilde{X}_2, \tilde{X}_1 \to \tilde{X}_3, \tilde{X}_3 \to \tilde{X}_2$, and $\{\tilde{X}_i \to \tilde{X}_{i+1}\}_{i=3}^{n-1}$. We name it "chain structure with *triangular head*". Interestingly, $\mathrm{GIN}(X_2, X_{3,\cdots,n})$, which is satisfied on Figure 2a, is also satisfied here. E.g., in Figure 2b ($n = 5$), the solution $\omega = [-dx - dey, x, y]^{\mathsf{T}}$, $x, y \in \mathbb{R}$. $\omega^{\mathsf{T}}\mathbf{Y}$ cancels $\tilde{E}_{1,3}$ from $\mathbf{Y}$, and thus $\omega^{\mathsf{T}}\mathbf{Y} \perp\!\!\!\perp \mathbf{Z}$. Actually, the chain structure with *triangular head* $\tilde{G}_{\triangledown\mathbb{C}}$ is *unidentifiable* with chain structure $\tilde{G}_{\mathbb{C}}$ w.r.t. GIN conditions, i.e., for any two sets of observed variables $\mathbf{Z}, \mathbf{Y} \subseteq \mathbf{X}$, $\mathrm{GIN}(\mathbf{Z}, \mathbf{Y})$ holds on $\tilde{G}_{\triangledown\mathbb{C}}$ if and only if $\mathrm{GIN}(\mathbf{Z}, \mathbf{Y})$ holds on $\tilde{G}_{\mathbb{C}}$. Consequently, if directly using any recursive algorithm by GIN as in Example 1, the output causal ordering would still be $\tilde{X}_1, \tilde{X}_2, \tilde{X}_3, \cdots$, which is incorrect due to $\tilde{X}_3 \to \tilde{X}_2$ in the triangular head.

The above examples show that it is not as simple as it seems to use GIN for one-measurement model: only part of the causal ordering can be identified, and worse yet, rather complicated error correction is needed to deal with possible incorrect orderings. However, after a closer look at the unidentifiable examples above, we find that actually more information can be uncovered beyond the GIN condition:

**Example 4** (Asymmetry beyond GIN). **1)** Consider Example 2 where only the root variable $\tilde{X}_1$ can be identified by GIN and $\tilde{X}_{2,3,4}$ are *unidentifiable*, i.e., permutation of the labeling of $X_{2,3,4}$ will still preserve the GIN condition over any two subsets $\mathbf{Z}, \mathbf{Y}$. However, there actually exists an asymmetry between $X_2$ and $X_{3,4}$: we could construct linear transformation of $X_{1,3,4}$: $\frac{cd-be}{d}X_1 + \frac{df+e}{d}X_3 - X_4$ s.t. it is independent of $X_2$, while there exists no such linear transformation of $X_{1,2,4}$ to be independent of $X_3$, and also for $X_{1,2,3}$ to $X_4$. **2)** Consider Examples 1 and 3 where $\tilde{G}_{\mathbb{C}}$ and $\tilde{G}_{\triangledown\mathbb{C}}$ are unidentifiable w.r.t. GIN conditions. Let $\mathbf{Z} := X_4$ and $\mathbf{Y} := X_{1,2,3}$, $\mathrm{GIN}(\mathbf{Z}, \mathbf{Y})$ is violated on both graphs. However, an asymmetry actually exists: on $\tilde{G}_{\triangledown\mathbb{C}}$, we could construct $aX_1 - X_2 + bX_3$ (which cancels $\tilde{E}_{1,3}$) to be independent to $X_4$, while this is impossible on $\tilde{G}_{\mathbb{C}}$.

To put simply, the motivation of independent *"pseudo-residual"* behind GIN actually limits the power of non-Gaussianity, with the coefficients vector $\omega$ only characterized from variance-covariance matrix (2nd-order). There are actually two cases for $\mathrm{GIN}(\mathbf{Z}, \mathbf{Y})$ to be violated: 1) though *not all* solution $\omega$ makes $\omega^{\mathsf{T}}\mathbf{Y} \perp\!\!\!\perp \mathbf{Z}$, there *exists* non-zero $\omega$ s.t. $\omega^{\mathsf{T}}\mathbf{Y} \perp\!\!\!\perp \mathbf{Z}$, and 2) there naturally exists *no* non-zero $\omega$ s.t. $\omega^{\mathsf{T}}\mathbf{Y} \perp\!\!\!\perp \mathbf{Z}$. The original GIN cannot distinguish between these two cases. Hence in this paper, we first aim to distinguish between the two, generalizing GIN condition to TIN condition.

## 3 With Transformed Independent Noise Condition

In the above discussion, one can see that the presence of measurement error affects the conditional independence relations among the variables and the independent noise condition. However, we will show in Theorem 3 (§4) that a specific type of independence conditions are shared between the underlying error-free variables and the measured variables with error. In this section, we will formulate such independence conditions and investigate their graphical implications for error-free variables (i.e., variables generated by the LiNGAM without measurement error). In §4, we will then extend the results to the measured variables. Please note that in contrast to other sections, the notation used in this section, including $\mathbf{X}$, $\mathbf{Y}$, and $\mathbf{S}$, denotes *error-free variables* generated by the LiNGAM.

### 3.1 Notations

Let $G$ be a directed acyclic graph with the vertex set $V(G) = [n] := \{1, 2, \cdots, n\}$ and edge set $E(G)$. A directed path $P = (i_0, i_1, \cdots, i_k)$ in $G$ is a sequence of vertices of $G$ where there is a directed edge from $i_j$ to $i_{j+1}$ for any $0 \leq j \leq k - 1$. We use notation $i \rightsquigarrow j$ to show that there

exists a directed path from vertex $i$ to $j$. Note that a single vertex is also a directed path, i.e., $i \rightsquigarrow i$ holds. Let $\mathbf{Z} \subseteq [n]$ be a subset of vertices. Define ancestors $\mathrm{Anc}(\mathbf{Z}) := \{j | \exists i \in \mathbf{Z}, j \rightsquigarrow i\}$. Note that $\mathbf{Z} \subseteq \mathrm{Anc}(\mathbf{Z})$. Further let $\mathbf{S}$ be a subset of vertices. We use notation $i \overset{\mathbf{S}}{\rightsquigarrow} j$ to show that there exists a directed path from vertex $i$ to $j$ without passing through $\mathbf{S}$, i.e., there exists a directed path $P = (i, m_0, \cdots, m_k, j)$ in $G$ s.t. $i, j \notin \mathbf{S}$ and $m_l \notin \mathbf{S}$ for any $0 \le l \le k$. Define *ancestors outside* $\mathbf{S}$ accordingly: for two vertex sets $\mathbf{Y}, \mathbf{S}$, denote ancestors of $\mathbf{Y}$ that have directed paths into $\mathbf{Y}$ without passing through $\mathbf{S}$ as $\mathrm{Anc}_{\mathrm{out}(\mathbf{S})}(\mathbf{Y}) := \{j | \exists i \in \mathbf{Y}, j \overset{\mathbf{S}}{\rightsquigarrow} i\}$. Note that the graphical definitions here can also be translated to *trek* [47] language (see Appendix C for details).

Assume random variables $\mathbf{X} := \{X_i\}_{i \in [n]}$ are generated by LiNGAM w.r.t. graph $G$, i.e.,

$$\mathbf{X} = \mathbf{A}\mathbf{X} + \mathbf{E} = \mathbf{B}\mathbf{E}, \text{ with } \mathbf{B} = (\mathbf{I} - \mathbf{A})^{-1}. \tag{2}$$

where $\mathbf{E} = \{E_i\}_{i \in [n]}$ are corresponding mutually independent exogenous noises. $\mathbf{A}$ is the adjacency matrix where entry $\mathbf{A}_{j,i}$ is the linear weight of direct causal effect of variable $X_i$ on $X_j$. $\mathbf{A}_{j,i} \ne 0$ if and only if there exists edge $i \to j$. $\mathbf{X}$ can also be written directly as a mixture of exogenous noises $\mathbf{X} = \mathbf{B}\mathbf{E}$. If the entry of mixing matrix $\mathbf{B}_{j,i} \ne 0$, then $i \in \mathrm{Anc}(\{j\})$. Note that here and in what follows, we use boldface letters $\mathbf{A}, \mathbf{B}$ to denote matrices, and use boldface letters $\mathbf{S}, \mathbf{W}, \mathbf{X}, \mathbf{Y}, \mathbf{Z}$ with notation abuse: it can denote vertices set, respective random variables set, or random vector. When we say "two variables sets $\mathbf{Z}, \mathbf{Y}$", if not otherwise specified, $\mathbf{Z}, \mathbf{Y}$ need not be disjoint.

### 3.2 Independent Linear Transformation Subspace and its Characterization

We first give the definition and characterization of the *independent linear transformation subspace*.

**Definition 3** (Independent linear transformation subspace). Let $\mathbf{Z}$ and $\mathbf{Y}$ be two subsets of random variables. Suppose the variables follow the linear non-Gaussian acyclic causal model. Denote:

$$\Omega_{\mathbf{Z};\mathbf{Y}} := \{\omega \in \mathbb{R}^{|\mathbf{Y}|} \mid \omega^{\mathsf{T}}\mathbf{Y} \perp\!\!\!\perp \mathbf{Z}\}. \tag{3}$$

By the property of independence, $\Omega_{\mathbf{Z};\mathbf{Y}}$ is closed under scalar multiplication and addition, and thus is a subspace in $\mathbb{R}^{|\mathbf{Y}|}$. In fact, $\Omega_{\mathbf{Z};\mathbf{Y}}$ can be characterized as a nullspace as follows:

**Theorem 1** (Characterization of $\Omega_{\mathbf{Z};\mathbf{Y}}$). *For two variables subsets $\mathbf{Z}$ and $\mathbf{Y}$, $\Omega_{\mathbf{Z};\mathbf{Y}}$ satisfies:*

$$\Omega_{\mathbf{Z};\mathbf{Y}} = \mathrm{null}(\mathbf{B}_{\mathbf{Y}, \mathrm{nzcol}(\mathbf{B}_{\mathbf{Z},:})}^{\mathsf{T}}). \tag{4}$$

*where $\mathrm{null}(\cdot)$ denotes nullspace. $\mathbf{B}_{\mathbf{Y}, \mathrm{nzcol}(\mathbf{B}_{\mathbf{Z},:})}$ denotes the submatrix of mixing matrix $\mathbf{B}$, with rows indexed by $\mathbf{Y}$ and columns indexed by $\mathrm{nzcol}(\mathbf{B}_{\mathbf{Z},:})$. $\mathrm{nzcol}(\mathbf{B}_{\mathbf{Z},:})$ denotes the column indices where the submatrix $\mathbf{B}_{\mathbf{Z},:}$ has non-zero entries. $\mathrm{nzcol}(\mathbf{B}_{\mathbf{Z},:})$ actually corresponds to the exogenous noises that constitute $\mathbf{Z}$. Particularly, if assuming "if $i \rightsquigarrow j$ then $\mathbf{B}_{j,i} \ne 0$", then, $\mathrm{nzcol}(\mathbf{B}_{\mathbf{Z},:}) = \mathrm{Anc}(\mathbf{Z})$.*

Proof of Theorem 1 is straight-forward by the Darmois–Skitovich theorem [21]: linear transformation $\omega^{\mathsf{T}}\mathbf{Y} \perp\!\!\!\perp \mathbf{Z}$ if and only if $\omega^{\mathsf{T}}\mathbf{Y}$ shares no common non-Gaussian exogenous noise components with $\mathbf{Z}$.

**Example 5** (Revisiting examples in §2 from $\Omega_{\mathbf{Z};\mathbf{Y}}$ perspective). For illustration, now we revisit the examples in §2 from the perspective of independent linear transformation subspace.

$$\begin{bmatrix} 1 & 0 & 0 & 0 & 0 \\ a & 1 & 0 & 0 & 0 \\ ab & b & 1 & 0 & 0 \\ abc & bc & c & 1 & 0 \\ abcd & bcd & cd & d & 1 \end{bmatrix}, \begin{bmatrix} 1 & 0 & 0 & 0 & 0 \\ a+bc & 1 & b & 0 & 0 \\ c & 0 & 1 & 0 & 0 \\ cd & 0 & d & 1 & 0 \\ cde & 0 & de & e & 1 \end{bmatrix}, \begin{bmatrix} 1 & 0 & 0 & 0 \\ a & 1 & 0 & 0 \\ ad+b & d & 1 & 0 \\ a(df+e)+bf+c & df+e & f & 1 \end{bmatrix} \tag{5}$$

Equation (5) shows the corresponding mixing matrix $\mathbf{B}$ for graph $\tilde{G}$ in Figures 2a to 2c, respectively. Suppose we have access to underlying variables $\tilde{X}_i$ and only focus on $G$. Colored blocks denote submatrices of $\mathbf{B}$. **1)** For the fully connected DAG (Figure 2c, the right matrix), to identify the root $\tilde{X}_1$, $\mathrm{GIN}(\tilde{X}_1, \tilde{X}_{2,3,4})$ is satisfied, corresponding to $\Omega_{\mathbf{Z};\mathbf{Y}} = \mathrm{null}(\ \ ^{\mathsf{T}})$. For $(\mathbf{Z}, \mathbf{Y}) := (\tilde{X}_2, \tilde{X}_{3,4})$ or $(\tilde{X}_4, \tilde{X}_{1,2,3})$, there exists no non-zero $\omega$ s.t. $\omega^{\mathsf{T}}\mathbf{Y} \perp\!\!\!\perp \mathbf{Z}$, because the lower part and are full row rank. However, if we set $(\mathbf{Z}, \mathbf{Y}) := (\tilde{X}_2, \tilde{X}_{1,3,4})$, we actually have $\tilde{X}_2 \perp\!\!\!\perp \frac{cd-be}{d}\tilde{X}_1 + \frac{df+e}{d}\tilde{X}_3 - \tilde{X}_4$, because the stacked two parts of has rank $2 < 3$ - though GIN is still violated because $\mathrm{cov}(\mathbf{Z}, \mathbf{Y})$ has rank $1 < 2$. **2)** For the chain structure $\tilde{G}_\mathbb{C}$ (Figure 2a,

the left matrix) and chain with triangular head $\tilde{G}_{\triangledown\mathbb{C}}$ (Figure 2b, the middle matrix), GIN is satisfied on $(\tilde{X}_1, \tilde{X}_{2,3,4,5})$, $\tilde{X}_2, \tilde{X}_{3,4,5})$, $(\tilde{X}_3, \tilde{X}_{4,5})$, with the $\mathbf{B}_{\mathbf{Y},\mathrm{Anc}(\mathbf{Z})}$ submatrices being ▮, ▮, ▮ respectively. Note that though the shape of submatrices are different between $\tilde{G}_{\mathbb{C}}$ and $\tilde{G}_{\triangledown\mathbb{C}}$, their ranks are always equal, and is thus unidentifiable by GIN. However, let $(\mathbf{Z}, \mathbf{Y}) := (\tilde{X}_4, \tilde{X}_{1,2,3})$, an asymmetry actually exists: in $\tilde{G}_{\triangledown\mathbb{C}}$, $\tilde{X}_4 \perp\!\!\!\perp a\tilde{X}_1 - \tilde{X}_2 + b\tilde{X}_3$, because in the right matrix ▮ has rank $2 < 3$, while in the left matrix ▮ is full row rank, so there is no such independence in $\tilde{G}_{\mathbb{C}}$.

## 3.3 Graphical Criteria of Independent Linear Transformation Subspace

§3.2 characterizes $\Omega_{\mathbf{Z};\mathbf{Y}}$ by submatrix of $\mathbf{B}$, which entails information on graph structure and edge weights. The following sections go one step further, explicitly exhibit the graphical criteria of $\Omega_{\mathbf{Z};\mathbf{Y}}$, and investigate how $\Omega_{\mathbf{Z};\mathbf{Y}}$ could help to identify the causal structure. We first give the assumption:

**Assumption 1** (Rank faithfulness). Denote by $\mathcal{B}(G)$ the parameter space of mixing matrix $\mathbf{B}$ consistent with the DAG $G$. For any two subsets of variables $\mathbf{Z}, \mathbf{Y} \subseteq \mathbf{X}$, we assume that

$$\mathrm{rank}(\mathbf{B}_{\mathbf{Y},\mathrm{Anc}(\mathbf{Z})}) = \max_{\mathbf{B}' \in \mathcal{B}(G)} \mathrm{rank}(\mathbf{B}'_{\mathbf{Y},\mathrm{Anc}(\mathbf{Z})}). \tag{6}$$

Roughly speaking, we assume there are no edge parameter couplings to produce coincidental low rank. This is slightly stronger than "$\mathrm{nzcol}(\mathbf{B}_{\mathbf{Z},:}) = \mathrm{Anc}(\mathbf{Z})$". See Appendix F.1 for elaboration. In other words, the graphical criteria holds on a dense open subset of the edge parameter space. Note that Assumption 1 is the only other parametric assumption we make besides LiNGAM throughout the paper, where violation of Assumption 1 is of Lebesgue measure 0, and LiNGAM is testable.

Graphically, we first define *vertex cut*, and then give the graphical criteria based on it:

**Definition 4** (Vertex cut). Let $\mathbf{W}, \mathbf{Y}, \mathbf{S}$ be three vertices subsets of $V(G)$ which need not be disjoint. We say that $\mathbf{S}$ is a *vertex cut* from $\mathbf{W}$ to $\mathbf{Y}$, if and only if there exists no directed paths in $G$ from $\mathbf{W}$ to $\mathbf{Y}$ without passing through $\mathbf{S}$. With basic notations in §3.1, the following statements are equivalent: **1)** $\mathbf{S}$ is a vertex cut from $\mathbf{W}$ to $\mathbf{Y}$; **2)** $\forall i \in \mathbf{W}, j \in \mathbf{Y}, i \not\rightsquigarrow j$ does not hold; **3)** $\mathrm{Anc}_{\mathrm{out}(\mathbf{S})}(\mathbf{Y}) \cap \mathbf{W} = \varnothing$; **4)** $\mathbf{S}$'s removal from $G$ ensures there is no directed paths from $\mathbf{W}\backslash\mathbf{S}$ to $\mathbf{Y}\backslash\mathbf{S}$.

More details on vertex cut are in Appendix C. Note that trivially $\mathbf{W}$ itself and $\mathbf{Y}$ itself are vertex cuts.

**Theorem 2** (Graphical criteria of $\Omega_{\mathbf{Z};\mathbf{Y}}$). *Let $\mathbf{Z}, \mathbf{Y}$ be two subsets of variables (vertices), we have:*

$$|\mathbf{Y}| - \dim(\Omega_{\mathbf{Z};\mathbf{Y}}) = \min\{|\mathbf{S}| \mid \mathbf{S} \text{ is a vertex cut from } \mathrm{Anc}(\mathbf{Z}) \text{ to } \mathbf{Y}\}. \tag{7}$$

*where $\dim(\Omega_{\mathbf{Z};\mathbf{Y}})$ denotes the dimension of the subspace $\Omega_{\mathbf{Z};\mathbf{Y}}$, i.e., the degree of freedom of $\omega$.*

By Theorem 1, $|\mathbf{Y}| - \dim(\Omega_{\mathbf{Z};\mathbf{Y}})$ is exactly the rank of $\mathbf{B}_{\mathbf{Y},\mathrm{Anc}(\mathbf{Z})}$. Theorem 2 can then be proved by a combinatorial interpretation of mixing matrices' determinants. See Appendix A for details.

From the graphical view, a vertex cut $\mathbf{S}$ from $\mathrm{Anc}(\mathbf{Z})$ to $\mathbf{Y}$ means that the causal effect from $\mathbf{Z}$ and the common causes of $\mathbf{Z}$ and $\mathbf{Y}$, if there is any, must affect $\mathbf{Y}$ through $\mathbf{S}$ - there is no any bypass. To interpret in *trek-separation* [47] language, it is equivalent to "$(\varnothing, \mathbf{S})$ *t-separates* $(\mathbf{Z}, \mathbf{Y})$"[2].

From the noise view, the noise components that constitute variables $\mathbf{Z}$ are exactly the exogenous noises corresponding to vertices $\mathrm{Anc}(\mathbf{Z})$. All these noises must contribute to $\mathbf{Y}$ (if any) via $\mathbf{S}$, and thus $\mathbf{Y}$ can be written as $\mathbf{Y} = L\mathbf{S} + \mathbf{E}'_{\mathbf{Y}}$, where $L$ denotes a linear transformation, and $\mathbf{E}'_{\mathbf{Y}} \perp\!\!\!\perp \mathbf{Z}$. Denote by $\mathbf{S}^*_{\mathbf{Z};\mathbf{Y}}$ the *critical vertex cut* from $\mathbf{Z}$ to $\mathbf{Y}$[3], the noise components of $\omega^\mathsf{T}\mathbf{Y}$ is exactly the exogenous noises corresponding to $\mathrm{Anc}_{\mathrm{out}(\mathbf{S}^*_{\mathbf{Z};\mathbf{Y}})}(\mathbf{Y})$.

## 3.4 Formal Definition of TIN

With the graphical criteria given in §3.3, we could use it for structure inference as long as we have the (dimension of) independent linear transformation subspace $\Omega_{\mathbf{Z};\mathbf{Y}}$. In §5 we will comprehensively discuss methods to estimate $\Omega_{\mathbf{Z};\mathbf{Y}}$, while for now, we could just safely suppose we have $\Omega_{\mathbf{Z};\mathbf{Y}}$: since independence is testable, theoretically one could get $\Omega_{\mathbf{Z};\mathbf{Y}}$ at least by traversing over $\mathbb{R}^{|\mathbf{Y}|}$.

Figure 3: Illustration of vertex cut $\mathbf{S}$ from $\mathrm{Anc}(\mathbf{Z})$ to $\mathbf{Y}$.

[2]See Definition 2.7 in [47] for definition of *t-separation*. See Appendix C for details.
[3]See Appendix D.1 for definition. Roughly speaking, *"critical"* means a *smallest* and *last* vertex cut.

Table 1: Examples of TIN on different $(\mathbf{Z}, \mathbf{Y})$ pairs over different graph structures in Figure 2.

| $(\mathbf{Z}, \mathbf{Y})$ | $(\{X_1, X_2\}, \{X_3, X_4, X_5\})$ | | | | $(\{X_1, X_2\}, \{X_4, X_5\})$ | | | | $(\{X_3\}, \{X_1, X_2, X_4, X_5\})$ | | | | $(\{X_1, X_4\}, \{X_3, X_4, X_5\})$ | | | |
|---|---|---|---|---|---|---|---|---|---|---|---|---|---|---|---|---|
| Graph ID | (a) | (b) | (c) | (d) | (a) | (b) | (c) | (d) | (a) | (b) | (c) | (d) | (a) | (b) | (c) | (d) |
| $\mathrm{TIN}(\mathbf{Z}, \mathbf{Y})$ | 1 | 1 | 2 | 1 | 1 | 1 | 2 | 1 | 3 | 2 | 3 | 1 | 2 | 2 | 3 | 2 |
| $\dim(\Omega_{\mathbf{Z};\mathbf{Y}})$ | 2 | 2 | 1 | 2 | 1 | 1 | 0 | 1 | 1 | 1 | 1 | 3 | 1 | 1 | 0 | 1 |
| $\mathrm{Anc}(\mathbf{Z})$ | $X_{1,2}$ | $X_{1,2,3}$ | $X_{1,2}$ | $X_{1,2}$ | $X_{1,2}$ | $X_{1,2,3}$ | $X_{1,2}$ | $X_{1,2}$ | $X_{1,2,3}$ | $X_{1,3}$ | $X_{1,2,3}$ | $X_3$ | $X_{1,2,3,4}$ | $X_{1,3,4}$ | $X_{1,2,3,4}$ | $X_{1,2,3,4}$ |
| $\mathbf{S}^*_{\mathbf{Z},\mathbf{Y}}$ | $X_3$ | $X_3$ | $X_{1,2}$ | $X_4$ | $X_4$ | $X_4$ | $X_{4,5}$ | $X_4$ | $X_{1,2,4}$ | $X_{1,3}$ | $X_{1,2,3}$ | $X_3$ | $X_{3,4}$ | $X_{3,4}$ | $X_{3,4,5}$ | $X_{3,4}$ |
| $\mathrm{A}_{o(\mathbf{S}^*)}(\mathbf{Y})$ | $X_{4,5}$ | $X_{4,5}$ | $X_{3,4,5}$ | $X_{3,5}$ | $X_5$ | $X_5$ | $\varnothing$ | $X_{3,5}$ | $X_5$ | $X_{2,4,5}$ | $X_{4,5}$ | $X_{1,2,4,5}$ | $X_5$ | $X_5$ | $\varnothing$ | $X_5$ |

**Definition 5** (TIN function). Let $\mathbf{Z}$ and $\mathbf{Y}$ be two subsets of observed random variables. Suppose variables follow LiNGAM. We define a function TIN as follows:

$$\mathrm{TIN}(\mathbf{Z}, \mathbf{Y}) := |\mathbf{Y}| - \dim(\Omega_{\mathbf{Z};\mathbf{Y}}). \tag{8}$$

TIN is a function that takes as input two random vectors $\mathbf{Z}, \mathbf{Y}$ and *returns an integer* in range $[0, |\mathbf{Y}|]$. Note that this is different to IN or GIN, which returns only a bool value (satisfied or not). GIN can be viewed as a special case of TIN, i.e., $\mathrm{GIN}(\mathbf{Z}, \mathbf{Y})$ is satisfied if and only if $\mathrm{TIN}(\mathbf{Z}, \mathbf{Y}) = \mathrm{rank}(\mathrm{cov}(\mathbf{Z}, \mathbf{Y})) < |\mathbf{Y}|$ (where $\mathrm{rank}(\mathrm{cov}(\cdot))$ can be characterized by [47]). IN can also be viewed as a special case of TIN, i.e., $\mathrm{IN}(\mathbf{Z}, Y_i)$ is satisfied if and only if $\mathrm{TIN}(\mathbf{Z}, \mathbf{Z} \cup \{Y_i\}) = |\mathbf{Z}|$.

**Example 6** (Review TIN graphical criteria on graphs). For better understanding of TIN, we demonstrate some representative examples of TIN over different graph structures. Results are shown in Table 1. We use the four graph structures in Figure 2. For illustration, assume we have access to latent variables to directly conduct TIN over $\tilde{\mathbf{X}}$. Graphical criteria correspond to $\tilde{G}$. For readability, we omit all ˜ notation. We assume there are 5 vertices in the fully connected DAG (Figure 2c): consider for example, $\mathrm{GIN}(X_3, X_{1,2,4,5})$ does not hold, since $\mathrm{rank}(\mathrm{cov}(\mathbf{Z}, \mathbf{Y}))$ is only one (restricted by $\mathbf{Z}$ size). However, there exists $\omega$ with degree of freedom 1 s.t. $\omega^{\mathsf{T}}\mathbf{Y} \perp\!\!\!\perp \mathbf{Z}$ - this corresponds to a 3-variables vertex cut $X_{1,2,3}$ ($\mathrm{Anc}(\mathbf{Z})$ itself). Consider on Figure 2d, $\mathrm{TIN}(X_{1,2}, X_{4,5}) = 1$, corresponding to the 1-variable vertex cut $X_4$. This example shows that vertex cut does not necessarily yield a d-separation pattern (but blocking, instead), since here $X_5 \not\perp\!\!\!\perp X_1 | X_4$.

The graphical criteria for TIN over any two sets of variables $\mathbf{Z}, \mathbf{Y}$ are given as above. Interestingly, we find that a special type of TIN is particularly simple in form and informative for structure inference:

**Lemma 1** (Graphical criteria of TIN on one-and-others). *Assume we have access to all variables $\mathbf{X}$ on a DAG $G$. For each singleton variable $X_i \in \mathbf{X}$, let $\mathbf{Z} := \{X_i\}$ and $\mathbf{Y} := \mathbf{X}\setminus\{X_i\}$, we have,*

$$\mathrm{TIN}(\{X_i\}, \mathbf{X}\setminus\{X_i\}) = \begin{cases} |\mathrm{Anc}(\{X_i\})| & X_i \text{ is a non-leaf node} \\ |\mathrm{Anc}(\{X_i\})| - 1 & X_i \text{ is a leaf node} \end{cases} \tag{9}$$

Due to a page limit, here we only give the main criteria of TIN. See Appendix D for more properties.

## 4   TIN **Condition-Based Method for Measurement Error Models**

In §3 we propose TIN condition over general LiNGAM model and give its graphical criteria. In this section, we aim to exploit TIN on our specific task of interest: measurement error models.

### 4.1   Identifiability of Ordered Group Decomposition

Under our problem setting where $\tilde{\mathbf{X}}$ follows LiNGAM, identifiability results can greatly benefit from the non-Gaussianity of data. [54] shows that the *ordered group decomposition* of $\tilde{G}$ is identifiable. First review definitions:

**Definition 6** (Pure leaf child). On a DAG $G$, a vertex $j$ is said to be a "pure leaf child" of another vertex $i$, iff $j$ is a leaf node with only one parent, $i$.

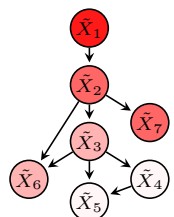

Figure 4: An example of ordered group decomposition.

Particularly, if a variable $\tilde{X}_j$ is a pure leaf child of $\tilde{X}_i$ in $\tilde{G}$, then $\tilde{X}_j$ and $\tilde{X}_j$ are naturally unidentifiable (e.g., $\tilde{X}_2$ and $\tilde{X}_7$ in Figure 4). The reason is that the exogenous noise $\tilde{E}_j$ of leaf $\tilde{X}_j$ does not contribute to any other variables, just like its measurement noise $E_j$. Consequently, $X_i$ and $X_j$ can be viewed as two equally positioned measurements of $\tilde{X}_i$, without any asymmetry.

**Definition 7** (Ordered group decomposition). Consider the underlying causal model $\tilde{G}$. The ordered group decomposition can be defined by the following procedure: at each step, remove root vertices and their pure leaf children nodes (if any) from graph, and append the removed ones as a new group. Repeat this procedure to remove root vertices from the remaining graph, until the graph is empty.

This procedure is equivalent to Definition 2 in [54][4]. See Figure 4: roots are removed from graph in order (color dark to light). The ordered group decomposition is $\tilde{X}_1 \rightarrow \tilde{X}_{2,7} \rightarrow \tilde{X}_{3,6} \rightarrow \tilde{X}_{4,5}$.

## 4.2 Exploit TIN to Identify Ordered Group Decomposition

Our task is to recover $\tilde{G}$ over $\tilde{\mathbf{X}}$ by testing TIN over $\mathbf{X}$. In notations above, $\tilde{\mathbf{X}}$ and $\mathbf{X}$ are strictly distinguished to denote unobserved and observed variables. However, actually this can be escaped:

**Theorem 3** (Equivalence of TIN over latent and observed variables). *For two **disjoint** observed variables subsets $\mathbf{Z}, \mathbf{Y}$, and their respective underlying latent variables subsets $\tilde{\mathbf{Z}}, \tilde{\mathbf{Y}}$,*

$$\mathrm{TIN}(\mathbf{Z}, \mathbf{Y}) = \mathrm{TIN}(\tilde{\mathbf{Z}}, \tilde{\mathbf{Y}}). \tag{10}$$

Theorem 3 can either be proved by graphical criteria, or by showing how the rank of submatrices of $\mathbf{B}$ is preserved among latent and observed variables. Interestingly, recall Proposition 1, we show the inequivalence of d-separation and IN condition held between $\tilde{\mathbf{X}}$ and $\mathbf{X}$ (*raw independence*). However, TIN condition, which essentially *finds transformed independence*, holds equivalently on latent and observed variables. With this equivalence, we can conduct TIN over observed variables $\mathbf{X}$ *just as if* we have access to the latent ones $\tilde{\mathbf{X}}$. Thus the problem can be restated without measurement error: assuming causal sufficiency, by only using TIN over disjoint $\mathbf{Z}, \mathbf{Y}$, to what extent is $G$ identifiable?

Under this equivalent problem, Lemma 1 can be used directly to identify the ordered group decomposition of $\tilde{G}$: for each singleton observed variable $X_i$, test TIN and assign an order $\mathrm{ord}(X_i) := \mathrm{TIN}(\{X_i\}, \mathbf{X} \backslash \{X_i\})$. Group variables with same $\mathrm{ord}$, and then sort the groups by their orders. Obviously, the ordered groups obtained by this procedure is consistent with Definition 7.

**Example 7** (TIN on Figure 4). $\mathrm{ord}(X_i)_{i=1}^7$ are respectively $1, 2, 3, 4, 4, 3, 2$ (can verify by characterization or graphically), so the group ordering is identified as $\tilde{X}_1 \rightarrow \tilde{X}_{2,7} \rightarrow \tilde{X}_{3,6} \rightarrow \tilde{X}_{4,5}$.

# 5 Estimating Linear Independent Transformation Subspace $\Omega_{\mathbf{Z};\mathbf{Y}}$

In the above sections we safely assume that we can always get $\Omega_{\mathbf{Z};\mathbf{Y}}$, since theoretically independence is testable and $\omega$ can be exhaustively traversed. In this section, we give practical methods to estimate $\Omega_{\mathbf{Z};\mathbf{Y}}$. Due to page limit we only give a summary for each. Please see Appendix E for details.

## 5.1 Tackling Down to Subsets of $\mathbf{Y}$

**Theorem 4** (TIN over $\mathbf{Y}$ subsets). *For two variables sets $\mathbf{Z}, \mathbf{Y}$, $\mathrm{TIN}(\mathbf{Z}, \mathbf{Y}) = k$ (assume $k > 0$), iff the following two conditions hold: 1) $\forall \mathbf{Y}' \subseteq \mathbf{Y}$ with $|\mathbf{Y}'| = k + 1$ (if any), there exists non-zero $\omega$ s.t. $\omega^\intercal \mathbf{Y}' \perp\!\!\!\perp \mathbf{Z}$; and 2) $\exists \mathbf{Y}' \subseteq \mathbf{Y}$ with $|\mathbf{Y}'| = k$, there exists no non-zero $\omega$ s.t. $\omega^\intercal \mathbf{Y}' \perp\!\!\!\perp \mathbf{Z}$.*

This transforms the task of estimating the *dimension* of $\Omega_{\mathbf{Z};\mathbf{Y}}$ to a simpler one: counting *size* of the subsets $\mathbf{Y}'$. Instead of *all* independence, here we only need to check *existence* of independence.

## 5.2 Constrained Independent Subspace Analysis (ISA)

Conduct Independent Subspace Analysis (ISA [48]) over variables $\mathbf{Z}$ and $\mathbf{Y}$ in the following form:

$$\mathbf{s} = \begin{bmatrix} \mathbf{I} & \mathbf{0} \\ \mathbf{0} & \mathbf{W_{YY}} \end{bmatrix} \begin{bmatrix} \mathbf{Z} \\ \mathbf{Y} \end{bmatrix}, \tag{11}$$

where the de-mixing matrix is masked to only update the lower-right $|\mathbf{Y}| \times |\mathbf{Y}|$ block $\mathbf{W_{YY}}$, with upper-left $|\mathbf{Z}| \times |\mathbf{Z}|$ block fixed as the identity and elsewhere fixed as zero. The independence between $\mathbf{W_{YY}}\mathbf{Y}$ as a group to $\mathbf{Z}$ is maximized. Since $\mathbf{W_{YY}}$ is invertible, its rows span the whole $\mathbb{R}^{|\mathbf{Y}|}$, so the maximum number of rows that achieves $\mathbf{W_{YY}}_{i,:}^\intercal \mathbf{Y} \perp\!\!\!\perp \mathbf{Z}$ is exactly the dimension of $\Omega_{\mathbf{Z};\mathbf{Y}}$.

## 5.3 Stacked Cumulants: Ranks Stopped Increasing

**Definition 8** (Stacked 2D slices of cumulants). For two variables sets $\mathbf{Z}, \mathbf{Y}$ and order $k \geq 2$, define:

$$\Psi_{\mathbf{Z};\mathbf{Y}}^{(k)} := \begin{bmatrix} \mathcal{C}_{\mathbf{Z},\mathbf{Y}}^{(2)\intercal} & \cdots & \mathcal{C}_{\mathbf{Z},\mathbf{Y}}^{(k)\intercal} \end{bmatrix}^\intercal, \text{ where } \mathcal{C}^{(k)} \text{ is matrix with } \mathcal{C}_{i,j}^{(k)} := \mathrm{cum}(\underbrace{X_i, \cdots, X_i}_{k-1 \text{ times}}, X_j). \tag{12}$$

$\Psi_{\mathbf{Z};\mathbf{Y}}^{(k)}$ is a $(k-1)|\mathbf{Z}| \times |\mathbf{Y}|$ matrix that vertically stacks 2D cumulants slices between $\mathbf{Z}, \mathbf{Y}$ with orders from 2 to $k$. When $k = 2$ it is $\mathrm{cov}(\mathbf{Z}, \mathbf{Y})$. Ranks of $\Psi_{\mathbf{Z};\mathbf{Y}}^{(k)}$ in the sequence $k = 2, 3, \ldots$ satisfies:

---

[4]There is actually trivial difference, depending on how case with multiple roots is considered. See Example 13.

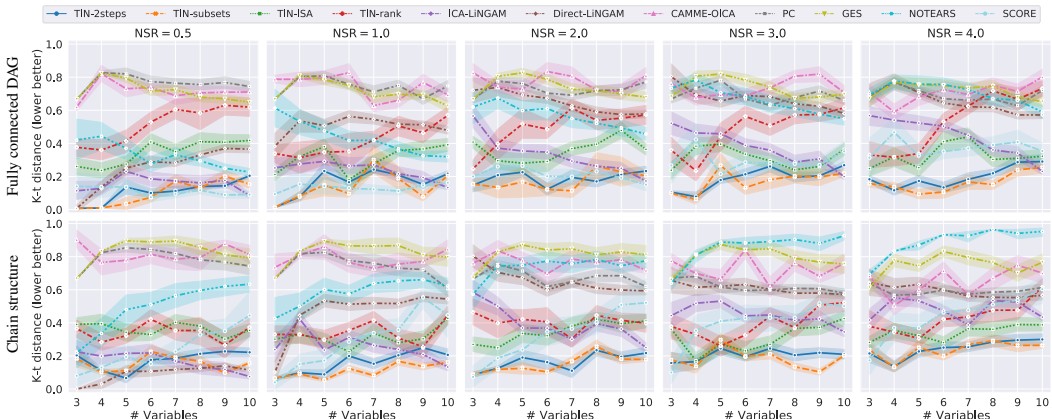

Figure 5: Distance to truth group ordering (lower better). 4 implementations of TIN + 7 competitors.

**Theorem 5** ($\mathrm{rank}(\Psi_{\mathbf{Z};\mathbf{Y}}^{(k)})$ stopped increasing). *For two variables sets* $\mathbf{Z}, \mathbf{Y}$, *there exists a finite order* $k \geq 2$ *s.t.* $\mathrm{rank}(\Psi_{\mathbf{Z};\mathbf{Y}}^{(k+1)}) = \mathrm{rank}(\Psi_{\mathbf{Z};\mathbf{Y}}^{(k)})$. *Moreover,* $\mathrm{TIN}(\mathbf{Z}, \mathbf{Y})$ *equals* $\mathrm{rank}(\Psi_{\mathbf{Z};\mathbf{Y}}^{(k)})$ *with this* $k$.

We will show detailed characterization and graphical criteria of $\Psi_{\mathbf{Z};\mathbf{Y}}^{(k)}$ in Appendix E.1.

### 5.4 TIN **in Two Steps: Solve Equations, and then Test for Independence**

Motivation of §5.3 is that independence yields zero cumulants (not only 2nd-order uncorrelatedness), i.e., $\Omega_{\mathbf{Z};\mathbf{Y}} \subseteq \mathrm{null}(\Psi_{\mathbf{Z};\mathbf{Y}}^{(k)})$, for any $k \geq 2$. Hence, we could solve equations introduced at each order $k$ and then check whether *all* solution $\omega \in \mathrm{null}(\Psi_{\mathbf{Z};\mathbf{Y}}^{(k)})$ makes $\omega^\mathsf{T}\mathbf{Y} \perp\!\!\!\perp \mathbf{Z}$ (similar to GIN procedure). More generally, $\Omega_{\mathbf{Z};\mathbf{Y}} \subseteq \mathrm{null}(\mathrm{cov}(f(\mathbf{Z}), \mathbf{Y}))$ for any real-valued function $f(\cdot)$. E.g., solve equations system $\{\mathrm{cov}(\log(\mathbf{Z}^2), \mathbf{Y})\omega = \mathbf{0};\ \mathrm{cov}(\sin(\mathbf{Z}), \mathbf{Y})\omega = \mathbf{0};\cdots\}$ and test whether $\omega^\mathsf{T}\mathbf{Y} \perp\!\!\!\perp \mathbf{Z}$ holds.

## 6 Experimental Results

In this section, we evaluate the performance of TIN in recovering the ordered group decomposition of $\tilde{G}$ under measurement error. Specifically, four implementations of TIN are evaluated: TIN-subsets(§5.1), TIN-ISA(§5.2), TIN-rank(§5.3), and TIN-2steps(§5.4). We compare our method with PC [43], GES [8], ICA-LiNGAM [38], Direct-LiNGAM [40], CAMME-OICA [54], NOTEARS [55], and SCORE [32]. Experiments are conducted on both synthetic and real-world data.

### 6.1 Synthetic Data

While our proposed method outputs correct ordered group decomposition for any $\tilde{G}$ (without assumptions on graph structure), in the following simulation we consider specifically two cases: fully connected DAG (Figure 2c) and chain structure (Figure 2a), of which the ordered group decomposition are both $\tilde{X}_1 \rightarrow \cdots \rightarrow \tilde{X}_{n-2}, \tilde{X}_{n-1,n}$. We consider $G$ with the number of vertices $n = 3, \cdots, 10$. Edges weights (i.e., the nonzero entries of matrix $\mathbf{A}$) are drawn uniformly from $[-0.9, -0.5] \cup [0.5, 0.9]$. Exogenous noises $\tilde{\mathbf{E}}$ are sampled from uniform $\cup[0, 1]$ to the power of $c$, $c \sim \cup[5, 7]$, and measurement errors are sampled from Gaussian $\mathcal{N}(0, 1)$ to the power of $c$, $c \sim \cup[2, 4]$. Sample size is $5,000$. Observations are generated by $X_i = \tilde{X}_i + E_i$. To show the effect of measurement error, we simulate with noise-to-signal ratio $\mathrm{NSR} := \mathrm{var}(E_i)/\mathrm{var}(\tilde{X}_i)$ in $\{0.5, 1, 2, 3, 4\}$. To evaluate the output group ordering, we use *Kendall tau distance* [22] to the ground-truth (in range $[0, 1]$, the lower the better). For algorithms returning DAG/PDAG, its ordering is first extracted according to Definition 7. Figure 5 shows the results. Each column subplot indicates an NSR setting. The error bar is from 50 random generated instances. We can see that TIN-subsets and TIN-2steps are two best methods (ranks first on $68/80$ cases), which steadily stick near x-axis. They both find/check independence by solving equations $\mathrm{cov}(f(\mathbf{Z}), \mathbf{Y})\omega = \mathbf{0}$. TIN-ISA is slightly inferior (though still ranks 3rd), maybe due to the bias to independence (of $\omega$ vectors) by ISA. TIN-rank fluctuates and performs worst (among TINs), especially on the fully connected dense case. This might be explained by higher order cumulants' sensitivity to outliers, and unreliable numerical rank tests (we simply use SVD and thresholding).

Among the competitors, ICA-LiNGAM and SCORE are the strongest two, which remain relatively stable with NSR growing larger (while e.g., Direct-LiNGAM deteriorates rapidly). Interestingly, they both perform much better on fully connected DAGs than on chain structures, and ICA-LiNGAM even performs better on larger graphs than on smaller ones. We will investigate the reason. Consider CAMME-OICA, it relies heavily on mixing matrix parameter initialization, and thus performs generally weak. Weak results by PC, GES might be because of the unfair setting for them: they output CPDAG and in both cases (fully connected and chain) here, naturally no ordering (directions) can be determined. More experimental configuration and results discussions are in Appendix G.2.

### 6.2 Real-World Data

Sachs's [33] is a real dataset that measures the expression levels of proteins in human cells under various phospholipids. The ground-truth graph structure [33] contains 17 edges on 11 variables (cell types), of which the ordered group decomposition is {plc,pkc}→{pip2,pip3,pka,p38,jnk}→{raf}→{mek}→{erk,akt}. Result given by TIN-subsets achieves the best distance score $0.33$: {plc,pkc,p38}→{raf,mek,pip2,pka,jnk}→{pip3}→{erk,akt}. This score on PC, GES, ICA-LiNGAM are $0.49$, $0.69$, $0.8$ respectively. E.g., PC outputs {raf,pka}→{mek}→{plc,pip2,pip3,pkc,p38,jnk}→{erk,akt}. Experiments on another dataset, Teacher Burnout [4], also show TIN's good performance. See Appendix H.5.

## 7 Conclusion and Discussions

In this work we define the **T**ransformed **I**ndependent **N**oise (TIN) condition based on LiNGAM causal model, which finds and checks for independence between a specific linear transformation of some variables and others. We provide graphical criteria of TIN, which might further improve identifiability of the latent-variable problem. Specifically on causal discovery under measurement error, we exploit TIN to achieve identifiability of ordered group decomposition.

We summarize the future work as three fold: **1)** For the measurement error model, in addition to the special type (one-over-others), TIN over general $\mathbf{Z}$, $\mathbf{Y}$ pairs can further improve identifiability. See Appendix F.2; **2)** TIN now only considers the dimension of $\Omega_{\mathbf{Z};\mathbf{Y}}$, while parameters might also help to recover edges weights. See Appendix F.3; and **3)** Reliable estimation of $\Omega_{\mathbf{Z};\mathbf{Y}}$ can be formulated as an orthogonal research problem. We believe there exists more solutions. See Appendix F.4.

### Acknowledgments and Disclosure of Funding

The authors would like to thank Georges Darmois, Viktor Skitovich, Bernt Lindström, Ira Gessel, Gérard Viennot, and Seth Sullivant for initializing the beautiful theorems that this work is built upon. Thank Feng Xie, Ruichu Cai, Biwei Huang, Clark Glymour, and Zhifeng Hao for the GIN condition. Thanks to the anonymous reviewers and Joseph Ramsey, Zeyu Tang, Yujia Zheng, Ignavier Ng, Justin Ding, Mengyao Lu, Haoqin Tu, Qiyu Wu, Muyang Li, Jinkun Cao, and Jinhao Zhu for helpful feedback, proofreading, and discussions. The work was partially supported by the NSF under Project Number A221500S001, by the NSF-Convergence Accelerator Track-D award #2134901, by NIH-NHLB1 9R01HL159805-05A1, by a grant from Apple Inc., and by a grant from KDDI Research Inc.

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
