# A Proofs of Main Results

## A.1 Proof of Proposition 1

**Proposition 1** (*rare* d-separation). *Suppose variables follow random measurement error model defined in Equation* (1). *For disjoint sets of observed variables* $\mathbf{Z}, \mathbf{Y}, \mathbf{S}$ *and their respective latent ones* $\tilde{\mathbf{Z}}, \tilde{\mathbf{Y}}, \tilde{\mathbf{S}}$, *d-separation* $\mathbf{Z} \perp\!\!\!\perp_d \mathbf{Y} | \mathbf{S}$ *holds, only when marginally* $\tilde{\mathbf{Z}} \perp\!\!\!\perp_d \tilde{\mathbf{Y}}$, *and* $\tilde{\mathbf{Z}} \perp\!\!\!\perp_d \tilde{\mathbf{Y}} | \tilde{\mathbf{S}}$ *hold.*

*Proof of Proposition 1.* The whole graph we consider is the graph $\tilde{G}$ among latent variables $\tilde{\mathbf{X}}$ and measurement edges $\tilde{X}_i \rightarrow X_i$. Consider observed variables $X_i, X_j$ and subset $\mathbf{S}$. Denote by "des" the descendants of some vertices on graph. By definition of d-separation, if $X_i \perp\!\!\!\perp_d X_j | \mathbf{S}$, then for every undirected path $p$ linking $X_i$ and $X_j$ (if there is any), $p$ is blocked by $\mathbf{S}$, i.e., *either 1)* there exists a collider $W$, s.t. $W \notin \mathbf{Z}$ and $\mathrm{des}(W) \cap \mathbf{S} = \varnothing$, *or 2)* there exists a non-collider $W$ s.t. $W \in \mathbf{S}$. Since observed variables are all leaf nodes of their respective latent nodes, for every undirected path $p$ linking $X_i$ and $X_j$, $p$ must be in form of $X_i - \tilde{X}_i - \cdots - \tilde{X}_j - X_j$. Hence, $W$ must be in latent nodes, and only *case 1)* is possible, which means that there exists a collider on every $p$ linking $X_i$ and $X_j$, and thus $X_i$ and $X_j$ is also d-separated by $\varnothing$ (conclusion 1). Specifically on *case 1)*, $W \notin \mathbf{S}$ is obvious (since $W \in \tilde{\mathbf{X}}$ and $\mathbf{S} \in \mathbf{X}$). And, by $\mathrm{des}(W) \cap \mathbf{S} = \varnothing$, we have $W \notin \tilde{\mathbf{S}}$ and $\mathrm{des}(W) \cap \tilde{\mathbf{S}} = \varnothing$ (easy to show since $\mathbf{S} \subseteq \mathrm{des}(\tilde{\mathbf{S}})$), and thus among latent nodes, there is $\tilde{X}_i \perp\!\!\!\perp_d \tilde{X}_j | \tilde{\mathbf{S}}$ (conclusion 2). Combining conclusion 1 and 2, let $\mathbf{S} := \varnothing$, we further have marginally $\tilde{X}_i \perp\!\!\!\perp_d \tilde{X}_j$ holds. $\qquad\square$

**Remark 1.** *Roughly speaking, the d-separation patterns among* $\tilde{\mathbf{X}}$ *usually do not hold among* $\mathbf{X}$ *(except for rare marginal ones), since the observed variables are all leaf nodes, which are not causes of any other (though the latent variables they intend to measure might be). By a similar proof procedure, we shall have a full version of Proposition 1: for observed variables* $X_i, X_j$ *and subset* $\mathbf{S}$,

1. *If* $X_i \perp\!\!\!\perp_d X_j | \mathbf{S}$, *then among latent variables, marginally* $\tilde{X}_i \perp\!\!\!\perp_d \tilde{X}_j$, *and* $\tilde{X}_i \perp\!\!\!\perp_d \tilde{X}_j | \tilde{\mathbf{S}}$ *holds.*
2. *If* $X_i \not\perp\!\!\!\perp_d X_j | \mathbf{S}$, *which means that there exists a path $p$ from $X_i$ to $X_j$ unblocked by $\mathbf{Z}$. And,*
   a) *If on $p$ there is a non-collider* $W \in \tilde{\mathbf{S}}$, *then* $\tilde{X}_i \perp\!\!\!\perp_d \tilde{X}_j | \tilde{\mathbf{S}}$;
   b) *Otherwise (on $p$ there is no non-collider* $W \in \tilde{\mathbf{S}}$), $\tilde{X}_i \not\perp\!\!\!\perp_d \tilde{X}_j | \tilde{\mathbf{S}}$.

## A.2 Proof of Theorem 1

**Theorem 1** (Characterization of $\Omega_{\mathbf{Z};\mathbf{Y}}$). *For two variables subsets* $\mathbf{Z}$ *and* $\mathbf{Y}$, $\Omega_{\mathbf{Z};\mathbf{Y}}$ *satisfies:*

$$\Omega_{\mathbf{Z};\mathbf{Y}} = \mathrm{null}(\mathbf{B}^{\mathsf{T}}_{\mathbf{Y},\mathrm{nzcol}(\mathbf{B}_{\mathbf{Z},:})}). \tag{4}$$

*where* $\mathrm{null}(\cdot)$ *denotes nullspace.* $\mathbf{B}_{\mathbf{Y},\mathrm{nzcol}(\mathbf{B}_{\mathbf{Z},:})}$ *denotes the submatrix of mixing matrix* $\mathbf{B}$, *with rows indexed by* $\mathbf{Y}$ *and columns indexed by* $\mathrm{nzcol}(\mathbf{B}_{\mathbf{Z},:})$. $\mathrm{nzcol}(\mathbf{B}_{\mathbf{Z},:})$ *denotes the column indices where the submatrix* $\mathbf{B}_{\mathbf{Z},:}$ *has non-zero entries.* $\mathrm{nzcol}(\mathbf{B}_{\mathbf{Z},:})$ *actually corresponds to the exogenous noises that constitute* $\mathbf{Z}$. *Particularly, if assuming "if* $i \rightsquigarrow j$ *then* $\mathbf{B}_{j,i} \neq 0$", *then,* $\mathrm{nzcol}(\mathbf{B}_{\mathbf{Z},:}) = \mathrm{Anc}(\mathbf{Z})$.

*Proof of Theorem 1.* We write variables in terms of linear combination of exogenous noises, $\mathbf{X} = \mathbf{B}\mathbf{E}$. For variables set $\mathbf{Z} = \mathbf{B}_{\mathbf{Z},:}\mathbf{E} = \mathbf{B}_{\mathbf{Z},\mathrm{nzcol}(\mathbf{B}_{\mathbf{Z},:})}\mathbf{E}_{\mathrm{nzcol}(\mathbf{B}_{\mathbf{Z},:})}$, where $\mathrm{nzcol}(\mathbf{B}_{\mathbf{Z},:})$ denotes the column indices where the submatrix $\mathbf{B}_{\mathbf{Z},:}$ has non-zero entries, i.e., $\mathbf{Z}$ contains and only contains noise terms $\mathbf{E}_{\mathrm{nzcol}(\mathbf{B}_{\mathbf{Z},:})}$. For a vector $\omega \in \mathbb{R}^{|\mathbf{Y}|}$, $\omega^{\mathsf{T}}\mathbf{Y} = \omega^{\mathsf{T}}\mathbf{B}_{\mathbf{Y},:}\mathbf{E} = \omega^{\mathsf{T}}\mathbf{B}_{\mathbf{Y},\mathrm{nzcol}(\mathbf{B}_{\mathbf{Z},:})}\mathbf{E}_{\mathrm{nzcol}(\mathbf{B}_{\mathbf{Z},:})} + \omega^{\mathsf{T}}\mathbf{B}_{\mathbf{Y},\sim\mathrm{nzcol}(\mathbf{B}_{\mathbf{Z},:})}\mathbf{E}_{\sim\mathrm{nzcol}(\mathbf{B}_{\mathbf{Z},:})}$ ("$\sim$" denotes complement set). By the Darmois–Skitovich theorem [21], $\omega^{\mathsf{T}}\mathbf{Y} \perp\!\!\!\perp \mathbf{Z}$ if and only if $\omega^{\mathsf{T}}\mathbf{Y}$ shares no common non-Gaussian noise terms with $\mathbf{Z}$, i.e., $\omega^{\mathsf{T}}\mathbf{B}_{\mathbf{Y},\mathrm{nzcol}(\mathbf{B}_{\mathbf{Z},:})} = 0$. Moreover, if assuming "if $i \rightsquigarrow j$ then $\mathbf{B}_{j,i} \neq 0$" (a weaker faithfulness assumption, see Appendix F.1), then, $\mathrm{nzcol}(\mathbf{B}_{\mathbf{Z},:}) = \mathrm{Anc}(\mathbf{Z})$, i.e., variables set $\mathbf{Z}$ contains and only contains exogenous noises w.r.t. its ancestors set. $\qquad\square$

## A.3 Proof of Theorem 2

**Theorem 2** (Graphical criteria of $\Omega_{\mathbf{Z};\mathbf{Y}}$). *Let* $\mathbf{Z}, \mathbf{Y}$ *be two subsets of variables (vertices), we have:*

$$|\mathbf{Y}| - \dim(\Omega_{\mathbf{Z};\mathbf{Y}}) = \min\{|\mathbf{S}| \mid \mathbf{S} \text{ is a vertex cut from } \mathrm{Anc}(\mathbf{Z}) \text{ to } \mathbf{Y}\}. \tag{7}$$

*where* $\dim(\Omega_{\mathbf{Z};\mathbf{Y}})$ *denotes the dimension of the subspace* $\Omega_{\mathbf{Z};\mathbf{Y}}$, *i.e., the degree of freedom of* $\omega$.

To prove Theorem 2 we mainly use the Lindström-Gessel-Viennot theorem [25, 13] in algebraic combinatorics, which gives a combinatorial interpretation of the determinants of certain matrices:

**Theorem 6** (Lindström-Gessel-Viennot theorem [25, 13]). *Let $G$ be a directed acyclic graph with vertex set $[n]$. Each directed edge $i \to j$ is assigned with a weight $e(i, j)$. For each directed path $P$ from vertices $i$ to $j$, let $\mathrm{wt}(P) := \Pi_{m \to l \in P} e(m, l)$, the product of the weights of the edges of the path. For any two vertices $i, j$, denote $\mathcal{P}(i, j)$ the set of all directed paths from $i$ to $j$. Write an $n \times n$ matrix $M$, with entries defined as $M_{i,j} = \sum_{P \in \mathcal{P}(i,j)} \mathrm{wt}(P)$, the sum of path weights over all paths from $i$ to $j$. For two subsets $S, T \subseteq [n]$ with $|S| = |T| = k$ (letters "S" means source and "T" means sink), we have:*

$$\det(M_{S,T}) = \sum_{\mathbf{P}=(P_1,\ldots,P_k) \colon S \to T} \mathrm{sign}(\sigma(\mathbf{P})) \prod_{i=1}^{k} \mathrm{wt}(P_i). \tag{A.1}$$

*where the sum is taken over all $k$-tuples $\mathbf{P} = (P_1, \ldots, P_k)$ of non-intersecting paths from $S$ to $T$, and $\sigma(\mathbf{P})$ is the sign of the corresponding permutation of elements in $\mathbf{P}$. "non-intersecting" means that for any two paths $P_i, P_j \in \mathbf{P}$ with $i \neq j$, $P_i$ and $P_j$ have no two vertices in common (not even endpoints). In particular, $\det(M_{S,T}) = 0$ if and only if there exists no such $k$-tuple non-intersecting paths, i.e., for every system of $k$ paths from $S$ to $T$, there exists two paths that share a vertex.*

Based on Theorem 6, we can readily give proof to Theorem 2. Note that in our setting where $\mathbf{X} = \mathbf{A}\mathbf{X} + \mathbf{E} = \mathbf{B}\mathbf{E}$ with $\mathbf{B} = (\mathbf{I} - \mathbf{A})^{-1}$, we know that $\mathbf{A}_{j,i}$ is the $e(i, j)$ above, and $\mathbf{B}$ is exactly $M^\intercal$, with $\mathbf{B}_{j,i} = \sum_{P \in \mathcal{P}(i,j)} \prod_{k \to l \in P} \mathbf{A}_{l,k}$, the total causal effect from $i$ to $j$.

*Proof of Theorem 2.* From Theorem 1 and Assumption 1 and the rank-nullity theorem, $|\mathbf{Y}| - \dim(\Omega_{\mathbf{Z};\mathbf{Y}})$ is equal to $\mathrm{rank}(\mathbf{B}_{\mathbf{Y},\mathrm{Anc}(\mathbf{Z})})$. By the max-flow min-cut theorem (vertex version, known as Menger's theorem) [9, 3, 26], the maximum amount of non-intersecting paths from source to sink is equal to the size of the minimum vertex cut from source to sink. Hence, if the minimum vertex cut from $\mathrm{Anc}(\mathbf{Z})$ to $\mathbf{Y}$ is of size $k$, then there exists a $k$-tuples of non-intersecting paths from some subset of $\mathrm{Anc}(\mathbf{Z})$ to some subset of $\mathbf{Y}$, and this is the largest possible non-intersecting paths system from $\mathrm{Anc}(\mathbf{Z})$ to $\mathbf{Y}$. By Theorem 6 and Assumption 1 (no parameter coupling to make coincidental low rank), this means that all $(k+1) \times (k+1)$ minors of $\mathbf{B}_{\mathbf{Y},\mathrm{Anc}(\mathbf{Z})}$ is zero and at least one $k \times k$ minor of $\mathbf{B}_{\mathbf{Y},\mathrm{Anc}(\mathbf{Z})}$ is non-zero. Hence $\mathrm{rank}(\mathbf{B}_{\mathbf{Y},\mathrm{Anc}(\mathbf{Z})}) = k$. $\qquad \square$

Interestingly, we find that our defined vertex cut has connection with trek-separation [47], i.e. "$\mathbf{S}$ is a vertex cut from $\mathrm{Anc}(\mathbf{Z})$ to $\mathbf{Y}$" is equivalent to "$(\varnothing, \mathbf{S})$ t-separates $(\mathbf{Z}, \mathbf{Y})$" (see Appendix C). Trek-separation theorem states that:

**Theorem 7** (Trek-separation for directed graphical models, Theorem 2.8 in [47]). *For two vertices sets $\mathbf{W}, \mathbf{Y}$, the variance-covariance matrix $\mathrm{cov}(\mathbf{W}, \mathbf{Y})$ has rank less than or or equal to $k$ for all covariance matrices consistent with the graph $G$ if and only if there exists subsets $\mathbf{S_W}, \mathbf{S_Y} \subseteq V(G)$ with $|\mathbf{S_W}| + |\mathbf{S_Y}| \leq k$ such that $(\mathbf{S_W}, \mathbf{S_Y})$ t-separates $(\mathbf{W}, \mathbf{Y})$. Consequently,*

$$\mathrm{rank}(\mathrm{cov}(\mathbf{W}, \mathbf{Y})) \leq \min\{|\mathbf{S_W}| + |\mathbf{S_Y}| \mid (\mathbf{S_W}, \mathbf{S_Y}) \text{ t-separates } (\mathbf{W}, \mathbf{Y})\} \tag{A.2}$$

*and equality holds for generic covariance matrices (i.e., no coincidental low rank in variance-covariance matrix) consistent with $G$.*

We now show that Theorem 2 can also be proved by trek-separation theorem:

*Proof of Theorem 2 (another version).* From Theorem 1 and Assumption 1 and the rank-nullity theorem, $|\mathbf{Y}| - \dim(\Omega_{\mathbf{Z};\mathbf{Y}})$ is equal to $\mathrm{rank}(\mathbf{B}_{\mathbf{Y},\mathrm{Anc}(\mathbf{Z})})$. Then, what is $\mathrm{rank}(\mathbf{B}_{\mathbf{Y},\mathrm{Anc}(\mathbf{Z})})$? Let us consider two variables sets $\mathrm{Anc}(\mathbf{Z})$ and $\mathbf{Y}$ and their respective variance-covariance matrix. We write $\mathrm{Anc}(\mathbf{Z})$ as mixed noise components $\mathrm{Anc}(\mathbf{Z}) = \mathbf{B}_{\mathrm{Anc}(\mathbf{Z}),\mathrm{Anc}(\mathbf{Z})}\mathbf{E}_{\mathrm{Anc}(\mathbf{Z})}$, where $\mathbf{B}_{\mathrm{Anc}(\mathbf{Z}),\mathrm{Anc}(\mathbf{Z})}$ is a square matrix which can be simultaneously permuted to lower triangular with diagonals one, and thus is full rank. Then write $\mathbf{Y} = \mathbf{B}_{\mathbf{Y},:}\mathbf{E} = \mathbf{B}_{\mathbf{Y},\mathrm{Anc}(\mathbf{Z})}\mathbf{E}_{\mathrm{Anc}(\mathbf{Z})} + \mathbf{B}_{\mathbf{Y},\sim\mathrm{Anc}(\mathbf{Z})}\mathbf{E}_{\sim\mathrm{Anc}(\mathbf{Z})}$, where the second part are $\mathbf{Y}$'s noise components that is not shared in $\mathrm{Anc}(\mathbf{Z})$, so is independent to $\mathrm{Anc}(\mathbf{Z})$ and can be dropped in calculating covariance. From above, $\mathrm{cov}(\mathrm{Anc}(\mathbf{Z}), \mathbf{Y}) = \mathbf{B}_{\mathrm{Anc}(\mathbf{Z}),\mathrm{Anc}(\mathbf{Z})}\Phi(\mathbf{E}_{\mathrm{Anc}(\mathbf{Z})})\mathbf{B}_{\mathbf{Y},\mathrm{Anc}(\mathbf{Z})}^{\intercal}$, where $\Phi(\mathbf{E}_{\mathrm{Anc}(\mathbf{Z})})$ is a diagonal matrix with diagonal entries being variance of exogenous noise terms in $\mathbf{E}_{\mathrm{Anc}(\mathbf{Z})}$. Since both $\mathbf{B}_{\mathrm{Anc}(\mathbf{Z}),\mathrm{Anc}(\mathbf{Z})}$ and $\Phi(\mathbf{E}_{\mathrm{Anc}(\mathbf{Z})})$ are full rank square matrices, $\mathrm{rank}(\mathbf{B}_{\mathbf{Y},\mathrm{Anc}(\mathbf{Z})})$ is equal to $\mathrm{rank}(\mathrm{cov}(\mathrm{Anc}(\mathbf{Z}), \mathbf{Y}))$.

According to Theorem 7 and Assumption 1, $\text{rank}(\text{cov}(\mathbf{W}, \mathbf{Y}))$ is equal to $\min\{|\mathbf{S_W}| + |\mathbf{S_Y}|| (\mathbf{S_W}, \mathbf{S_Y})$ t-separates $(\mathbf{W}, \mathbf{Y})\}$. Further we obtain a lemma: if $\mathbf{W} = \text{Anc}(\mathbf{W})$, i.e., ancestors are self-contained in $\mathbf{W}$, then $\text{rank}(\text{cov}(\mathbf{W}, \mathbf{Y})) = \min\{|\mathbf{S}| \mid (\varnothing, \mathbf{S})$ t-separates $(\mathbf{W}, \mathbf{Y})\}$. This can be proved by that, for self-contained $\mathbf{W}$, for any $(\mathbf{S_W}, \mathbf{S_Y})$ that t-separates $(\mathbf{W}, \mathbf{Y})$, $(\varnothing, \mathbf{S_W} \cup \mathbf{S_Y})$ also t-separates $(\mathbf{W}, \mathbf{Y})$. Another lemma is that, $(\varnothing, \mathbf{S})$ t-separates $(\text{Anc}(\mathbf{Z}), \mathbf{Y})$ if and only if $(\varnothing, \mathbf{S})$ t-separates $(\mathbf{Z}, \mathbf{Y})$ (see Appendix C).

With lemmas above, we immediately have that $\text{rank}(\mathbf{B}_{\mathbf{Y}, \text{Anc}(\mathbf{Z})})$ is equal to the size of the minimum vertices set $\mathbf{S}$ s.t. $(\varnothing, \mathbf{S})$ t-separates $(\mathbf{Z}, \mathbf{Y})$, i.e., the size of the minimum vertex cut from $\text{Anc}(\mathbf{Z})$ to $\mathbf{Y}$. $\qquad\square$

### A.4 Proof of Theorem 3

**Theorem 3** (Equivalence of TIN over latent and observed variables). *For two **disjoint** observed variables subsets $\mathbf{Z}, \mathbf{Y}$, and their respective underlying latent variables subsets $\tilde{\mathbf{Z}}, \tilde{\mathbf{Y}}$,*

$$\text{TIN}(\mathbf{Z}, \mathbf{Y}) = \text{TIN}(\tilde{\mathbf{Z}}, \tilde{\mathbf{Y}}). \tag{10}$$

*Proof of Theorem 3.* Theorem 3 can either be proved by the graphical criteria (where observed variables are all leaf nodes), or by mathematically showing how rank of submatrices of $\mathbf{B}$ preserves among latent and observed variables. Consider the latent $\tilde{\mathbf{X}}$ in LiNGAM:

$$\tilde{\mathbf{X}} = \tilde{\mathbf{A}}\tilde{\mathbf{X}} + \tilde{\mathbf{E}}; \quad \tilde{\mathbf{X}} = \tilde{\mathbf{B}}\tilde{\mathbf{E}}; \quad \text{variance-covariance matrix } \tilde{\mathbf{\Sigma}} = \tilde{\mathbf{B}}\Phi_{\tilde{\mathbf{E}}}\tilde{\mathbf{B}}^\mathsf{T} \tag{A.3}$$

when $X_i = \tilde{X}_i + E_i$, write latent and observed variables together, we have:

$$\left[\frac{\tilde{\mathbf{X}}}{\mathbf{X}}\right] = \mathbf{A}' \cdot \left[\frac{\tilde{\mathbf{X}}}{\mathbf{X}}\right] + \left[\frac{\tilde{\mathbf{E}}}{\mathbf{E}}\right], \quad \text{where } \mathbf{A}' = \left[\begin{array}{c|c}\tilde{\mathbf{A}} & \mathbf{0} \\ \hline \mathbf{I} & \mathbf{0}\end{array}\right],$$

$$\left[\frac{\tilde{\mathbf{X}}}{\mathbf{X}}\right] = \mathbf{B}' \cdot \left[\frac{\tilde{\mathbf{E}}}{\mathbf{E}}\right], \quad \text{where } \mathbf{B}' = \left[\begin{array}{c|c}\tilde{\mathbf{B}} & \mathbf{0} \\ \hline \tilde{\mathbf{B}} & \mathbf{I}\end{array}\right], \tag{A.4}$$

$$\mathbf{\Sigma}' = \text{cov}\left(\left[\frac{\tilde{\mathbf{X}}}{\mathbf{X}}\right]\right) = \left[\begin{array}{c|c}\tilde{\mathbf{\Sigma}} & \tilde{\mathbf{\Sigma}} \\ \hline \tilde{\mathbf{\Sigma}} & \tilde{\mathbf{\Sigma}} + \Phi_{\mathbf{E}}\end{array}\right], \quad \text{where } \Phi_{\mathbf{E}} = \text{diag}(\text{var}(\mathbf{E})).$$

More generally, when observations are measured with $X_i = c_i\tilde{X}_i + E_i$, let $\mathbf{C} = \text{diag}([c_1, \cdots, c_n]^\mathsf{T})$:

$$\mathbf{A}' = \left[\begin{array}{c|c}\tilde{\mathbf{A}} & \mathbf{0} \\ \hline \mathbf{C} & \mathbf{0}\end{array}\right], \quad \mathbf{B}' = \left[\begin{array}{c|c}\tilde{\mathbf{B}} & \mathbf{0} \\ \hline \mathbf{C}\tilde{\mathbf{B}} & \mathbf{I}\end{array}\right], \quad \mathbf{\Sigma}' = \left[\begin{array}{c|c}\tilde{\mathbf{\Sigma}} & \tilde{\mathbf{\Sigma}}\mathbf{C}^\mathsf{T} \\ \hline \mathbf{C}\tilde{\mathbf{\Sigma}} & \mathbf{C}\tilde{\mathbf{\Sigma}}\mathbf{C}^\mathsf{T} + \Phi_{\mathbf{E}}\end{array}\right] \tag{A.5}$$

For two disjoint sets $\mathbf{Z}, \mathbf{Y}$, consider $\mathbf{B}_{\mathbf{Y}, \text{Anc}(\mathbf{Z})}$, a submatrix in $[\mathbf{C}\tilde{\mathbf{B}} \mid \mathbf{I}]$. $\text{Anc}(\mathbf{Z}) = \text{Anc}(\tilde{\mathbf{Z}}) \cup \mathbf{Z}$, where for the second $\mathbf{Z}$ parts, its indexed columns in $\mathbf{B}_{\mathbf{Y},:}$ must be all zero (since $\mathbf{Y}$ and $\mathbf{Z}$ are disjoint indices in $\mathbf{I}$), and thus can be dropped. For the first $\text{Anc}(\tilde{\mathbf{Z}})$ part, $\mathbf{B}_{\mathbf{Y}, \text{Anc}(\tilde{\mathbf{z}})}$ is $\mathbf{B}_{\tilde{\mathbf{Y}}, \text{Anc}(\tilde{\mathbf{z}})}$ with rows scaled by $\mathbf{C}$, and thus the rank holds. Consequently, for two disjoint vertices sets $\mathbf{Z}, \mathbf{Y}$, $\text{TIN}(\mathbf{Z}, \mathbf{Y}) = \text{TIN}(\tilde{\mathbf{Z}}, \tilde{\mathbf{Y}})$. $\qquad\square$

### A.5 Proofs of Other Results

For other lemmas and theorems in this paper: GIN, IN as special cases of TIN and Lemma 1 follows directly from the graphical criteria in Theorem 2. Theorem 4 can be proved in a similar way as the proof to Theorem 2, where the subsets and subdeterminants are considered. For ranks stopped increasing in Theorem 5, please refer to Appendix E.1.

## B Using GIN Condition-Based Algorithm Under 2-Measurements Model

As is illustrated in §2, when each latent variable $\tilde{X}_i$ has two pure measurements $X_{i_1}, X_{i_2}$ (by "pure" it means that each of $X_{i_1}, X_{i_2}$ has only one latent parent $\tilde{X}_i$ and no observed parents), graph structure $\tilde{G}$ over latent variables is fully identifiable by GIN (a simpler case). This is already a breakthrough comparing to existed methods [42, 46, 23], which only identify a partial graph.

Here is an illustrating example: consider a simple 2-variables example, $\tilde{X} \to \tilde{Y}$, with their respective measurements $X_1, X_2$ and $Y_1, Y_2$. One may check the entailed vanishing correlations: $\rho_{X_1,Y_1}\rho_{X_2,Y_2} = \rho_{X_1,Y_2}\rho_{X_2,Y_1}$, $\rho_{X_1,X_2}\rho_{Y_1,Y_2} \neq \rho_{X_1,Y_1}\rho_{X_2,Y_2}$, and $\rho_{X_1,X_2}\rho_{Y_1,Y_2} \neq \rho_{X_1,Y_2}\rho_{X_2,Y_1}$, where $\rho$ denotes correlation coefficient. These (in)equations exhibit no asymmetry between $\tilde{X}$ and $\tilde{Y}$. Indeed, for the inverse direction $\tilde{X} \leftarrow \tilde{Y}$, all the Tetrad constraints among $X_1, X_2, Y_1, Y_2$ hold the same. Therefore, the direction between $\tilde{X}$ and $\tilde{Y}$ is unidentifiable.

However, the GIN condition can identify an asymmetry: $\text{GIN}(X_1, Y_{1,2})$ holds, while $\text{GIN}(Y_1, X_{1,2})$ is violated, and thus the direction $\tilde{X} \to \tilde{Y}$ is identified. One can see this from the definition of GIN (Definition 2).

Below we give the general algorithm of using GIN to fully identify $\tilde{G}$. Note that here by "GIN condition", it is actually a bit different from the original paper [50]: it takes into account one more thing than the original definition: the degeneration of $\omega$ (see Appendix D for details). Here is the procedure:

Given $2n$ measured variables (where $n$ is the number of vertices in $\tilde{G}$), let two variables be $\mathbf{Y}$ and the rest $2n-2$ variables be $\mathbf{Z}$, $\text{GIN}(\mathbf{Z}, \mathbf{Y})$ if and only if these two variables are the two measurements of a same latent variable. Following this, the $2n$ measured variables can first be pairwise clustered, and labeled as $\{X_{i_1}, X_{i_2}\}_{i=1,\cdots,n}$. One may also obtain this pairwise labeling by prior knowledge (e.g., in survey questions design, one already knows which two questions indicate a same latent factor).

Then, find the graph structure $\tilde{G}$ over $n$ latent variables:

---

**Algorithm 1** Identifying graph structure of $\tilde{G}$ in 2-measurements case

---

**Input:** Labeled $2n$ measurements $\mathbf{X} = \{X_{i_1}, X_{i_2}\}_{i=1,\cdots,n}$ and corresponding data samples
**Output:** Graph structure of $\tilde{G}$
1: Initialize ordered list $K := \varnothing$, remaining indexes $U := \{1, \cdots, n\}$, parents dictionary $P := \{\}$;
2: Denote a half of measurements $\mathbf{X_1} = \{X_{i_1}\}_{i=1,\cdots,n}$;
3: **while** there are more than one remaining index in $U$ **do**
4:     Find one $j \in U$ with $\text{GIN}(\mathbf{Z}, \mathbf{X_1})$, where $\mathbf{Z} := \{X_{i_2} | i \in K \cup \{j\}\}$; //pick from another half
5:     Append $j$ to the end of $K$. Let $U := U \setminus \{j\}$;
6: **end while**
7: Append the only one remaining index in $U$ to the end of $K$;
8: **for** vertex index $j$ in causal ordering list $K$ **do**
9:     Let $A := \{i | i \text{ earlier than } j \text{ in } K\}$, $\mathbf{Z} := \{X_{i_1} | i \in A\}$, $\mathbf{Y} := \{X_{i_2} | i \in A \cup \{j\}\}$;
10:     $\text{GIN}(\mathbf{Z}, \mathbf{Y})$ must hold, with solution $\omega$. Let $P[j] := \{i \in A | \omega \text{ on } X_{i_2} \text{ is non-degenerated}\}$;
11: **end for**
12: **Return:** Graph structure $\tilde{G}$ where each vertex $j$ has direct parents $P[j]$

---

Algorithm 1 follows a similar procedure as Direct-LiNGAM [40]: Lines 3-5 sorts the vertices by causal ordering, where there is no edge from later ones to earlier ones. Then according to degeneration graphical criteria in Appendix D, Line 10 identifies the direct parents set of each vertex from its causally earlier vertices set.

Further consider the coefficients. Denote the linear coefficients of latent variable $\tilde{X}_i$ to two measurements $X_{i_{1,2}}$ as $\alpha_{i_{1,2}}$ respectively. The ratio $\alpha_{i_1}/\alpha_{i_2}$ is accessible when testing GIN for pairwise clusters. Then, in Line 10, to identify parents set for each vertex $j$, we find from $A$, the vertices earlier than $j$ in ordered list $K$. We write a scaling vector $\mathbf{s} := \{\alpha_{j_2}/\alpha_{i_2} | i \in A\}$, and denote the coefficients vector from $A$ to $j$ as $\mathbf{c}$ (zero if no direct edge). Note that here $\omega$ must only have a free degree of one (according to Theorem 2 and Appendix D.1, critical vertex cut is $A$). So set the value of $\omega$ on $X_{j_2}$ as $-1$, then the value of $\omega$ on other $X_{i_2}$s is exactly the point-wise multiplication of $\mathbf{s}$ and $\mathbf{c}$. If we further assume that linear coefficients from latent variables to measurements are all same (e.g., one, $X_{i_{1,2}} = \tilde{X}_i + E_{i_{1,2}}$), or equivalently, the measurement errors are uni-variance, then the coefficients among $\tilde{G}$ is also fully identifiable.

## C  Elaboration on Vertex Cut and Graph Definitions

We first give more detailed definitions to the concepts in §3.

**Definition 9** (Directed paths). A directed path $P = (i_0, i_1, \cdots, i_k)$ in $G$ is a sequence of vertices of $G$ where there is a directed edge from $i_j$ to $i_{j+1}$ for any $0 \leq j \leq k - 1$. We use notation $i \rightsquigarrow j$ to show that there exists a directed path from vertex $i$ to $j$.

**Remark 2.** *Note that a single vertex is also a directed path, i.e., $i \rightsquigarrow i$ holds true.*

**Definition 10** (Directed paths without passing through $\mathbf{S}$). Let $\mathbf{S}$ be a subset of vertices. We use notation $i \overset{\mathbf{S}}{\not\rightsquigarrow} j$ to show that there exists a directed path from vertex $i$ to $j$ without passing through $\mathbf{S}$, i.e., there exists a directed path $P = (i, m_0, \cdots, m_k, j)$ in $G$ s.t. $i, j \notin \mathbf{S}$ and $m_l \notin \mathbf{S}$ for any $0 \leq l \leq k$.

**Remark 3.** *Note that when $\mathbf{S}$ is empty, $i \rightsquigarrow j$ is equivalent to $i \overset{\mathbf{S}}{\not\rightsquigarrow} j$.*

**Definition 11** (Ancestors). Let $\mathbf{W}$ be a subset of vertices. Ancestors $\mathrm{Anc}(\mathbf{W}) \coloneqq \{j | \exists i \in \mathbf{W}, j \rightsquigarrow i\}$.

**Remark 4.** *Note that $\mathbf{W} \subseteq \mathrm{Anc}(\mathbf{W})$. Under faithfulness assumption (no parameter coupling), $\mathrm{Anc}(\mathbf{W})$ means all noise components that $\mathbf{W}$ carries, i.e., writing the corresponding variables set $\{X_i | i \in \mathbf{W}\}$ as linear combination of noises, it contains and only contains exogenous noises from $\{E_i | i \in \mathrm{Anc}(\mathbf{W})\}$.*

**Definition 12** (Ancestors outside $\mathbf{S}$). Let $\mathbf{W}, \mathbf{S}$ be two subsets of vertices. We denote ancestors of $\mathbf{W}$ that has directed paths into $\mathbf{W}$ without passing through $\mathbf{S}$ as $\mathrm{Anc}_{\mathrm{out}(\mathbf{S})}(\mathbf{W}) \coloneqq \{j | \exists i \in \mathbf{W}, j \overset{\mathbf{S}}{\not\rightsquigarrow} i\}$.

**Remark 5.** *According to definitions above,*

1. $\mathrm{Anc}_{\mathrm{out}(\varnothing)}(\mathbf{W}) = Anc(\mathbf{W})$. $\mathrm{Anc}_{\mathrm{out}(\mathbf{W})}(\mathbf{W}) = \varnothing$.
2. $\mathbf{S} \cap \mathrm{Anc}_{\mathrm{out}(\mathbf{S})}(\mathbf{W}) = \varnothing$. $\mathbf{W} \backslash \mathbf{S} \subseteq \mathrm{Anc}_{\mathrm{out}(\mathbf{S})}(\mathbf{W})$.
3. *For overlapped $\mathbf{S}, \mathbf{W}$, $\mathrm{Anc}_{\mathrm{out}(\mathbf{S})}(\mathbf{W}) = \mathrm{Anc}_{\mathrm{out}(\mathbf{S})}(\mathbf{W} \backslash \mathbf{S})$.*
4. *Roughly speaking, $\mathrm{Anc}_{\mathrm{out}(\mathbf{S})}(\mathbf{W})$ means all noise components that can contribute to $\mathbf{W}$ without passing $\mathbf{S}$. With slight notation abuse, we can write variables $\mathbf{W}$ as $\mathbf{W} = A\mathbf{S} + \mathbf{E_W}$, where $A\mathbf{S}$ is a linear transformation to $\mathbf{S}$, and $\mathbf{E_W}$ is a linear transformation to exogenous noises set that contains and only contains $\{E_i | i \in \mathrm{Anc}_{\mathrm{out}(\mathbf{S})}(\mathbf{W})\}$.*

**Definition 13** (Existence of causal effect from $\mathbf{W}_1$ to $\mathbf{W}_2$). Let $\mathbf{W}_1, \mathbf{W}_2$ be two subsets of vertices. We say there exists causal effect from $\mathbf{W}_1$ to $\mathbf{W}_2$ if and only if there exists a directed path $i \rightsquigarrow j$ with $i \in \mathbf{W}_1$ and $j \in \mathbf{W}_2$.

**Remark 6.** *According to definitions above,*

1. *Note that if $\mathbf{W}_1$ and $\mathbf{W}_2$ are not disjoint, then there must exist causal effect from $\mathbf{W}_1$ to $\mathbf{W}_2$.*
2. *An equivalent definition is that, $\mathrm{Anc}(\mathbf{W}_2) \cap \mathbf{W}_1 \neq \varnothing$.*

**Definition 14** (Existence of causal effect from $\mathbf{W}_1$ to $\mathbf{W}_2$ without passing through $\mathbf{S}$). Let $\mathbf{W}_1, \mathbf{W}_2, \mathbf{S}$ be three subsets of vertices. We say there exists causal effect from $\mathbf{W}_1$ to $\mathbf{W}_2$ without passing through $\mathbf{S}$ if and only if there exists a directed no-passing path $i \overset{\mathbf{S}}{\not\rightsquigarrow} j$ with $i \in \mathbf{W}_1$ and $j \in \mathbf{W}_2$.

**Remark 7.** *According to definitions above,*

1. *An equivalent definition is that, $\mathrm{Anc}_{\mathrm{out}(\mathbf{S})}(\mathbf{W}_2) \cap \mathbf{W}_1 \neq \varnothing$.*
2. *There exists no causal effect from $\mathbf{S}$ to $\mathbf{W}_1$ without passing $\mathbf{S}$, i.e., $\mathbf{S} \cap \mathrm{Anc}_{\mathrm{out}(\mathbf{S})}(\mathbf{W}) = \varnothing$.*
3. *This definition shows whether $\mathbf{S}$ chokes **all** directed paths from $\mathbf{W}_1$ to $\mathbf{W}_2$.*
4. *By Definition 4, the following statements are equivalent: 1) there exists no causal effect from $\mathbf{W}_1$ to $\mathbf{W}_2$ without passing through $\mathbf{S}$; 2) $\mathbf{S}$ is a vertex cut from $\mathbf{W}_1$ to $\mathbf{W}_2$; 3) $\forall i \in \mathbf{W}_1, j \in \mathbf{W}_2, i \overset{\mathbf{S}}{\not\rightsquigarrow} j$ does not hold; 4) $\mathrm{Anc}_{\mathrm{out}(\mathbf{S})}(\mathbf{W}_2) \cap \mathbf{W}_1 = \varnothing$; 5) $\mathbf{S}$'s removal from $G$ ensures there is no directed paths from $\mathbf{W}_1 \backslash \mathbf{S}$ to $\mathbf{W}_2 \backslash \mathbf{S}$.*

Now we have complete our graphical definitions. Let us also review trek-separation [47].

**Definition 15** (Trek). A *trek* in $G$ from $i$ to $j$ is an ordered pair of directed paths $(P_1, P_2)$ where $P_1$ has sink $i$, $P_2$ has sink $j$, and both $P_1$ and $P_2$ have the same source $k$. Note that one or both of $P_1$ and $P_2$ may consist of a single vertex, e.g., $((i), (i))$ is a trek from vertex $i$ to $i$.

**Definition 16** (t-separation). Let $\mathbf{W}, \mathbf{Y}, \mathbf{S_W}, \mathbf{S_Y}$ be four subsets of $V(G)$ which need not be disjoint. We say that the pair $(\mathbf{S_W}, \mathbf{S_Y})$ *trek separates* (or *t-separates*) $\mathbf{W}$ from $\mathbf{Y}$ if for every trek $(P_1, P_2)$ from a vertex in $\mathbf{W}$ to a vertex in $\mathbf{Y}$, either $P_1$ contains a vertex in $\mathbf{S_W}$ or $P_2$ contains a vertex in $\mathbf{S_Y}$.

The above two definitions are directly from [47]. By the "ancestors" related definitions introduced above and in §3, we can immediately get an equivalent restatement of t-separation as:

**Theorem 8** (Restatement of t-separation). *Let* $\mathbf{W}, \mathbf{Y}, \mathbf{S_W}, \mathbf{S_Y}$ *be four subsets of* $V(G)$ *which need not be disjoint. The pair* $(\mathbf{S_W}, \mathbf{S_Y})$ *t-separates* $\mathbf{W}$ *from* $\mathbf{Y}$*, if and only if there exists no causal effect from* $\mathrm{Anc}_{\mathrm{out}(\mathbf{S_W})}(\mathbf{W})$ *to* $\mathbf{Y}$ *without passing passing* $\mathbf{S_Y}$ *(see Definition 14).*

Note that the above graph condition also has an equivalent restatement:

*$\cdots$ if and only if there exists no causal effect from* $\mathrm{Anc}_{\mathrm{out}(\mathbf{S_Y})}(\mathbf{Y})$ *to* $\mathbf{W}$ *without passing passing* $\mathbf{S_W}$.

Both mean that $\mathrm{Anc}_{\mathrm{out}(\mathbf{S_W})}(\mathbf{W}) \cap \mathrm{Anc}_{\mathrm{out}(\mathbf{S_Y})}(\mathbf{Y}) = \varnothing$, i.e., if some noise components can flow into $\mathbf{W}$ without passing $\mathbf{S_W}$, then it cannot also flow into $\mathbf{Y}$ without passing $\mathbf{S_Y}$, or vice versa.

**Remark 8.** *Further, by definitions above and the rank constraints (in trek-separation theorem Theorem 7), we have the followings:*

1. $\mathrm{rank}(\mathrm{cov}(\mathbf{W}, \mathbf{Y})) \geq |\mathbf{W} \cap \mathbf{Y}|$, *since we must have* $\mathbf{W} \cap \mathbf{Y} \subseteq \mathbf{S_W} \cup \mathbf{S_Y}$ *if* $(\mathbf{S_W}, \mathbf{S_Y})$ *t-separates* $\mathbf{W}$ *from* $\mathbf{Y}$*, otherwise some unblocked vertex is in* $\mathrm{Anc}_{\mathrm{out}(\mathbf{S_W})}(\mathbf{W}) \cap \mathrm{Anc}_{\mathrm{out}(\mathbf{S_Y})}(\mathbf{Y})$.
2. $\mathrm{rank}(\mathrm{cov}(\mathbf{W}, \mathbf{Y})) \leq \min(|\mathbf{W}|, |\mathbf{Y}|)$, *since* $(\mathbf{W}, \varnothing)$ *and* $(\varnothing, \mathbf{Y})$ *always t-separates* $\mathbf{W}$ *from* $\mathbf{Y}$*, i.e.,* $\mathrm{Anc}_{\mathrm{out}(\mathbf{W})}(\mathbf{W}) = \varnothing$ *or* $\mathrm{Anc}_{\mathrm{out}(\mathbf{Y})}(\mathbf{Y}) = \varnothing$.
3. $\mathrm{rank}(\mathrm{cov}(\mathbf{W}, \mathbf{Y})) \leq |\mathrm{Anc}(\mathbf{W}) \cap \mathrm{Anc}(\mathbf{Y})|$, *since* $(\mathrm{Anc}(\mathbf{W}) \cap \mathrm{Anc}(\mathbf{Y}), \varnothing)$ *and* $(\varnothing, \mathrm{Anc}(\mathbf{W}) \cap \mathrm{Anc}(\mathbf{Y}))$ *always t-separates* $\mathbf{W}$ *from* $\mathbf{Y}$.
4. *The pair* $(\mathbf{S_W}, \mathbf{S_Y})$ *t-separates* $\mathbf{W}$ *from* $\mathbf{Y}$*, if and only if the pair* $(\mathbf{S_Y}, \mathbf{S_W})$ *t-separates* $\mathbf{Y}$ *from* $\mathbf{W}$.

From above we have seen the interpretation of t-separation from the "ancestors" language set. Then combining Definition 4 and Theorem 8, we know that the following statements are equivalent: **1)** $\mathbf{S}$ is a vertex cut from $\mathrm{Anc}(\mathbf{Z})$ to $\mathbf{Y}$; **2)** $(\varnothing, \mathbf{S})$ t-separates $(\mathbf{Z}, \mathbf{Y})$; **3)** There exists no causal effect from $\mathrm{Anc}(\mathbf{Z})$ to $\mathbf{Y}$ without passing through $\mathbf{S}$; **4)** There exists no causal effect from $\mathrm{Anc}_{\mathrm{out}(\mathbf{S})}(\mathbf{Y})$ to $\mathbf{Z}$.

# D   More Properties of $\mathrm{TIN}$ Condition

## D.1   Critical Vertex Cut

From the above §3 and Appendix C graphical criteria, we know that $\mathrm{TIN}(\mathbf{Z}, \mathbf{Y})$ is equal to the size of the minimum vertex cut from $\mathrm{Anc}(\mathbf{Z})$ to $\mathbf{Y}$.

**Remark 9.** *Following Definition 4, we first elaborate more on vertex cut:*

1. *For any* $\mathbf{Z}, \mathbf{Y}$*, any superset of* $\mathbf{Y}$ *(including* $\mathbf{Y}$*) is a vertex cut from* $\mathrm{Anc}(\mathbf{Z})$ *to* $\mathbf{Y}$.
2. *For any* $\mathbf{Z}, \mathbf{Y}$*, any superset of* $\mathrm{Anc}(\mathbf{Z})$ *(including* $\mathrm{Anc}(\mathbf{Z})$*) is a vertex cut from* $\mathrm{Anc}(\mathbf{Z})$ *to* $\mathbf{Y}$.
3. *For any vertex cut from* $\mathrm{Anc}(\mathbf{Z})$ *to* $\mathbf{Y}$*,* $\mathrm{Anc}(\mathbf{Z}) \cap \mathbf{Y} \subseteq \mathbf{S}$ *(to choke single vertex paths).*
4. *Following point 3, for overlapped* $\mathbf{Z}, \mathbf{Y}$ *in testing* $\mathrm{TIN}$ *condition, any vertex cut* $\mathbf{S}$ *must contains (at least)* $\mathbf{Z} \cap \mathbf{Y}$ *(the observed/testable intersection) as its subset.*
5. *Following point 4, if* $\mathbf{Y} \subseteq \mathbf{Z}$*, then there exists no non-zero* $\omega$ *s.t.* $\omega^\mathsf{T} \mathbf{Y} \perp\!\!\!\perp \mathbf{Z}$.
6. *Note that though expressed as "*$\mathbf{S}$ *is a vertex cut from* $\mathrm{Anc}(\mathbf{Z})$ *to* $\mathbf{Y}$*", it never implicitly implies a causal ordering of* $\mathbf{Z} \to \mathbf{S} \to \mathbf{Y}$*. E.g., in graph* $D \leftarrow A \to C \leftarrow B$*, consider* $\mathrm{TIN}(\mathbf{Z} = \{A\}, \mathbf{Y} = \{B, C\}) = 1$ *where the minimum vertex cut is* $\mathbf{S} = \{C\}$*, not causally earlier than* $\mathbf{Y}$*;* $\mathrm{TIN}(\mathbf{Z} = \{B, C\}, \mathbf{Y} = \{A, D\}) = 1$ *with the minimum vertex cut* $\mathbf{S} = \{A\}$*, but* $\mathbf{Z}$ *is neither causally earlier than* $\mathbf{Y}$ *nor than* $\mathbf{S}$.
7. *Following point 6, roughly speaking,* $\mathrm{TIN}$ *tells size of the minimum vertex cut, but not exactly the causal ordering. For the existence of non-zero* $\omega$ *s.t.* $\omega^\mathsf{T} \mathbf{Y} \perp\!\!\!\perp \mathbf{Z}$*, there can be some vertices in* $\mathbf{Y}$ *that are in or causally earlier than* $\mathbf{Z}$*, i.e.* $\mathrm{Anc}(\mathbf{Z}) \cap \mathbf{Y} \neq \varnothing$ *- as long as there are not "too many" (less than the cardinality of possible* $\mathbf{S}$*).*

8. *Note that the minimum vertex cut may not be unique. E.g., 1) Consider example in point 6, both $\mathbf{S} = \{C\}$ and $\mathbf{S} = \{A\}$ are minimum vertex cuts in $\mathrm{TIN}(\mathbf{Z} = \{A\}, \mathbf{Y} = \{B, C\}) = 1$. 2) Consider a chain structure with $\mathrm{TIN}(\mathbf{Z} = \{X_1\}, \mathbf{Y} = \{X_2, \cdots, X_n\}) = 1$, both $\mathbf{S} = \{X_1\}$ and $\mathbf{S} = \{X_2\}$ are minimum vertex cuts.*

Following point 6 of Remark 9, since the minimum vertex cut from $\mathrm{Anc}(\mathbf{Z})$ to $\mathbf{Y}$ may not be unique in a $\mathrm{TIN}(\mathbf{Z}, \mathbf{Y})$, to better use the graphical criteria, now we define the *critical vertex cut*:

**Definition 17** (Critical vertex cut). Denote $\mathcal{S}(\mathbf{Z}, \mathbf{Y})$ the collection of all sets $\mathbf{S} \subseteq V(G)$ s.t. $\mathbf{S}$ is a minimum vertex cut from $\mathrm{Anc}(\mathbf{Z})$ to $\mathbf{Y}$ ("minimum" means that $|\mathbf{S}| = \mathrm{TIN}(\mathbf{Z}, \mathbf{Y})$). For a vertex cut $\mathbf{S} \in \mathcal{S}(\mathbf{Z}, \mathbf{Y})$, we say $\mathbf{S}$ is *critical* if and only if there exists no causal effect from all (other) minimum vertex cuts to $\mathbf{Y}$ without passing through $\mathbf{S}$, i.e. $\mathrm{Anc}_{\mathrm{out}(\mathbf{S})}(\mathbf{Y}) \cap \mathrm{Anc}(\bigcup \mathcal{S}(\mathbf{Z}, \mathbf{Y})) = \varnothing$.

**Remark 10.** *Roughly speaking, when there are multiple minimum vertex cuts, i.e., these multiple sets can all cut from $\mathrm{Anc}(\mathbf{Z})$ to $\mathbf{Y}$, then a critical one means a "last" one (furthest from $\mathbf{Z}$, deepest to $\mathbf{Y}$): it not only cuts $\mathrm{Anc}(\mathbf{Z})$ to $\mathbf{Y}$, but also cuts all other vertex cuts to $\mathbf{Y}$. E.g., consider examples in point 8 of Remark 9, 1) $\{C\}$ is critical while $\{A\}$ is not, because $\{C\}$ can cut $\{A\}$ to $\{B, C\}$, but $\{A\}$ cannot cut $\{C\}$ to $\{B, C\}$. 2) $\{X_2\}$ is critical while $\{X_1\}$ is not.*

**Theorem 9** (Uniqueness of critical gin-separation set). *For two vertices sets $\mathbf{Z}$ and $\mathbf{Y}$ and their respective $\mathrm{TIN}(\mathbf{Z}, \mathbf{Y})$, there exists one and only one corresponding critical vertex cut, denoted as $\mathbf{S}_{\mathbf{Z}, \mathbf{Y}}^*$.*

### D.2 Noise Components of Linear Transformation $\omega^\intercal \mathbf{Y}$

From above Appendix D.1 we defined the critical vertex cut $\mathbf{S}_{\mathbf{Z}, \mathbf{Y}}^*$ behind a $\mathrm{TIN}(\mathbf{Z}, \mathbf{Y})$, with special property on it. Now we analyze the linear transformation $\omega^\intercal \mathbf{Y}$:

**Theorem 10** (Noise components of linear transformation $\omega^\intercal \mathbf{Y}$). *For two vertices sets $\mathbf{Z}$ and $\mathbf{Y}$ and their respective $\mathrm{TIN}(\mathbf{Z}, \mathbf{Y})$, for generic choice of $\omega$ (i.e., no coincidental noise cancelling by $\omega$), the corresponding linear transformation $\omega^\intercal \mathbf{Y}$ contains and only contains exogenous noises introduced by vertices that has directed paths to $\mathbf{Y}$ without passing through the critical vertex cut $\mathbf{S}_{\mathbf{Z}, \mathbf{Y}}^*$, i.e., $\mathcal{E}(\omega^\intercal \mathbf{Y}) = \{E_i | i \in \mathrm{Anc}_{\mathrm{out}(\mathbf{S}_{\mathbf{Z}, \mathbf{Y}}^*)}(\mathbf{Y})\}$, where $\mathcal{E}(\cdot)$ denotes the exogenous noises components set that a variable $\cdot$ is constituted of, and $E_i$ is the exogenous noise from vertex $i$.*

**Remark 11.** *A vertex cut $\mathbf{S}$ from $\mathrm{Anc}(\mathbf{Z})$ to $\mathbf{Y}$ yields that all noise components that $\mathbf{Z}$ carries (i.e., $\mathrm{Anc}(\mathbf{Z})$) cannot flow into (causal affects / contribute to) $\mathbf{Y}$ without passing through $\mathbf{S}$, then $\mathbf{Y}$ can be written as $\mathbf{Y} = L\mathbf{S} + \mathbf{E}_{\mathbf{Y}}'$, where $L$ denotes a linear transformation, and $\mathbf{E}_{\mathbf{Y}}'$ denotes noise components that can contribute to $\mathbf{Y}$ without passing through $\mathbf{S}$ (i.e., $\mathrm{Anc}_{\mathrm{out}(\mathbf{S})}(\mathbf{Y})$) - so $\mathbf{E}_{\mathbf{Y}}' \perp\!\!\!\perp \mathbf{Z}$, but not necessarily $\mathbf{E}_{\mathbf{Y}}' \perp\!\!\!\perp \mathbf{S}$.*

Also, we define $\Omega_{\mathbf{Z}; \mathbf{Y}}$ as $\{\omega | \omega^\intercal \mathbf{Y} \perp\!\!\!\perp \mathbf{Z}\}$, while actually for such $\omega$, $\omega^\intercal \mathbf{Y}$ is independent to more variables:

**Theorem 11** (Full version of $\omega^\intercal \mathbf{Y}$ independence). *For two vertices sets $\mathbf{Z}$ and $\mathbf{Y}$ and their respective critical vertex cut $\mathbf{S}_{\mathbf{Z}, \mathbf{Y}}^*$, for any variable $X_i \in \mathbf{X}$ (i.e., respective vertex $i \in V(G)$), $\omega^\intercal \mathbf{Y} \perp\!\!\!\perp X_i$ if and only if there exists no causal effect from $\mathrm{Anc}_{\mathrm{out}(\mathbf{S}_{\mathbf{Z}, \mathbf{Y}}^*)}(\mathbf{Y})$ to $\{i\}$, i.e., $\mathrm{Anc}_{\mathrm{out}(\mathbf{S}_{\mathbf{Z}, \mathbf{Y}}^*)}(\mathbf{Y}) \cap \mathrm{Anc}(\{i\}) = \varnothing$.*

**Remark 12.** *With Definition 13 and Theorem 11, we can immediately get the following:*

1. *$\omega^\intercal \mathbf{Y} \perp\!\!\!\perp \mathbf{Z}$ - it can be derived from Theorem 11, Definition 11 and Definition 17.*
2. *$\omega^\intercal \mathbf{Y} \perp\!\!\!\perp \mathrm{Anc}(\mathbf{Z})$ - it can be derived from Theorem 11 and Definition 17.*
3. *Theorem 11 is straightforward by seeing $\omega^\intercal \mathbf{Y}$ as a linear transformation of its noise sources $\{E_i | i \in \mathrm{Anc}_{\mathrm{out}(\mathbf{S}_{\mathbf{Z}, \mathbf{Y}}^*)}(\mathbf{Y})\}$. Then any variable is independent to $\omega^\intercal \mathbf{Y}$ if and only if it does not carry noise from these sources (i.e., vertex has no ancestors in $\mathrm{Anc}_{\mathrm{out}(\mathbf{S}_{\mathbf{Z}, \mathbf{Y}}^*)}(\mathbf{Y})$), by the Darmois–Skitovich theorem. With Theorem 11, after testing on $\mathrm{TIN}(\mathbf{Z}, \mathbf{Y})$, one can do more independence test over other variables (as long as they are observed/testable), and may get more information about the whole graph structure and the location of critical vertex cut.*

Further, we notice that in the independent linear transformation subspace $\Omega_{\mathbf{Z}; \mathbf{Y}}$, some indices of $\omega$ may be degenerated (i.e., fixed to zero). Consider following examples for an intuition: 1) On a chain structure Figure 2a with $\mathrm{TIN}(\{X_2\}, \{X_1, X_3, X_4, \cdots, X_n\}) = 2$, $\omega$ index on $X_1$ must

be zero (not include $X_1$ in linear transformation) to make $\omega^\intercal \mathbf{Y} \perp\!\!\!\perp \mathbf{Z}$, while in a fully connected DAG Figure 2c with also $\text{TIN}(\{X_2\}, \{X_1, X_3, X_4, \cdots, X_n\}) = 2$, $\omega$ is not degenerated on any indices. 2) On a chain structure Figure 2a or a chain structure with triangular head Figure 2b, $\text{TIN}(\{X_1, X_3\}, \{X_2, X_4, X_5, \cdots, X_n\}) = 2$ holds, while $\omega$ index on $X_2$ must be zero. 3) in Figure 2d, $\text{TIN}(\{X_1\}, \{X_2, X_5\}) = 1$, while actually $\omega$ is degenerated on $X_5$ index, which means that the linear transformation actually does not include $X_5$ and is just trivially $X_2$ independent of $X_1$ (here $\mathbf{S}^*_{\mathbf{Z}, \mathbf{Y}}$ is just $X_5$).

Now, we would like to first give mathematical characterization for such $\omega$ indices degeneration:

**Theorem 12.** *Since* $\Omega_{\mathbf{Z}; \mathbf{Y}} = \text{null}(\mathbf{B}^\intercal_{\mathbf{Y}, \text{Anc}(\mathbf{Z})})$, $\Omega_{\mathbf{Z}; \mathbf{Y}}$ *degenerates on an index* $y \in \mathbf{Y}$ *if and only if: remove the corresponding y-th column in* $\mathbf{B}^\intercal_{\mathbf{Y}, \text{Anc}(\mathbf{Z})}$ *to get submatrix* $\mathbf{B}^\intercal_{\mathbf{Y} \setminus \{y\}, \text{Anc}(\mathbf{Z})}$, *the rank of submatrix is one less than the rank of full matrix* $\mathbf{B}^\intercal_{\mathbf{Y}, \text{Anc}(\mathbf{Z})}$.

Then we give the equivalent graphical criteria for such $\omega$ indices degeneration:

We already know that the vertex cut $\mathbf{S}^*_{\mathbf{Z}, \mathbf{Y}}$ cuts $\text{Anc}(\mathbf{Z})$ to $\mathbf{Y}$. Moreover, each part of $\mathbf{S}^*_{\mathbf{Z}, \mathbf{Y}}$ has its "own indispensable work" in cutting, so we first define:

**Definition 18** (Local cut scope). For each vertex $s \in \mathbf{S}^*_{\mathbf{Z}, \mathbf{Y}}$, define its local choke scope as $\text{LC}(s) :=$ $\{y \in \mathbf{Y}|$ there exists causal effect from $\text{Anc}(\mathbf{Z})$ to $\{y\}$ without passing through $\mathbf{S}^*_{\mathbf{Z}, \mathbf{Y}} \setminus \{s\}\}$. Furthermore, for each subset $S \subseteq \mathbf{S}^*_{\mathbf{Z}, \mathbf{Y}}$, define $\text{LC}(S) := \{y \in \mathbf{Y}|$ there exists causal effect from $\text{Anc}(\mathbf{Z})$ to $\{y\}$ without passing through $\mathbf{S}^*_{\mathbf{Z}, \mathbf{Y}} \setminus S\}$.

**Remark 13.** *With Definition 18 we have the following:*
1. $\mathbf{S}^*_{\mathbf{Z}, \mathbf{Y}} = \varnothing$ *if and only if* $\mathbf{Z}, \mathbf{Y}$ *are marginally independent, i.e.,* $\mathbf{B}_{\mathbf{Y}, \text{Anc}(\mathbf{Z})}$ *are all zero (no shared noise components).*
2. $\text{LC}(s)$ *means the part of* $\mathbf{Y}$ *that would not be cut/choked, had there been no s. In other word, the part of* $\mathbf{Y}$ *that s has its own indispensable work.*
3. $\text{LC}(S) = \cup_{s \in S} LC(s)$. $\text{LC}(S_1 \cup S_2) = \text{LC}(S_1) \cup \text{LC}(S_2)$.
4. $S \subseteq \text{Anc}(\text{LC}(S))$.
5. *For any subset* $S$, $|\text{LC}(S)| \geq |S|$ *(so* $|\text{LC}(s)| \geq 1$ *for any vertex s).*
6. *For any two different vertices* $s_1, s_2$, *it does not necessarily yield that* $\text{LC}(s_1) \cap \text{LC}(s_2) = \varnothing$ *- they may work together to cut/choke a part and either is indispensable for this part.*
7. $\text{LC}(\mathbf{S}^*_{\mathbf{Z}, \mathbf{Y}})$ *may not be the whole* $\mathbf{Y}$, *but a proper subset. The rest* $\mathbf{Y} \setminus \text{LC}(\mathbf{S}^*_{\mathbf{Z}, \mathbf{Y}})$ *is exactly part of* $\mathbf{Y}$ *that is marginally independent to* $\mathbf{Z}$ *(i.e., no directed paths from* $\text{Anc}(\mathbf{Z})$ *to that part).*

**Theorem 13** (Graphical criteria for degeneration). $\Omega_{\mathbf{Z}; \mathbf{Y}}$ *degenerates on on the indexes subset* $Y \subset \mathbf{Y}$ *if and only if: there exists a subset* $S \subset \mathbf{S}^*_{\mathbf{Z}, \mathbf{Y}}$ *such that its local choke scope* $\text{LC}(S) = Y$, *and* $|Y| = |S| = |\text{LC}(S)|$.

**Remark 14.** *We already have Theorem 12 for math condition. And for graphical criteria Theorem 13:*
1. *A rough interpretation: in general we would expect a smaller $S$ to choke a larger $Y$. However, if for an $S$, its local choke scope $Y$ is of the same size as $S$, then removing $S$ will only affect the same size $Y$ (a feeling that this $S$ is "wasted"). Then this part $Y$ will be degenerated.*
2. *From $Y$ side, it means that this $Y$ requires a same size of separation set $S$ to choke (a feeling that $Y$ is "too expensive").*
3. $\text{TIN}(\mathbf{Z}, \mathbf{Y} \setminus Y) = \text{TIN}(\mathbf{Z}, \mathbf{Y}) - |Y|$, *with critical vertex cut being* $\mathbf{S}^*_{\mathbf{Z}, \mathbf{Y}} \setminus S$, *and with no degeneration.*
4. *Note that degeneration does not yield independence, i.e., if* $\text{TIN}(\mathbf{Z}, \mathbf{Y})$ *with* $\Omega_{\mathbf{Z}; \mathbf{Y}}$ *degenerated on $Y_k$, it does not necessarily yield that* $\omega^\intercal \mathbf{Y} \perp\!\!\!\perp Y_k$. *Because $\omega$ is applied to variables, not noise components. For example, the v-structure* $\{A, B\} \to C$, $\text{TIN}(A, BC) = 1$ *with $C$ degenerated. But* $\omega^\intercal BC$, *which is simply $B$, is not independent to variable $C$.*
5. *The inverse direction of 4. is also not sufficient: if for* $\text{TIN}(\mathbf{Z}, \mathbf{Y})$ *and some* $Y_k \in \mathbf{Y}$, *there is also* $\omega^\intercal \mathbf{Y} \perp\!\!\!\perp Y_k$ *($\omega^\intercal \mathbf{Y}$ independent to not only to $\mathbf{Z}$ but also some part in $\mathbf{Y}$), it still does not necessarily yield that $Y_k$ is degenerated. E.g., in chain structure Figure 2a,* $\text{TIN}(\{\tilde{X}_1\}, \{\tilde{X}_2, \cdots, \tilde{X}_n\}) = 1$ *with* $\omega^\intercal \mathbf{Y}$ *also independent to $\tilde{X}_2$, but $\tilde{X}_2$ index is not degenerated in* $\Omega_{\mathbf{Z}; \mathbf{Y}}$.
6. *Note that while as a special case of point 5, in measurement error case, independence in* $\mathbf{Y}$ *yields degeneration. Because each variable $Y_k \in \mathbf{Y}$ is associated with measurement*

*noise $E_k$ which is only in $Y_k$, not in any other variables (so cannot be cancelled). Then to make $\omega^\intercal\mathbf{Y} \perp\!\!\!\perp Y_k$, at least $E_k$ must be removed, i.e., $Y_k$ degenerated. E.g., in chain structure Figure 2a, $\text{TIN}(\{X_1\},\{X_2,\cdots,X_n\}) = 1$ with no degeneration. So $\omega^\intercal\mathbf{Y}$ is only independent of $X_1$, not any other in observed variables $\mathbf{Y} = \{X_2,\cdots,X_n\}$ - specifically, $\omega^\intercal\mathbf{Y} \perp\!\!\!\perp \{\tilde{X}_1, \tilde{X}_2, X_1\}$.*

Table 2: Full version of Table 1 with more properties on TIN. Examples of TIN on different $(\mathbf{Z},\mathbf{Y})$ pairs over different graph structures in Figure 2.

| $(\mathbf{Z},\mathbf{Y})$ | $(\{X_1,X_2\},\{X_3,X_4,X_5\})$ | | | | $(\{X_1,X_2\},\{X_4,X_5\})$ | | | | $(\{X_3\},\{X_1,X_2,X_4,X_5\})$ | | | | $(\{X_1,X_4\},\{X_3,X_4,X_5\})$ | | | |
|---|---|---|---|---|---|---|---|---|---|---|---|---|---|---|---|---|
| Graph in Figure 2 | (a) | (b) | (c) | (d) | (a) | (b) | (c) | (d) | (a) | (b) | (c) | (d) | (a) | (b) | (c) | (d) |
| $\text{TIN}(\mathbf{Z},\mathbf{Y})$ | 1 | 1 | 2 | 1 | 1 | 1 | 2 | 1 | 3 | 2 | 3 | 1 | 2 | 2 | 3 | 2 |
| $\text{GIN}(\mathbf{Z},\mathbf{Y})$ | TRUE | TRUE | TRUE | TRUE | TRUE | TRUE | FALSE | TRUE | FALSE | FALSE | FALSE | TRUE | TRUE | TRUE | FALSE | TRUE |
| $\dim(\Omega_{\mathbf{Z};\mathbf{Y}})$ | 2 | 2 | 1 | 2 | 1 | 1 | 0 | 1 | 1 | 2 | 1 | 3 | 1 | 1 | 0 | 1 |
| $\text{rk}(\Sigma_{\mathbf{Z}\mathbf{Y}})$ | 1 | 1 | 2 | 1 | 1 | 1 | 2 | 1 | 1 | 1 | 1 | 1 | 2 | 2 | 2 | 2 |
| $\text{Anc}(\mathbf{Z})$ | $X_{1,2}$ | $X_{1,2,3}$ | $X_{1,2}$ | $X_{1,2}$ | $X_{1,2}$ | $X_{1,2,3}$ | $X_{1,2}$ | $X_{1,2}$ | $X_{1,2,3}$ | $X_{1,3}$ | $X_{1,2,3}$ | $X_3$ | $X_{1,2,3,4}$ | $X_{1,3,4}$ | $X_{1,2,3,4}$ | $X_{1,2,3,4}$ |
| $\mathbf{S}^*_{\mathbf{Z},\mathbf{Y}}$ | $X_3$ | $X_3$ | $X_{1,2}$ | $X_4$ | $X_4$ | $X_4$ | $X_{4,5}$ | $X_4$ | $X_{1,2,4}$ | $X_{1,3}$ | $X_{1,2,3}$ | $X_3$ | $X_{3,4}$ | $X_{3,4}$ | $X_{3,4,5}$ | $X_{3,4}$ |
| $\text{A}_{\text{o}(\mathbf{S}^*)}(\mathbf{Y})$ | $X_{4,5}$ | $X_{4,5}$ | $X_{3,4,5}$ | $X_{3,5}$ | $X_5$ | $X_5$ | $\varnothing$ | $X_{3,5}$ | $X_5$ | $X_{2,4,5}$ | $X_{4,5}$ | $X_{1,2,4,5}$ | $X_5$ | $X_5$ | $\varnothing$ | $X_5$ |
| $\mathcal{E}(\omega^\intercal\mathbf{Y})$ | $E_{4,5}$ | $E_{4,5}$ | $E_{3,4,5}$ | $E_{3,5}$ | $E_5$ | $E_5$ | $\varnothing$ | $E_{3,5}$ | $E_5$ | $E_{2,4,5}$ | $E_{4,5}$ | $E_{1,2,4,5}$ | $E_5$ | $E_5$ | $\varnothing$ | $E_5$ |
| $\omega^\intercal\mathbf{Y} \perp\!\!\!\perp$ to | $X_{1,2,3}$ | $X_{1,2,3}$ | $X_{1,2}$ | $X_{1,2}$ | $X_{1,2,3,4}$ | $X_{1,2,3,4}$ | const | $X_{1,2}$ | $X_{1,2,3,4}$ | $X_{1,3}$ | $X_{1,2,3}$ | $X_3$ | $X_{1,2,3,4}$ | $X_{1,2,3,4}$ | const | $X_{1,2,3,4}$ |
| $\omega$ degenerate | \\ | \\ | \\ | \\ | \\ | \\ | $\omega_{4,5}$ | \\ | $\omega_{1,2}$ | \\ | \\ | \\ | $\omega_3$ | $\omega_3$ | $\omega_{3,4,5}$ | \\ |

Table 2 is a full version of Table 1, where we could use examples to better understand the above properties about TIN condition: e.g., different cases for $\text{GIN}(\mathbf{Z},\mathbf{Y})$ to be violated (see rank of $\mathbf{B}_{\mathbf{Y},\text{Anc}(\mathbf{Z})}$ and rank of $\text{cov}(\mathbf{Z},\mathbf{Y})$); noise components of $\omega^\intercal\mathbf{Y}$ is exactly corresponding to $\text{Anc}_{\text{out}(\mathbf{S}^*_{\mathbf{Z},\mathbf{Y}})}(\mathbf{Y})$; the graphical criteria for some $\omega$ indices degeneration, etc.

### D.3 Subsets Implications of the TIN Condition

In §5 we give Theorem 4 for estimation of $\Omega_{\mathbf{Z};\mathbf{Y}}$, by tackling down $\mathbf{Y}$ to subsets:

**Theorem 4** (TIN over $\mathbf{Y}$ subsets). *For two variables sets $\mathbf{Z},\mathbf{Y}$, $\text{TIN}(\mathbf{Z},\mathbf{Y}) = k$ (assume $k > 0$), iff the following two conditions hold: 1) $\forall\mathbf{Y}' \subseteq \mathbf{Y}$ with $|\mathbf{Y}'| = k + 1$ (if any), there exists non-zero $\omega$ s.t. $\omega^\intercal\mathbf{Y}' \perp\!\!\!\perp \mathbf{Z}$; and 2) $\exists\mathbf{Y}' \subseteq \mathbf{Y}$ with $|\mathbf{Y}'| = k$, there exists no non-zero $\omega$ s.t. $\omega^\intercal\mathbf{Y}' \perp\!\!\!\perp \mathbf{Z}$.*

**Remark 15.** *About how to use this "big to small" property, here are some notes:*

1. *Condition 1) can also be "$|\mathbf{Y}'| \geq k + 1$" (a weaker/stronger version).*
2. *This can be shown by that if a set $\mathbf{S}$ is a vertex cut from $\text{Anc}(\mathbf{Z})$ to $\mathbf{Y}$, then $\mathbf{S}$ is also a vertex cut from any subset of $\text{Anc}(\mathbf{Z})$ to any subset of $\mathbf{Y}$.*
3. *It does not yield that all these TIN conditions on subsets $\mathbf{Y}'$ has a same rank $k$, and even with a same rank, not necessarily a same critical vertex cut $\mathbf{S}^*_{\mathbf{Z},\mathbf{Y}}$. E.g., consider a 3-v-structure $\{A,B,C\} \to D$, $\text{TIN}(A,BCD) = 1$ and $\mathbf{S}^*_{\mathbf{Z},\mathbf{Y}} = D$, while $\text{TIN}(A,BC) = 0$ ($\mathbf{Y}' = BC$) and $\mathbf{S}^*_{\mathbf{Z},\mathbf{Y}} = \varnothing$. $\text{TIN}(A,ABD) = 1$ with $\mathbf{S}^*_{\mathbf{Z},\mathbf{Y}} = A$, while $\text{TIN}(A,BD) = 1$ with $\mathbf{S}^*_{\mathbf{Z},\mathbf{Y}} = D$ (though $A$ is still a minimum vertex cut, it is not critical).*
4. *Any transformation vector $\omega \in \Omega_{\mathbf{Z};\mathbf{Y}'}$ is also in $\Omega_{\mathbf{Z};\mathbf{Y}}$, with the other $\mathbf{Y}\backslash\mathbf{Y}'$ indices set to zero.*
5. *It does not yield that for $\mathbf{Y}'$ with $|\mathbf{Y}'| \leq k$ there exists no non-zero vector $\omega$ to make $\omega^\intercal\mathbf{Y}' \perp\!\!\!\perp \mathbf{Z}$ (so in condition 2) it is "$\exists\mathbf{Y}' \subseteq \mathbf{Y}$").*
6. *Theorem 4 can help the estimation of $\Omega_{\mathbf{Z};\mathbf{Y}}$ (existence is easier to check than dimension of all), and can also help the pruning process when we need to test over $\mathbf{Y}$ with size from big to small (to find latent clusters).*

Then, with a same $\mathbf{Y}$ but different $\mathbf{Z}$, we also have the following properties:

**Lemma 2** (Subset of whole independence set). *For two variables sets $\mathbf{Z}$ and $\mathbf{Y}$ and their respective $\text{TIN}(\mathbf{Z},\mathbf{Y})$, denote $\text{Ind}_{\mathbf{Z},\mathbf{Y}} := \{i|\omega^\intercal\mathbf{Y} \perp\!\!\!\perp X_i\}$. From Theorem 11 we have $\text{Ind}_{\mathbf{Z},\mathbf{Y}} = \{i|\,\text{Anc}_{\text{out}(\mathbf{S}^*_{\mathbf{Z},\mathbf{Y}})}(\mathbf{Y}) \cap \text{Anc}(\{i\}) = \varnothing\}$. Then, $\forall\mathbf{Z}' \subseteq \text{Ind}_{\mathbf{Z},\mathbf{Y}}$, $\text{TIN}(\mathbf{Z}',\mathbf{Y}) \leq \text{TIN}(\mathbf{Z},\mathbf{Y})$. Specifically, if $\mathbf{Z} \subseteq \mathbf{Z}'$, then $\text{TIN}(\mathbf{Z}',\mathbf{Y}) = \text{TIN}(\mathbf{Z},\mathbf{Y})$, and moreover, the independent linear transformation subspace is the same: $\Omega_{\mathbf{Z};\mathbf{Y}} = \Omega_{\mathbf{Z}';\mathbf{Y}}$, and the critical vertex cut over all such $\mathbf{Z}'$ is also the same as $\mathbf{S}^*_{\mathbf{Z},\mathbf{Y}}$.*

More properties about subset implications (e.g., combination and expansion of $\mathbf{Z}$ and more independent variables) can be derived from e.g., Theorem 11. Another interesting question is, except for pruning in practical algorithms or for easier estimation, how to use these subset implication relationships to help identify the graph structure?

# E  Methods Details for Estimating $\Omega_{\mathbf{Z};\mathbf{Y}}$

## E.1  For TIN-rank: Stacked Cumulants

To estimate the subspace $\Omega_{\mathbf{Z};\mathbf{Y}}$, we give a method named "ranks stopped increasing" in §5.3 based on cumulants among variables. Now we give more details on this method.

**Definition 19** (Cumulants [31])**.** Define cumulant among $k$ variables $X_{i_1}, \ldots, X_{i_k}$ as:

$$\mathrm{cum}\,(X_{i_1}, \ldots, X_{i_k}) = \sum_{(A_1, \ldots, A_L)} (-1)^{L-1}(L-1)! \mathbb{E}\left[\prod_{j \in A_1} X_j\right] \mathbb{E}\left[\prod_{j \in A_2} X_j\right] \cdots \mathbb{E}\left[\prod_{j \in A_L} X_j\right], \tag{E.1}$$

where the sum is taken over all partitions $(A_1, \ldots, A_L)$ of the set $\{i_1, \ldots, i_k\}$.

**Remark 16.** *About cumulant defined in Definition 19:*

1. *Suppose variables are zero-meaned, then sum is taken over all partitions where each $A_i$ has size at least 2. For example, in the following:*
2. $\mathrm{cum}(X_i) = 0$.
3. $\mathrm{cum}(X_{i_1}, X_{i_2}) = \mathbb{E}[X_{i_1} X_{i_2}] = \mathrm{cov}(X_{i_1}, X_{i_2})$.
4. $\mathrm{cum}(X_{i_1}, X_{i_2}, X_{i_3}) = \mathbb{E}[X_{i_1} X_{i_2} X_{i_3}] = $ *3rd order moment of* $(X_{i_1}, X_{i_2}, X_{i_3})$.
5. $\mathrm{cum}(X_{i_1}, X_{i_2}, X_{i_3}, X_{i_4}) = \mathbb{E}[X_{i_1} X_{i_2} X_{i_3} X_{i_4}] - \mathbb{E}[X_{i_1} X_{i_2}]\mathbb{E}[X_{i_3} X_{i_4}]$
$$- \mathbb{E}[X_{i_1} X_{i_3}]\mathbb{E}[X_{i_2} X_{i_4}]$$
$$- \mathbb{E}[X_{i_1} X_{i_4}]\mathbb{E}[X_{i_2} X_{i_3}].$$
6. *As is shown above, the 4-th order cumulant is not equal to the 4-th order momentum. In general, cumulant$\neq$momentum when order $k \geq 4$. We use cumulant, for reason in point 7:*
7. *If variables $X_{i_1}, \ldots, X_{i_k}$ are mutually independent, then $\mathrm{cum}\,(X_{i_1}, \ldots, X_{i_k}) = 0$. Note that it is zero cumulant, not zero momentum.*

**Definition 20** (Cross cumulant tensor)**.** For a random vector $\mathbf{X} = [X_1, \cdots, X_m]^{\mathsf{T}}$, denote its cross cumulant tensor at order $k$ as $\mathcal{T}_{\mathbf{X}}^{(k)}$, an $\underbrace{m \times \cdots \times m}_{k \text{ times}}$ tensor, where each entry

$$\mathcal{T}_{\mathbf{X}}^{(k)}{}_{i_1, \cdots, i_k} := \mathrm{cum}(X_{i_1}, \cdots, X_{i_k}). \tag{E.2}$$

Now suppose these random variables follow a linear acyclic SEM model, with $\mathbf{X} = \mathbf{A}\mathbf{X} + \mathbf{E}$. Because of acyclicity, we could also write $\mathbf{X} = \mathbf{B}\mathbf{E}$, where $\mathbf{B} = (\mathbf{I} - \mathbf{A})^{-1}$. Then we have the following:

**Theorem 14** (Cross cumulant tensor in linear acyclic SEM)**.** *$k$-th order cross cumulant tensor equals*

$$\mathcal{T}_{\mathbf{X}}^{(k)} = \mathcal{T}_{\mathbf{E}}^{(k)} \cdot \underbrace{\mathbf{B} \cdot \cdots \cdot \mathbf{B}}_{k \text{ times}}, \tag{E.3}$$

*where $\mathcal{T}_{\mathbf{E}}^{(k)}$ is the $k$-th order cross cumulant tensor of $\mathbf{E}$, and '$\cdot$' denotes the tensor dot, i.e.,*

$$\mathcal{T}_{\mathbf{X}}^{(k)}{}_{i_1, \cdots, i_k} = \sum_{j_1, \cdots, j_k} \mathcal{T}_{\mathbf{E}}^{(k)}{}_{j_1, \cdots, j_k} \mathbf{B}_{i_1, j_1} \cdots \mathbf{B}_{i_k, j_k} \tag{E.4}$$

*Since exogenous noises $\mathbf{E}$ are mutually independent, $\mathcal{T}_{\mathbf{E}}^{(k)}$ is a diagonal tensor. In this case, the above equation needs not to be summed over all Cartesian product $[m]^k$, but just over each $j \in [m]$.*

**Remark 17.** *About cross cumulant tensor in Linear acyclic SEM in Theorem 14:*

1. *For example, in 2nd order case, $\mathcal{T}_{\mathbf{X}}^{(2)}$ is the cross covariance matrix $\Sigma := \mathrm{cov}(\mathbf{X}, \mathbf{X})$. We have $\Sigma = \mathbf{B}\Phi\mathbf{B}^{\mathsf{T}}$, where $\Phi$ is a diagonal matrix with entries $\Phi_{i,i} = \mathrm{var}(E_i)$.*
2. *Proof to 1: for every two variables $X_i, X_j$, $\mathrm{cov}(X_i, X_j) = \sum_k \mathbf{B}_{ik}\mathbf{B}_{jk}\,\mathrm{var}(E_k)$.*
3. *Point 2 means that the covariance between $X_i, X_j$ is contributed by all noise that is contained in both $X_i$ and $X_j$. By 'common noise', we mean 'confounders', 'common ancestors', or the 'top-node' of each trek between $(X_i, X_j)$ - and this is the start of the proof to trek-separation.*
4. *In general, any order of the cumulant $\mathrm{cum}(X_{i_1}, \cdots, X_{i_k})$ is contributed by the 'common noise' that $X_{i_1}, \cdots, X_{i_k}$ all share, i.e., $\bigcap_{l \in [k]} \mathrm{Anc}(X_{i_l})$, the common ancestors.*

Since we only care the pairwise relationship between any **two** subsets $\mathbf{Z}, \mathbf{Y}$, we can take a 2D matrix slice out from each order of cross cumulant tensors:

**Definition 21** (2D slice of cross cumulant tensor). For a random vector $\mathbf{X}$ with $k$-th order cross cumulant tensor $\mathcal{T}_{\mathbf{X}}^{(k)}$, denote its 2D matrix slice of $k$-th order cross cumulant tensor as $\mathcal{C}^{(k)}$, where

$$\mathcal{C}_{i,j}^{(k)} := \mathrm{cum}(\underbrace{X_i, \cdots, X_i}_{k-1 \text{ times}}, X_j) = \mathcal{T}_{\mathbf{X}}^{(k)}{}_{i,\cdots,i,j}. \tag{E.5}$$

**Remark 18.** *About 2D slice of cross cumulant tensor defined in Definition 21:*

1. *For simplicity, here we omit the subscript $\mathbf{X}$ in $\mathcal{C}_{\mathbf{X}}^{(k)}$ and just write as $\mathcal{C}^{(k)}$.*
2. *In particular, when $k = 2$, $\mathcal{C}^{(2)}$ is the variance covariance matrix $\Sigma_{\mathbf{X}}$.*
3. *$\mathcal{C}^{(k)}$ is $n \times n$ matrix, and is not necessarily symmetric when $k > 2$.*

Then similar to Theorem 14, we formulate 2D slice of cross cumulant tensor in linear acyclic SEM:

**Theorem 15** (2D slice of cross cumulant tensor in linear acyclic SEM). *$\mathcal{C}^{(k)}$ equals*

$$\mathcal{C}^{(k)} = \mathbf{B}^{k-1} \cdot \Phi_{\mathbf{E}}^{(k)} \cdot \mathbf{B}^{\mathsf{T}}, \tag{E.6}$$

*where $\mathbf{B}^{k-1}$ is the element-wise power (i.e., $\mathbf{B}^{k-1} = \underbrace{\mathbf{B} \circ \cdots \circ \mathbf{B}}_{k-1 \text{ times}}$, '$\circ$' denotes element-wise product (Hadamard product), and $\Phi_{\mathbf{E}}^{(k)}$ is a diagonal matrix with entries $\Phi_{\mathbf{E}}^{(k)}{}_{i,i} = \mathrm{cum}(\underbrace{E_i, \cdots E_i}_{k \text{ times}})$.*

*Moreover, for two vertices sets $\mathbf{Z}, \mathbf{Y}$, similar to Theorem 1, we have*

$$\begin{aligned} \mathcal{C}_{\mathbf{Z},\mathbf{Y}}^{(k)} &= \mathbf{B}_{\mathbf{Z},:}^{k-1} \cdot \Phi_{\mathbf{E}}^{(k)} \cdot \mathbf{B}_{\mathbf{Y},:}^{\mathsf{T}} \\ &= \mathbf{B}_{\mathbf{Z},\mathrm{Anc}(\mathbf{Z})}^{k-1} \cdot \Phi_{\mathbf{E}}^{(k)} \cdot \mathbf{B}_{\mathbf{Y},\mathrm{Anc}(\mathbf{Z})}^{\mathsf{T}}, \end{aligned} \tag{E.7}$$

*where e.g., $\mathcal{C}_{\mathbf{Z},\mathbf{Y}}^{(k)}$ denotes the submatrix of $\mathcal{C}^{(k)}$ with rows indexed by $\mathbf{Z}$ and columns indexed by $\mathbf{Y}$.*

Proof to Theorem 15 is straightforward by plugging Definition 21 into tensor dot of Theorem 14.

Since independence yields zero cumulant, we have that for two vertices sets $\mathbf{Z}, \mathbf{Y}$ and $\omega \in \mathbb{R}^{|\mathbf{Y}|}$, if $\omega^{\mathsf{T}} \mathbf{Y} \perp\!\!\!\perp \mathbf{Z}$, then $\mathcal{C}_{\mathbf{Z},\mathbf{Y}}^{(k)} \omega = 0$. In other words,

$$\Omega_{\mathbf{Z};\mathbf{Y}} \subseteq \mathrm{null}(\mathcal{C}_{\mathbf{Z},\mathbf{Y}}^{(k)}), \text{ for any } k \geq 2. \tag{E.8}$$

This can be shown by two ways: one is that $\mathrm{cum}(\mathbf{Z}, \cdots, \mathbf{Z}, \omega^{\mathsf{T}} \mathbf{Y}) = \mathrm{cum}(\mathbf{Z}, \cdots, \mathbf{Z}, \mathbf{Y})\omega$, another is to use Equation (4) we build in Theorem 1: $\omega^{\mathsf{T}} \mathbf{Y} \perp\!\!\!\perp \mathbf{Z} \Leftrightarrow \mathbf{B}_{\mathbf{Y},\mathrm{Anc}(\mathbf{Z})}^{\mathsf{T}} \omega = 0$.

We shall also recap the original GIN condition: first solve equation by $\mathrm{cov}(\mathbf{Z}, \mathbf{Y})$, then check whether any solution $\omega$ satisfies $\omega^{\mathsf{T}} \mathbf{Y} \perp\!\!\!\perp \mathbf{Z}$ (i.e., whether $\mathrm{null}(\mathrm{cov}(\mathbf{Z}, \mathbf{Y})) = \Omega_{\mathbf{Z};\mathbf{Y}}$). However, when GIN is not satisfied (i.e., $\Omega_{\mathbf{Z};\mathbf{Y}} \subsetneq \mathrm{null}(\mathrm{cov}(\mathbf{Z}, \mathbf{Y}))$), it is not necessarily that $\Omega_{\mathbf{Z};\mathbf{Y}} = \mathbb{R}_{\mathbf{0}}$ - e.g., the rank may just be limited by the size of $\mathbf{Z}$. This is exactly the motivation why we need to further generalize GIN to TIN: can we escape from the 'unwanted restriction on rank' (e.g., size of $\mathbf{Z}$) and find exactly the $\Omega_{\mathbf{Z};\mathbf{Y}}$? Fortunately, by above implication from independence to zero cumulant, we could solve equation not only by 2-nd order $\mathrm{cov}(\mathbf{Z}, \mathbf{Y})$, but more (on any order) $\mathcal{C}_{\mathbf{Z},\mathbf{Y}}^{(k)}$.

**Definition 22** (Stacked 2D slices of cumulants). For two vertices sets $\mathbf{Z}, \mathbf{Y}$ and order $k \geq 2$, define:

$$\Psi_{\mathbf{Z};\mathbf{Y}}^{(k)} := \begin{bmatrix} \mathcal{C}_{\mathbf{Z},\mathbf{Y}}^{(2)} \\ \vdots \\ \mathcal{C}_{\mathbf{Z},\mathbf{Y}}^{(k)} \end{bmatrix} \tag{E.9}$$

$\Psi_{\mathbf{Z};\mathbf{Y}}^{(k)}$ is a $(k-1)|\mathbf{Z}| \times |\mathbf{Y}|$ matrix that vertically stacks 2D cumulants slices between $\mathbf{Z}, \mathbf{Y}$ with order from 2 to $k$. Since independence yields zero cumulant, similarly we have

$$\Omega_{\mathbf{Z};\mathbf{Y}} \subset \mathrm{null}(\Psi_{\mathbf{Z};\mathbf{Y}}^{(k)}), \text{ for any } k \geq 2. \tag{E.10}$$

For example, a fully connected DAG with 4 variables $\{X_1, X_2, X_3, X_4\}^\intercal$, the edges parameters are:

$$\mathbf{A} = \begin{bmatrix} 0 & 0 & 0 & 0 \\ a & 0 & 0 & 0 \\ b & d & 0 & 0 \\ c & e & f & 0 \end{bmatrix}; \mathbf{B} = \begin{bmatrix} 1 & 0 & 0 & 0 \\ a & 1 & 0 & 0 \\ ad+b & d & 1 & 0 \\ a(df+e)+bf+c & df+e & f & 1 \end{bmatrix} \tag{E.11}$$

Denote cumulants of exogenous noises $\varphi_i^{(k)} := \mathrm{cum}(\underbrace{E_i, \cdots, E_i}_{k \text{ times}})$.

1) Let $\mathbf{Z} := \{X_1\}, \mathbf{Y} := \{X_2, X_3, X_4\}$, we have:

$$\Psi_{\mathbf{Z};\mathbf{Y}}^{(2)} = \begin{bmatrix} a\varphi_1^{(2)} & (ad+b)\varphi_1^{(2)} & (a(df+e)+bf+c)\varphi_1^{(2)} \end{bmatrix};$$

$$\Psi_{\mathbf{Z};\mathbf{Y}}^{(3)} = \begin{bmatrix} a\varphi_1^{(2)} & (ad+b)\varphi_1^{(2)} & (a(df+e)+bf+c)\varphi_1^{(2)} \\ a\varphi_1^{(3)} & (ad+b)\varphi_1^{(3)} & (a(df+e)+bf+c)\varphi_1^{(3)} \end{bmatrix}; \cdots \tag{E.12}$$

The independence subspace

$$\Omega_{\mathbf{Z};\mathbf{Y}} = \mathrm{null}(\mathbf{B}_{\mathbf{Y},Anc(Z)}^\intercal) = \mathrm{null}\left( \begin{bmatrix} a & ad+b & a(df+e)+bf+c \end{bmatrix} \right), \text{ dimension=2.} \tag{E.13}$$

Observe that $\mathrm{null}(\Psi_{\mathbf{Z};\mathbf{Y}}^{(2)}) = \Omega_{\mathbf{Z};\mathbf{Y}}$, (and also $= \mathrm{null}(\Psi_{\mathbf{Z};\mathbf{Y}}^{(3)}) = \cdots$).

2) Let $\mathbf{Z} := \{X_2\}, \mathbf{Y} := \{X_1, X_3, X_4\}$, we have:

$$\Psi_{\mathbf{Z};\mathbf{Y}}^{(2)} = \begin{bmatrix} a\varphi_1^{(2)} & a(ad+b)\varphi_1^{(2)} + d\varphi_2^{(2)} & a(a(df+e)+bf+c)\varphi_1^{(2)} + (df+e)\varphi_2^{(2)} \end{bmatrix};$$

$$\Psi_{\mathbf{Z};\mathbf{Y}}^{(3)} = \begin{bmatrix} a\varphi_1^{(2)} & a(ad+b)\varphi_1^{(2)} + d\varphi_2^{(2)} & a(a(df+e)+bf+c)\varphi_1^{(2)} + (df+e)\varphi_2^{(2)} \\ a^2\varphi_1^{(3)} & a^2(ad+b)\varphi_1^{(3)} + d\varphi_2^{(3)} & a^2(a(df+e)+bf+c)\varphi_1^{(3)} + (df+e)\varphi_2^{(3)} \end{bmatrix}; \cdots$$
$$\tag{E.14}$$

The independence subspace

$$\Omega_{\mathbf{Z};\mathbf{Y}} = \mathrm{null}(\mathbf{B}_{\mathbf{Y},Anc(Z)}^\intercal) = \mathrm{null}\left( \begin{bmatrix} 1 & ad+b & a(df+e)+bf+c \\ 0 & d & df+e \end{bmatrix} \right), \text{ dimension=1.} \tag{E.15}$$

Clearly $\mathrm{null}(\Psi_{\mathbf{Z};\mathbf{Y}}^{(2)}) \neq \Omega_{\mathbf{Z};\mathbf{Y}}$, since the rank of $\Psi_{\mathbf{Z};\mathbf{Y}}^{(2)}$ is only 1. However, as long as there is no parameter coupling in cumulants, or specifically, $\frac{a\varphi_1^{(3)}}{\varphi_1^{(2)}} \neq \frac{\varphi_2^{(3)}}{\varphi_2^{(2)}}$, then $\mathrm{null}(\Psi_{\mathbf{Z};\mathbf{Y}}^{(3)}) = \Omega_{\mathbf{Z};\mathbf{Y}}$ (with the rank increasing to 2). We could verify the solution:

$$\omega^\intercal \mathbf{Y} = \frac{cd - be}{d}E_1 + \frac{df+e}{d}((ad+b)E_1 + dE_2 + E_3)$$
$$- ((a(df+e)+bf+c)E_1 + (df+e)E_2 + fE_3 + E_4) \tag{E.16}$$
$$= \text{contains only } \{E_C, E_D\}, \text{ and thus } \omega^\intercal \mathbf{Y} \perp\!\!\!\perp \mathbf{Z}.$$

According to original GIN definition, there is only GIN($\{X_1\}, \{X_2, X_3, X_4\}$), and $X_2, X_3, X_4$ cannot be distinguished. However here by using TIN, we could also identify $X_2$.

3) Let $\mathbf{Z} := \{X_3\}$ or $\{X_4\}, \mathbf{Y} := \mathbf{X} \backslash \mathbf{Z}$, there is no non-zero $\omega$ s.t., $\omega^\intercal \mathbf{Y} \perp\!\!\!\perp \mathbf{Z}$. Observe that:

$$\Omega_{\mathbf{Z};\mathbf{Y}} = \mathbb{R}_{\mathbf{0}} = \mathrm{null}(\Psi_{\mathbf{Z};\mathbf{Y}}^{(k)}) = \cdots = \mathrm{null}(\Psi_{\mathbf{Z};\mathbf{Y}}^{(4)}) \subsetneq \mathrm{null}(\Psi_{\mathbf{Z};\mathbf{Y}}^{(3)}) \subsetneq \mathrm{null}(\Psi_{\mathbf{Z};\mathbf{Y}}^{(2)}).$$

Above example gives us a motivation to use a sequence of stacked 2D cumulants $\{\Psi_{\mathbf{Z};\mathbf{Y}}^{(i)}\}_{i=2,3,\cdots}$.

**Remark 19.** *About this sequence of stacked 2D cumulants, we have:*

1. *$\Psi_{\mathbf{Z};\mathbf{Y}}^{(i+1)}$ contains $\Psi_{\mathbf{Z};\mathbf{Y}}^{(i)}$ as some-rows-indexed submatrix, so:*
2. *Rank does not drop, i.e., $\mathrm{rank}(\Psi_{\mathbf{Z};\mathbf{Y}}^{(i+1)}) \geq \mathrm{rank}(\Psi_{\mathbf{Z};\mathbf{Y}}^{(i)})$.*
3. *Nullspaces $\mathrm{null}(\Psi_{\mathbf{Z};\mathbf{Y}}^{(i+1)}) \subseteq \mathrm{null}(\Psi_{\mathbf{Z};\mathbf{Y}}^{(i)})$.*
4. *Independent subspace $\Omega_{\mathbf{Z};\mathbf{Y}} \subseteq \mathrm{null}(\Psi_{\mathbf{Z};\mathbf{Y}}^{(i)})$, for any $k \geq 2$.*

5. $\operatorname{rank}(\Psi_{\mathbf{Z};\mathbf{Y}}^{(i)}) \leq |\mathbf{Y}| - \dim(\Omega_{\mathbf{Z};\mathbf{Y}})$, *for any $k \geq 2$.*

Note that in above statements, no assumptions on edge parameters and noise components' cumulants are made, and they are purely by definition. Then, does there exist a finite integer $K \in \mathbb{N}^+$ where the shrinking nullspaces stop hereafter at $\Omega_{\mathbf{Z};\mathbf{Y}}$, i.e., $\Omega_{\mathbf{Z};\mathbf{Y}} = \operatorname{null}(\Psi_{\mathbf{Z};\mathbf{Y}}^{(i)})$, for any $i \geq K$? The answer is yes, under the generic assumptions on edge parameters and noise components' cumulants:

**Assumption 2** (Generic edge parameters and noise components' cumulants). On a LiNGAM instance $\mathbf{L} = \mathcal{G}(G, \mathbf{B}, \mathbf{E})$ defined by graph structure $G$, edge parameters $\mathbf{B}$ and noise components $\mathbf{E}$, assume that for any two variables sets $\mathbf{Z}, \mathbf{Y}$ and order $k \geq 2$,

$$\operatorname{rank}(\Psi_{\mathbf{Z};\mathbf{Y}}^{(k)}; \mathbf{L}) = \max_{\mathbf{B}', \mathbf{E}'} \{\operatorname{rank}(\Psi_{\mathbf{Z};\mathbf{Y}}^{(k)}; \mathbf{L}') \mid \mathbf{L}' = \mathcal{G}(G, \mathbf{B}', \mathbf{E}')\}, \tag{E.17}$$

where $\mathbf{B}', \mathbf{E}'$ are traversed over the whole edge parameters and noise components space. This is to assume that there is no coincidental low rank parameterized by the LiNGAM instance $\mathbf{L}$. Note that Assumption 2 is stronger than Assumption 1 in §3. Here Assumption 2 assumes not only generic edge parameters, but also noise parameters.

Under Assumption 2 we have the following graphical criteria over stacked 2D cumulants:

**Theorem 16.** *For two vertices sets $\mathbf{Z}, \mathbf{Y}$ and order $k \geq 2$, we define a new DAG associated with $G$, denoted as $\hat{G}^{(k)}$, which has $kn$ vertices $\{1, 2, \cdots, n\} \cup \{1^{(2)}, 2^{(2)}, \cdots, n^{(2)}\} \cup \cdots \cup \{1^{(k)}, 2^{(k)}, \cdots, n^{(k)}\}$ with edges $i \to j$ if $i \to j$ is in $G$, $\{j^{(l)} \to i^{(l)}\}_{l=2,\cdots,k}$ if $i \to j$ is in $G$, and $\{i^{(l)} \to i\}_{l=2,\cdots,k}$ for $i \in [n]$. Define a new vertices set $\mathbf{Z}' := \cup\{i^{(2)}, \cdots, i^{(k)}\}_{i \in \mathbf{Z}}$, then we have:*

$$\operatorname{rank}(\Psi_{\mathbf{Z};\mathbf{Y}}^{(k)}) = \min\{|\mathbf{S}| \mid \mathbf{S} \text{ is a vertex cut from } \mathbf{Z}' \text{ to } \mathbf{Y} \text{ on } \hat{G}^{(k)}\}. \tag{E.18}$$

Note that the trek-separation theorem can be viewed as a special case of Theorem 16 with $k = 2$, where "$(\mathbf{S}_{\mathbf{W}}, \mathbf{S}_{\mathbf{Y}})$ t-separates $(\mathbf{W}, \mathbf{Y})$" is equivalent to "$\mathbf{S}_{\mathbf{W}}' \cup \mathbf{S}_{\mathbf{Y}}$ vertex cuts $\mathbf{W}'$ to $\mathbf{Y}$". The proof to Theorem 16 also basically follow the proof to Theorem 2.8 in [47]: using the Lindström-Gessel-Viennot theorem [25, 13], the max-flow min-cut theorem (vertex version, known as Menger's theorem) [9, 3, 26], and applying the Cauchy–Binet determinant expansion formula and Schur properties repeatedly on the Hadamard products in Equation (E.6).

With the graphical criteria stated in Theorem 16 and under generic Assumption 2, we could have a method to implement TIN by ranks of stacked cumulants in sequence:

**Theorem 17** (Use ranks' stopped increasing to implement TIN). *For two variables sets $\mathbf{Z}, \mathbf{Y}$, there must exists a finite order $k \geq 2$ s.t.*

$$\operatorname{rank}(\Psi_{\mathbf{Z};\mathbf{Y}}^{(k+1)}) = \operatorname{rank}(\Psi_{\mathbf{Z};\mathbf{Y}}^{(k)}). \tag{E.19}$$

*Moreover, this one-step-stop yields an infinite-steps-stop, i.e.,*

$$\operatorname{rank}(\Psi_{\mathbf{Z};\mathbf{Y}}^{(l)}) = \operatorname{rank}(\Psi_{\mathbf{Z};\mathbf{Y}}^{(k)}), \text{ for any } l > k. \tag{E.20}$$

*and, this stopped-increasing rank equals exactly to* $\operatorname{TIN}(\mathbf{Z}, \mathbf{Y})$*, i.e., s.t.*

$$\operatorname{rank}(\Psi_{\mathbf{Z};\mathbf{Y}}^{(k)}) = \operatorname{TIN}(\mathbf{Z}, \mathbf{Y}) = |\mathbf{Y}| - \dim(\Omega_{\mathbf{Z};\mathbf{Y}}). \tag{E.21}$$

The original GIN condition using only covariance matrix can be viewed as a special case, which could be implemented as "$\operatorname{rank}(\Psi_{\mathbf{Z};\mathbf{Y}}^{(2)}) = \operatorname{rank}(\Psi_{\mathbf{Z};\mathbf{Y}}^{(3)})$".

Note that independence test is not used in this method. We could also use independence tests to test whether $\operatorname{null}(\Psi_{\mathbf{Z};\mathbf{Y}}^{(k)})$ is equal to $\Omega_{\mathbf{Z};\mathbf{Y}}$, just like the 2-steps method in §5.4. Independence yields zero cumulants, and also yields independence among functions of variables. Hence in term of solving equations system, $\operatorname{null}(\Psi_{\mathbf{Z};\mathbf{Y}}^{(k)})$ and $\operatorname{cov}(f(\mathbf{Z}), \mathbf{Y})\omega = \mathbf{0}$ are both correct. However, the latter does not have additional graphical criteria as Theorem 16. Empirically, the latter performs better, since higher order cumulants yield higher order exponential, which is sensitive to outliers.

### E.2 For TIN-2steps, TIN-subsets, and TIN-ISA

Implementation details for these three methods can be referred in Appendix G.1. Specifically, TIN-ISA directly follows the Independent Subspace Analysis (ISA) from the original paper [48].

## F Discussions

### F.1 Details on Assumptions

In this paper, except for the LiNGAM assumption for the causal model, we also give Assumption 1 in §3:

**Assumption 1** (Rank faithfulness). Denote by $\mathcal{B}(G)$ the parameter space of mixing matrix $\mathbf{B}$ consistent with the DAG $G$. For any two subsets of variables $\mathbf{Z}, \mathbf{Y} \subseteq \mathbf{X}$, we assume that

$$\text{rank}(\mathbf{B}_{\mathbf{Y},\text{Anc}(\mathbf{Z})}) = \max_{\mathbf{B}' \in \mathcal{B}(G)} \text{rank}(\mathbf{B}'_{\mathbf{Y},\text{Anc}(\mathbf{Z})}). \tag{6}$$

Roughly speaking, Assumption 1 assumes there are no edge parameter couplings to produce coincidental low rank. Note that violation of Assumption 1 is of Lebesgue measure 0, and LiNGAM is testable. Here we discuss more details on Assumption 1 by two examples of violation:

$$X_1 \xrightarrow{a} X_2 \xrightarrow{b} X_3 \atop c \tag{F.1}$$

*Violation example 1*: Consider the graph in Equation (F.1), if coincidentally the edge weights $c = -ab$, then the noise components $E_1$ will be cancelled from $X_3$, and marginally $X_1 \perp\!\!\!\perp X_3$. In this violation, graphically $\text{Anc}(X_3) = \{X_1, X_2, X_3\}$, but the column indices of $\mathbf{B}_{X_3,:}$ with non-zero entries is just $\{X_2, X_3\}$.

$$X_3 \xrightarrow{e} X_4 \; ; \; \mathbf{A} = \begin{bmatrix} 0 & 0 & 0 & 0 \\ 0 & 0 & 0 & 0 \\ c & a & 0 & 0 \\ d & b & e & 0 \end{bmatrix} \; ; \; \mathbf{B} = \begin{bmatrix} 1 & 0 & 0 & 0 \\ 0 & 1 & 0 & 0 \\ c & a & 1 & 0 \\ ce+d & ae+b & e & 1 \end{bmatrix} \tag{F.2}$$

*Violation example 2*: Consider the graph in Equation (F.2). Let $\mathbf{Z} := \{X_1, X_2\}$ and $\mathbf{Y} := \{X_3, X_4\}$, by the graphical criteria we have $\text{TIN}(\mathbf{Z}, \mathbf{Y}) = 2$, with the critical vertex cut $\mathbf{S}^*_{\mathbf{Z};\mathbf{Y}} = \{X_3, X_4\}$. Mathematically, $\mathbf{B}_{\mathbf{Y};\text{nzcol}(\mathbf{B}_{\mathbf{Z},:})} = \begin{bmatrix} c & a \\ ce+d & ae+b \end{bmatrix}$, which has rank 2 under generic parameters choice. However, if $bc = ad$, then coincidentally the rank will drop to 1, and thus Assumption 1 is violated. Note that in this violation example, there is no noise cancelling (like violation example 1), i.e., here $\text{nzcol}(\mathbf{B}_{\mathbf{Z},:})$ is exactly $\text{Anc}(\mathbf{Z})$, but there is still coincidental low rank by parameter coupling.

Now we further discuss an example where Assumption 1 is satisfied (and thus is a valid case in this paper), but is not a valid case in the trek-separation paper [47] or the GIN paper [50]:

$$X_1 \xrightarrow{2} X_3 \xrightarrow{1} X_4 \atop {1 \searrow \quad \nearrow -1 \atop X_2} \; ; \; \mathbf{A} = \begin{bmatrix} 0 & 0 & 0 & 0 \\ 1 & 0 & 0 & 0 \\ 2 & -1 & 0 & 0 \\ 0 & 0 & 1 & 0 \end{bmatrix} \; ; \; \mathbf{B} = \begin{bmatrix} 1 & 0 & 0 & 0 \\ 1 & 1 & 0 & 0 \\ 1 & -1 & 1 & 0 \\ 1 & -1 & 1 & 1 \end{bmatrix} \tag{F.3}$$

*Satisfaction example 3*: Consider the graph in Equation (F.2). For every pair of $\mathbf{Z}, \mathbf{Y}$, there is no coincidental low rank in $\mathbf{B}_{\mathbf{Y},\text{Anc}(\mathbf{Z})}$. Hence, Assumption 1 is satisfied. E.g., let $\mathbf{Z} := \{X_2\}, \mathbf{Y} := \{X_3, X_4\}$, by the graphical criteria $\text{TIN}(\mathbf{Z}, \mathbf{Y}) = 1$ (with $\mathbf{S}^*_{\mathbf{Z};\mathbf{Y}} = \{X_3\}$), and $\mathbf{B}_{\mathbf{Y},\text{Anc}(\mathbf{Z})}$ is also of rank 1. However, if we carefully choose noise components' parameters so that the variance of exogenous noise $E_1$ and $E_2$ are equal ($\text{var}(E_1) = \text{var}(E_2)$), then the variance-covariance matrix $\text{cov}(\{X_2\}, \{X_3, X_4\})$ would be $[0 \quad 0]$ (coincidentally dropped to rank 1). This coincidental low rank is due to noise parameters, and will not affect our proposed method in this paper, because we directly find $\Omega_{\mathbf{Z};\mathbf{Y}}$. However, e.g., in GIN where $\omega$ is characterized by 2nd-order variance-covariance matrix, by solving equation here, any $w \in \mathbb{R}^2$ is a solution. Then, not every linear combination of $X_3$ and $X_4$ is independent to $X_2$, so GIN will output 'GIN$(\mathbf{Z}, \mathbf{Y})$ violated' in this case, though according to the graphical criteria, GIN$(\mathbf{Z}, \mathbf{Y})$ is satisfied here.

## F.2 More than Ordered Group Decomposition can be Identified

In this paper, we use the TIN condition to identify the ordered group decomposition of $\tilde{G}$ in the measurement error model. Specifically, we only use a special type of TIN, one-and-others (Lemma 1). However, actually by using the TIN condition over more general pairs of $\mathbf{Z}, \mathbf{Y}$, more information of $\tilde{G}$ can be identified.

For example, in the chain structure (Figure 2a) and the fully connected DAG (Figure 2c), the ordered group decomposition are both $\{\tilde{X}_1\} \to \{\tilde{X}_2\} \to \cdots \to \{\tilde{X}_{n-2}\} \to \{\tilde{X}_{n-1}, \tilde{X}_n\}$. However, the two can actually be distinguished: In the fully connected DAG, $\text{TIN}(\{X_2\}, \{X_3, \cdots, X_n\}) = 2$, while in the chain structure, $\text{TIN}(\{X_2\}, \{X_3, \cdots, X_n\}) = 1$. Even under a same pair of $\mathbf{Z}, \mathbf{Y}$, the $\omega$ degeneration may be different. E.g., $\text{TIN}(\{X_2\}, \{X_1, X_3, X_4, \cdots\}) = 2$ in both graphs. However, in the chain structure, $\omega$ is degenerated on the index $X_1$ (i.e., the linear combination of $\mathbf{Y}$ cannot include $X_1$. If $\omega_1 X_1 + \omega_3 X_3 + \omega_4 X_4 + \cdots$ is independent to $X_2$, then $\omega_1$ must be zero), while there is no degeneration of $\omega$ in the fully connected DAG.

Generally speaking, our final objective is to identify an "equivalence class" of $\tilde{G}$ w.r.t. the TIN condition. We have talked about the concept of "unidentifiable" in §2. Here, two graphs (either non-isomorphic or isomorphic but with labelling permutation) are unidentifiable w.r.t. the TIN condition, if and only if for any two pairs $\mathbf{Z}, \mathbf{Y}$, $\text{TIN}(\mathbf{Z}, \mathbf{Y})$ are same (with same degeneration).

About "equivalence class", we already knew some features that an equivalence class should possess, e.g., a variable is naturally unidentifiable with its pure leaf child in $\tilde{G}$ (see Definition 6). Apparently, there are more such features to be discovered. Here are some of the examples:

**Example 8** (Equivalence class for the chain structure). Consider a chain structure with 5 nodes $\tilde{X}_1 \to \cdots \to \tilde{X}_5$, and the following graphs with 5 nodes:

1. 5 edges: $\tilde{X}_1 \to \cdots \to \tilde{X}_5$, with an additional $\tilde{X}_3 \to \tilde{X}_5$.
2. 5 edges: $\tilde{X}_1 \to \cdots \to \tilde{X}_5$, with an additional $\tilde{X}_2 \to \tilde{X}_4$.
3. 5 edges: $\tilde{X}_1 \to \cdots \to \tilde{X}_5$, with an additional $\tilde{X}_1 \to \tilde{X}_3$.
4. 6 edges: $\tilde{X}_1 \to \cdots \to \tilde{X}_5$, with additional $\tilde{X}_1 \to \tilde{X}_3$ and $\tilde{X}_3 \to \tilde{X}_5$.

For these five non-isomorphic graphs, with two equivalent permutations of each (swap the labeling of $\tilde{X}_4$ and $\tilde{X}_5$) - these 10 graphs form an equivalence class. One might be curious: what if a graph with one more edge, i.e.,

5. 7 edges: edges: $\tilde{X}_1 \to \cdots \to \tilde{X}_5$, with additional $\tilde{X}_1 \to \tilde{X}_3$, $\tilde{X}_2 \to \tilde{X}_4$ and $\tilde{X}_3 \to \tilde{X}_5$.

However, this graph is no longer in the equivalence class. For example, $\text{TIN}(X_3, X_{4,5}) = 1$ for the chain structure (and its equivalence class), while $\text{TIN}(X_3, X_{4,5}) = 2$ for this graph.

**Example 9** (An equivalence class with one unique graph). Consider a $\tilde{G}$ with 5 nodes and 7 edges: $\tilde{X}_1 \to \{\tilde{X}_2, \tilde{X}_3, \tilde{X}_4\}$, $\tilde{X}_2 \to \{\tilde{X}_3, \tilde{X}_5\}$, and $\tilde{X}_3 \to \{\tilde{X}_4, \tilde{X}_5\}$: surprisingly, its equivalence class contains only one graph, itself. I.e., by TIN conditions this structure should be uniquely recovered.

With the equivalence class, the identifiability result could be improved, and constrained O-ICA may be further applied to identify a final graph. It would be an interesting future work to characterize the "equivalence class" w.r.t. TIN, and then design an algorithm to identify it.

## F.3 More than Dimension of $\Omega_{\mathbf{Z};\mathbf{Y}}$: Parameters

Currently we only care about the *dimension* of the independent subspace $\Omega_{\mathbf{Z};\mathbf{Y}}$, but not the exact *parameters*. If we have obtained exactly the $\Omega_{\mathbf{Z};\mathbf{Y}}$, we could write its basis matrix $M_{\Omega_{\mathbf{Z};\mathbf{Y}}}$ in shape $|\mathbf{Y}| \times \dim(\Omega_{\mathbf{Z};\mathbf{Y}})$, with each column vector being a basis. Then, the subspace spanned by row vectors of $\mathbf{B}_{\mathbf{Y},\text{Anc}(\mathbf{Z})}$, which reflects edge parameters, is exactly the left nullspace of $M_{\Omega_{\mathbf{Z};\mathbf{Y}}}$.

The degeneration of $\omega$ we discussed in Appendix F.3 and Theorem 13 is actually a special case of recovering information from $\Omega_{\mathbf{Z};\mathbf{Y}}$ parameters. For edge parameters, it means that rank of $\mathbf{B}_{\mathbf{Y},\text{Anc}(\mathbf{Z})}$ will drop one if deleting the respective degenerated columns in $\mathbf{Y}$. More general exploitation of $\Omega_{\mathbf{Z};\mathbf{Y}}$ parameters is an interesting future work.

## F.4 More Possible Solutions for Estimation of $\Omega_{\mathbf{Z};\mathbf{Y}}$

In §5 we propose four methods to estimate $\Omega_{\mathbf{Z};\mathbf{Y}}$: tackling down to subsets of $\mathbf{Y}$ (§5.1), constrained independent subspace analysis (ISA) (§5.2), stacked cumulants' ranks stopped increasing (§5.3), and TIN in two steps: solve equations, and then test for independence (§5.4). Generally, reliable

estimation of $\Omega_{\mathbf{Z};\mathbf{Y}}$ can be formulated as an orthogonal research problem, and we believe that there exists more solutions.

For example, if we only care about the dimension of $\Omega_{\mathbf{Z};\mathbf{Y}}$, the following heuristic method might help. The intuition is that, uniformly sample infinite many random points on the surface of a unit sphere (centered on origin point) at $\mathbb{R}^n$, denote $d^{(k)}$ the average distance from these points to a subspace in $\mathbb{R}^n$ with dimension $k$ ($0 \leq k \leq n$). Then this average distance is monotonic over $k$: $d^{(k_1)} < d^{(k_2)}$ if and only if $k_1 > k_2$. For example, on an 2D circle, $d^{(0)} = 1$ (to center; radius), $d^{(1)} = 2/\pi$ (to diameter), and $d^{(2)} = 0$ (already on 2D); on a 3D sphere surface, $d^{(0)} = 1$ (to center; radius), $d^{(1)} = \pi/4$ (to diameter), $d^{(2)} = 1/2$ (to diameter plane), and $d^{(3)} = 0$ (already on 3D).

If we assume the independence tests return a bool value (independent or not), then this method will not help, because generally, the measure of $\Omega_{\mathbf{Z};\mathbf{Y}}$ relative to $\mathbb{R}^{|\mathbf{Y}|}$ is always zero. However, if we assume that, for a unit vector $\omega \in \mathbb{R}^{|\mathbf{Y}|}$, there exists a monotonic relationship between the independence strength of $\mathrm{Ind}(\omega^\mathsf{T}\mathbf{Y}; \mathbf{Z})$ (e.g., mutual information) and the distance to the subspace $\mathrm{dist}(\omega; \Omega_{\mathbf{Z};\mathbf{Y}})$, then we could have a non-parametric method to recover $\tilde{G}$: for each variable $X_i$, uniformly sample many $\{\omega_l\}_{l=1,\cdots}$ from $\mathbb{R}^{(n-1)}$ and calculate the average independence $\mathrm{avg}_l\, \mathrm{Ind}(\omega_l^\mathsf{T}[\mathbf{X} \backslash X_i]; X_i)$, then sort $X_i$ by their respective average independence (i.e., dimensions of there respective $\Omega_{\mathbf{Z};\mathbf{Y}}$) to get an estimation of the group ordering.

## F.5 What if Causal Sufficiency is Not Satisfied in $\tilde{G}$?

In this paper we assumed causal sufficiency relative to $\tilde{\mathbf{X}}$. Though it is reasonable to assume causal sufficiency in this context (which, to the best of our knowledge, is indeed a common assumption in the current literature of causal discovery with measurement error), this assumption itself, is a strong one and is not testable. Thus, it would be interesting to investigate the case where causal sufficiency is violated (in a sense of "latents of latents"): Will TIN-based method still output a correct ordering? If not, by which correction rules or algorithm relaxations can the identifiability be still partially preserved? We leave this as an interesting future work. For now, we try to provide some hints from examples (where we still use the Lemma 1-based method in this paper):

**Example 10** (A still (partially) identifiable case). Consider a chain structure $\tilde{X}_1 \to \tilde{X}_2 \to \cdots \to \tilde{X}_n$ (or similarly, a fully connected DAG) with a common hidden confounder $\tilde{L}$ pointing to them all: $\tilde{L} \to \{\tilde{X}_i\}_{i=1}^n$. If $\tilde{L}$ is not measured and only measurements $\mathbf{X} = \{X_1, \cdots, X_n\}$ are available, we have now: $\mathrm{ord}(X_1) = \mathrm{TIN}(X_1, \mathbf{X} \backslash X_1) = 2$, $\mathrm{ord}(X_2) = 3$, $\cdots$, $\mathrm{ord}(X_{n-3}) = n - 2$, and $\mathrm{ord}(X_{n-2}) = \mathrm{ord}(X_{n-1}) = \mathrm{ord}(X_n) = n - 1$. We shall see that: 1) The causal ordering of all but the last 3 variables is identifiable. While without $\tilde{L}$ (our previous result), this identifiability result is all but the last 2 variables (see Example 6), and 2) the existence of hidden (root) confounder(s) will also be reported, since there is no root (with $\mathrm{ord} = 1$) found across measurements $\mathbf{X}$.

**Example 11** (A no-longer identifiable case). Consider a simple fork $\tilde{X}_2 \leftarrow \tilde{X}_1 \to \tilde{X}_3$, with a hidden confounder $\tilde{L}$: $\tilde{L} \to \tilde{X}_1$ and $\tilde{L} \to \tilde{X}_2$. Then, $\mathrm{ord}(X_1) = \mathrm{TIN}(X_1, \mathbf{X} \backslash X_1) = 2$, $\mathrm{ord}(X_2) = 1$, and $\mathrm{ord}(X_3) = 2$. Sorting by $\mathrm{ord}$, we have the group decomposition as $\{\tilde{X}_2\} \to \{\tilde{X}_1, \tilde{X}_3\}$, while this is incorrect: there exists directed edge(s) from later groups to earlier groups, $\tilde{X}_1 \to \tilde{X}_2$.

## F.6 What if some Measurements are Caused by Multiple Latent Variables?

In this paper, we consider the measurement error model, where each measurement is caused by only one latent variable. For GIN, it can generally handle the cases where measurements are caused by multiple latent variables, as long as each latent variable has enough pure indicators. Interestingly however, we find that this may also be relaxed for our case (where there are not enough pure indicators), and our TIN-based method may still work (in identifying the correct group ordering). See below for some simple examples:

Consider a 3-nodes chain structure $\tilde{A} \to \tilde{B} \to \tilde{C}$, and their respective measurements $A, B, C$. We have the ordered group decomposition $\{\tilde{A}\} \to \{\tilde{B}, \tilde{C}\}$, with $\mathrm{ord}$ being 1 and 2. Then, what if we add an edge from a latent variable to another measured variable? There are $3 \times 2 = 6$ ways of adding an edge. Surprisingly, among these 6 ways, there are 5 which preserves exactly the same TIN results over $A, B, C$. The only one difference is by adding $\tilde{C} \to A$, where $\mathrm{TIN}(A, BC) = 2$, instead of 1. It would be interesting to generalize this observation: What if more nodes? What if more edges?

## G  Implementation and Evaluation

### G.1  Implementation Details

In this section we provide the information required to reproduce our results reported in the main text. We also commit to making our implementations of TIN public.

**Simulation setup**   In simulation we consider specifically two cases: fully connected DAG (Figure 2c) and chain structure (Figure 2a), of which the ordered group decomposition are both $\tilde{X}_1 \to \cdots \to \tilde{X}_{n-2}, \tilde{X}_{n-1,n}$. We consider $\tilde{G}$ with the number of vertices $n = 3, \cdots, 10$. Edges weights (i.e., the nonzero entries of matrix $\mathbf{A}$) are drawn uniformly from $[-0.9, -0.5] \cup [0.5, 0.9]$. Exogenous noises $\tilde{\mathbf{E}}$ are sampled from uniform $\cup[0, 1]$ to the power of $c$, $c \sim \cup[5, 7]$, and measurement errors are sampled from Gaussian $\mathcal{N}(0, 1)$ to the power of $c$, $c \sim \cup[2, 4]$. Sample size is $5, 000$. Observations are generated by $X_i = \tilde{X}_i + E_i$. To show the effect of measurement error, we simulate with noise-to-signal ratio $\mathrm{NSR} := \mathrm{var}(E_i)/\mathrm{var}(\tilde{X}_i)$ in $\{0.5, 1, 2, 3, 4\}$. On each configuration (under a graph type, measurement noise scaling, and the number of vertices), 50 random graphs are generated for repeated experiments.

**PC**   We use the implementation from the `causal-learn` package[5]. Kernel-based conditional independence test [52] is used. For speed consideration, datasets are downsampled to $1, 000$ on PC runs. The significance level `alpha` is set to $0.05$. to Definition 7.

**GES**   We use the implementation from the `causal-learn` package[6]. The score used is `local-BIC-score` [37].

**Direct-LiNGAM and ICA-LiNGAM**   We use the implementation from the `lingam` package[7]. Note that for Direct-LiNGAM, actually the method used is based on pairwise likelihood ratios [20].

**CAMME-OICA**   We use the implementation from `LFOICA`[8] (Likelihood-Free Overcomplete ICA). It estimates the mixing matrix by first transforming random noise into components, and then mimic the mixing procedure from components to noise with MMD score as a metric.

Below we give details on implementations of TIN. Specifically,

**Independence test**   We use the `HSIC` (Hilbert-Schmidt independence criterion) test [14] with the implementation from `lingam` package [9]. The kernel width is set to $0.1$ times the standard deviation of the data samples. The significance level `alpha` of p-value is set to $0.05$. Note that when the noise-to-signal ratio is large (e.g. $> 3$), usually observed variables are already 'independent enough' (i.e., with p-value given by HSIC test on raw data samples already $> 0.05$). In this case, we use the difference of $\frac{1000 * \text{severity}}{\text{sample size}}$ between $\mathbf{Z}; \mathbf{Y}$ and $\mathbf{Z}; \omega^{\intercal}\mathbf{Y}$ to show how much independence is *'gained'* by linear transformation. The threshold for this criterion is set to $0.5$.

**TIN-ISA**   We implement the constrained ISA where the de-mixing matrix is masked to only update the lower-right $|\mathbf{Y}| \times |\mathbf{Y}|$ block $\mathbf{W_{YY}}$, with upper-left $|\mathbf{Z}| \times |\mathbf{Z}|$ block fixed as the identity and elsewhere fixed as zero. We follow [29] for the estimation of conditional score function. Independence between $\mathbf{Z}$ and $\omega^{\intercal}\mathbf{Y}$ for each row of $\mathbf{W_{YY}}$ is then tested by HSIC test, as is described aboce.

**TIN-rank**   Numerical rank of a 2D matrix is calculated by SVD (singular value decomposition), with tolerance $\epsilon$ set to $0.005$. Singular values below threshold $T$ are considered zero, where $T = \epsilon * \max(S) * \max(M, N)$. $S$ is all singular values, and $M, N$ are shape of the 2D matrix. According to $Theorem\ 5$, we use the rank where stacked 2D slices of cumulants stops increasing rank as the output of TIN.

---

[5]https://github.com/cmu-phil/causal-learn/blob/main/causallearn/search/ConstraintBased/PC.py
[6]https://github.com/cmu-phil/causal-learn/blob/main/causallearn/search/ScoreBased/GES.py
[7]https://github.com/cdt15/lingam
[8]https://github.com/dingchenwei/Likelihood-free_OICA
[9]https://github.com/cdt15/lingam/blob/master/lingam/hsic.py

**TIN-2steps**  To solve euqations system $\{\mathrm{cov}(f(\mathbf{Z}), \mathbf{Y})\omega = \mathbf{0}\}$, functions $f$ contain: $\mathbf{Z}$, $\mathbf{Z}^2$, $\mathbf{Z}^3$, $|\mathbf{Z}|$, $e^{\mathbf{Z}}$, $\log(|\mathbf{Z}|)$, $\sin(\mathbf{Z})$, $\cos(\mathbf{Z})$, $\mathrm{sigmoid}(\mathbf{Z})$, $\tanh(\mathbf{Z})$. Nullspace is calculated by SVD, while we do not set a hard threshold of singular value to determine its space (like TIN-rank). Instead, we test HSIC between $\mathbf{Z}$ and $\omega^{\intercal}\mathbf{Y}$ for each $\omega$ in the $|\mathbf{Y}| \times |\mathbf{Y}|$ unitary matrix $\mathbf{V}$, and count the number of independence achieved.

**TIN-subsets**  The core to find the existence of transformed independence is similar to TIN-2steps. Then, for the part of traversing over $\mathbf{Y}$'s subsets, "all $\mathbf{Y}'$ ..." and "exists a $\mathbf{Y}'$ ..." are characterized by $90\%$ and $10\%$ percentile of the independence statistics (e.g., p-value of HSIC test) respectively.

**Noise synthesis**  Edges weights (i.e., the nonzero entries of matrix $\mathbf{A}$) are drawn uniformly from $[-0.9, -0.5] \cup [0.5, 0.9]$. Exogenous noises $\tilde{\mathbf{E}}$ of the latent variables are sampled from uniform $\cup[0, 1]$ to the power of $c$, $c \sim \cup[5, 7]$, and measurement errors are sampled from Gaussian $\mathcal{N}(0, 1)$ to the power of $c$, $c \sim \cup[2, 4]$. Sample size is $5,000$. Below we show a synthetic dataset with $\tilde{G}$ being a fully connected DAG with $n = 7$, and the noise-to-signal ratio being 3 (Figures 6 and 7):

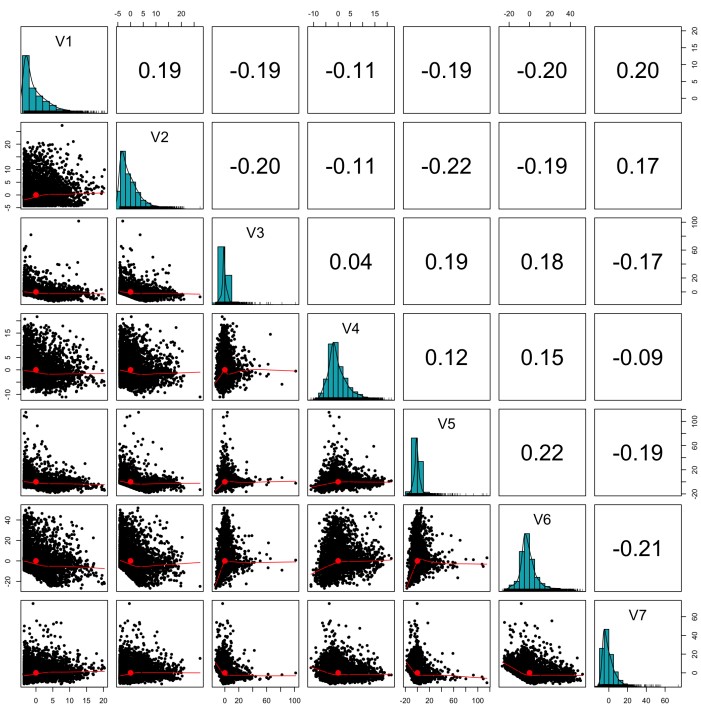

Figure 6: The scatter plot matrix for seven observed variables in $\mathbf{X}$. From the first column we could see that, though $\tilde{X}_1$ is a root variable in $\tilde{G}$, regressing neither of $\{X_2, \cdots, X_7\}$ on $X_1$ will make the regression residual independent to the regressor $X_1$, due to the presence of measurement error.

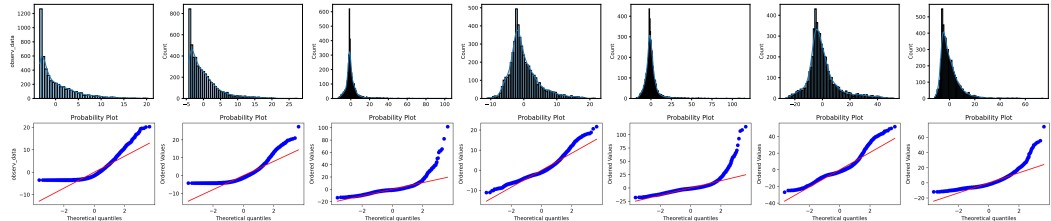

Figure 7: The histogram plot and Q-Q plot to Gaussian distribution of seven observed variables in $\mathbf{X}$. We could see the non-Gaussianity of data.

## G.2 Evaluation Details

To evaluate the output group ordering, we use *Kendall tau distance* [22] to the ground-truth (in range $[0, 1]$, the lower the better). Kendall tau distance counts the number of pairwise disagreements between two orderings. Specifically, for two variables $i, j$ and a grouped ordering $\tau$, we define:

$$\mathtt{cmp}(i, j, \tau) = \begin{cases} 1 & \text{in } \tau, i \text{ is in an earlier group than } j \\ 0 & \text{in } \tau, i \text{ is in a same group with } j \\ -1 & \text{in } \tau, i \text{ is in a later group than } j \end{cases} \tag{G.1}$$

Then, for the $n$ variables $[n]$ and two ordered group decompositions $\tau_1, \tau_2$ on them, the Kendall tau distance is defined as:

$$\mathtt{ktdist}(\tau_1, \tau_2) = \frac{2}{n(n-1)} \sum_{i,j \in [n],\ i<j} \mathrm{sign}(\mathtt{cmp}(i, j, \tau_1) \neq \mathtt{cmp}(i, j, \tau_2)) \tag{G.2}$$

For algorithms returning DAG/PDAG, its ordering is first extracted according to Definition 7. Specifically, for PDAG, a vertex's ancestors is defined as all vertices that has mixed paths (no directed edges backward) to it.

For intuition, here we give some typical examples: if the true ordering is $\{\{X_1\}, \{X_2\}, \{X_3\}, \{X_4, X_5\}\}$, the following group orderings have the respective distances:

1. $\{\{X_5\}, \{X_4\}, \{X_3\}, \{X_2, X_1\}\}$: 1.0;
2. $\{\{X_5\}, \{X_4, X_3, X_2, X_1\}\}$: 1.0;
3. $\{\{X_1, X_2, X_3, X_4, X_5\}\}$: 0.9;
4. $\{\{X_2, X_4\}, \{X_1, X_3, X_5\}\}$: 0.8;
5. $\{\{X_1, X_4\}, \{X_2, X_3, X_5\}\}$: 0.7;
6. $\{\{X_1, X_2, X_4\}, \{X_3, X_5\}\}$: 0.6;
7. $\{\{X_2\}, \{X_1, X_4\}, \{X_3, X_5\}\}$: 0.5;
8. $\{\{X_1, X_3\}, \{X_2, X_4, X_5\}\}$: 0.4;
9. $\{\{X_1, X_2\}, \{X_3, X_4, X_5\}\}$: 0.3;
10. $\{\{X_1, X_2\}, \{X_3\}, \{X_4, X_5\}\}$: 0.1;

# H  Detailed Elaboration on Examples

To explain the TIN condition's definition, characterization, and graphical criteria, in this section we provide some step-by-step derivation of typical examples.

## H.1  Motivation Examples

We first give an example of GIN to show our motivation (§2):

**Example 12** (GIN on chain structure Figure 2a). Let $\mathbf{Z} := \{X_1\}, \mathbf{Y} := \{X_2, X_3, X_4, X_5\}$. Calculate $\mathrm{cov}(X_1, X_2) = \mathrm{cov}(\tilde{X}_1 + E_1, \tilde{X}_2 + E_2) = \mathrm{cov}(\tilde{X}_1, \tilde{X}_2) = \mathrm{cov}(\tilde{X}_1, a\tilde{X}_1 + \tilde{E}_2) = a\,\mathrm{var}(\tilde{X}_1)$. Similarly, we get the covariance matrix

$$\mathrm{cov}(\mathbf{Z}, \mathbf{Y}) = \begin{bmatrix} a\,\mathrm{var}(\tilde{X}_1) & ab\,\mathrm{var}(\tilde{X}_1) & abc\,\mathrm{var}(\tilde{X}_1) & abcd\,\mathrm{var}(\tilde{X}_1) \end{bmatrix} \tag{H.1}$$

Solve the linear homogeneous equations $\mathrm{cov}(\mathbf{Z}, \mathbf{Y})\omega = \mathbf{0}$, we have $\omega = [-bx - bcy - bcdz, x, y, z]^\mathsf{T}$, $x, y, z \in \mathbb{R}$. Plug $\omega$ into $\omega^\mathsf{T}\mathbf{Y}$:

$$\omega^\mathsf{T}\mathbf{Y} = (-b\!\!\!/x - bc\!\!\!/y - bcd\!\!\!/z)(\tilde{X\!\!\!/}_2 + E_2) + x(b\tilde{X\!\!\!/}_2 + \tilde{E}_3 + E_3)$$
$$+ y(c(b\tilde{X\!\!\!/}_2 + \tilde{E}_3) + \tilde{E}_4 + E_4) + z(d(c(b\tilde{X\!\!\!/}_2 + \tilde{E}_3) + \tilde{E}_4) + \tilde{E}_5 + E_5)$$
$$= \text{does not contain } \{\tilde{E}_1, \tilde{E}_2, E_1\}, \text{ thus } \omega^\mathsf{T}\mathbf{Y} \perp\!\!\!\perp X_1, \text{ by the Darmois–Skitovich theorem [21].} \tag{H.2}$$

By above we have $\mathrm{GIN}(\{X_1\}, \{X_2, \cdots, X_n\})$ satisfied.

## H.2 Examples of Ordered Group Decomposition

In Definition 7 we define the ordered group decomposition of a graph. Actually there is slight difference between Definition 7 and Definition 2 in [54]. According to Definition 7, when the graph has only one subroots at each step, the two definitions are the same. However, when there are multiple subroots at some step, Definition 7 takes all such subroots (and their pure leaf children) as a new group, while in [54], each group has one and only one non-leaf node, and thus only one subroot (and its pure leaf children) is taken (and removed from graph) as a new group, which yields multiple ordered group decompositions. Here we only return one ordered group decomposition mainly for simplicity. To obtain the multiple ordered group decompositions defined in [54] from our result, we could do the following: for each pair of $X_i, X_j$ in a new group with multiple variables, test TIN with $\mathbf{Z}$ being $X_i$ and $\mathbf{Y}$ being $X_j$ and all variables in the previous groups, to see whether $X_i, X_j$ are in a same *cluster*. Below we give an example to show this slight difference between two definitions:

**Example 13** (Ordered group decomposition with multiple subroots)**.** Consider the graph $D \leftarrow A \to C \leftarrow B$. The ordered group decomposition we defined in Definition 7 is $\{A, B, D\} \to \{C\}$, while Definition 2 in [54] will give two ordered group decompositions: $\{A, D\} \to \{B, C\}$, and $\{B\} \to \{A, C, D\}$.

## H.3 A Concrete Example of Using TIN on Fully Connected DAG

Consider a fully connected DAG (Figure 2c) with 4 variables $\{X_1, X_2, X_3, X_4\}^\mathsf{T}$ (suppose we have access directly to the measurement-error-free variables and directly test TIN over them), the edges parameters are:

$$\mathbf{A} = \begin{bmatrix} 0 & 0 & 0 & 0 \\ a & 0 & 0 & 0 \\ b & d & 0 & 0 \\ c & e & f & 0 \end{bmatrix} ; \mathbf{B} = \begin{bmatrix} 1 & 0 & 0 & 0 \\ a & 1 & 0 & 0 \\ ad+b & d & 1 & 0 \\ a(df+e)+bf+c & df+e & f & 1 \end{bmatrix} \tag{H.3}$$

**1)** Let $\mathbf{Z} := \{X_1\}, \mathbf{Y} := \{X_2, X_3, X_4\}$, we have the independent subspace:

$$\Omega_{\mathbf{Z};\mathbf{Y}} = \mathrm{null}(\mathbf{B}_{\mathbf{Y},\mathrm{Anc}(Z)}^\mathsf{T}) = \mathrm{null}\left([a \quad ad+b \quad a(df+e)+bf+c]\right), \text{ dimension=2}. \tag{H.4}$$

Two basis of $\Omega_{\mathbf{Z};\mathbf{Y}}$ are:

$$\begin{aligned} \omega_1 &= \left[-\tfrac{b+ad}{a} \quad 1 \quad 0\right]^\mathsf{T}, \\ \omega_2 &= \left[-\tfrac{c+ae+bf+adf}{a} \quad 0 \quad 1\right]^\mathsf{T} \end{aligned} \tag{H.5}$$

For any $\omega = k_1\omega_1 + k_2\omega_2, k_1, k_2 \in \mathbb{R}$, $\omega^\mathsf{T}\mathbf{Y}$ does not contain noise $E_1$, so $\omega^\mathsf{T}\mathbf{Y} \perp\!\!\!\perp \mathbf{Z}$. We have $\mathrm{TIN}(\mathbf{Z}, \mathbf{Y}) = |\mathbf{Y}| - \dim(\Omega_{\mathbf{Z};\mathbf{Y}}) = 3 - 2 = 1$. Graphically, the minimum vertex cut from $\mathrm{Anc}(\{X_1\}) = \{X_1\}$ to $\{X_2, X_3, X_4\}$ is $\{X_1\}$, with size 1. And, $|\mathrm{Anc}(X_1)| = 1$.

**2)** Let $\mathbf{Z} := \{X_2\}, \mathbf{Y} := \{X_1, X_3, X_4\}$, we have the independence subspace:

$$\Omega_{\mathbf{Z};\mathbf{Y}} = \mathrm{null}(\mathbf{B}_{\mathbf{Y},\mathrm{Anc}(Z)}^\mathsf{T}) = \mathrm{null}\left(\begin{bmatrix} 1 & ad+b & a(df+e)+bf+c \\ 0 & d & df+e \end{bmatrix}\right), \text{ dimension=1}. \tag{H.6}$$

One basis of $\Omega_{\mathbf{Z};\mathbf{Y}}$ is:

$$\omega_1 = \left[\tfrac{cd-be}{d} \quad \tfrac{df+e}{d} \quad -1\right]^\mathsf{T} \tag{H.7}$$

For any $\omega = k_1\omega_1, k_1 \in \mathbb{R}$, $\omega^\mathsf{T}\mathbf{Y}$ does not contain noise $E_1, E_2$, so $\omega^\mathsf{T}\mathbf{Y} \perp\!\!\!\perp \mathbf{Z}$. We have $\mathrm{TIN}(\mathbf{Z}, \mathbf{Y}) = |\mathbf{Y}| - \dim(\Omega_{\mathbf{Z};\mathbf{Y}}) = 3 - 1 = 2$. Graphically, the minimum vertex cut from $\mathrm{Anc}(\{X_2\}) = \{X_1, X_2\}$ to $\{X_1, X_3, X_4\}$ is $\{X_1, X_2\}$, with size 2. And, $|\mathrm{Anc}(X_2)| = 2$.

**3)** Let $\mathbf{Z} := \{X_3\}, \mathbf{Y} := \{X_1, X_2, X_4\}$, we have the independence subspace:

$$\Omega_{\mathbf{Z};\mathbf{Y}} = \mathrm{null}(\mathbf{B}_{\mathbf{Y},\mathrm{Anc}(Z)}^\mathsf{T}) = \mathrm{null}\left(\begin{bmatrix} 1 & a & a(df+e)+bf+c \\ 0 & 1 & df+e \\ 0 & 0 & f \end{bmatrix}\right), \text{ dimension=0}. \tag{H.8}$$

$\mathbf{B}_{\mathbf{Y},\mathrm{Anc}(Z)}^\mathsf{T}$ is full column rank, so that there exists no non-zero $\omega$ s.t. $\omega^\mathsf{T}\mathbf{Y} \perp\!\!\!\perp \mathbf{Z}$, i.e., $\Omega_{\mathbf{Z};\mathbf{Y}}$ contains only origin point $\mathbb{R}_\mathbf{0}$, with dimension 0. We have $\mathrm{TIN}(\mathbf{Z}, \mathbf{Y}) = |\mathbf{Y}| - \dim(\Omega_{\mathbf{Z};\mathbf{Y}}) = 3 - 0 = 3$.

Graphically, the minimum vertex cut from $\mathrm{Anc}(\{X_3\}) = \{X_1, X_2, X_3\}$ to $\{X_1, X_2, X_4\}$ is $\{X_1, X_2, X_3\}$ or $\{X_1, X_2, X_4\}$, with size 3. And, $|\mathrm{Anc}(X_3)| = 3$.

**4)** Let $\mathbf{Z} := \{X_4\}, \mathbf{Y} := \{X_1, X_2, X_3\}$, we have the independence subspace:

$$\Omega_{\mathbf{Z};\mathbf{Y}} = \mathrm{null}(\mathbf{B}_{\mathbf{Y},\mathrm{Anc}(Z)}^{\mathsf{T}}) = \mathrm{null}\left(\begin{bmatrix} 1 & a & ad+b \\ 0 & 1 & d \\ 0 & 0 & 1 \\ 0 & 0 & 0 \end{bmatrix}\right), \text{ dimension=0.} \tag{H.9}$$

$B_{\mathbf{Y},\mathrm{Anc}(Z)}^{\mathsf{T}}$ is full column rank, so that there exists no non-zero $\omega$ s.t. $\omega^{\mathsf{T}}\mathbf{Y} \perp\!\!\!\perp \mathbf{Z}$, i.e., $\Omega_{\mathbf{Z};\mathbf{Y}}$ contains only origin point $\mathbb{R}_0$, with dimension 0. We have $\mathrm{TIN}(\mathbf{Z}, \mathbf{Y}) = |\mathbf{Y}| - \dim(\Omega_{\mathbf{Z};\mathbf{Y}}) = 3 - 0 = 3$. Graphically, the minimum vertex cut from $\mathrm{Anc}(\{X_4\}) = \{X_1, X_2, X_3, X_4\}$ to $\{X_1, X_2, X_3\}$ is $\{X_1, X_2, X_3\}$, with size 3. And, $|\mathrm{Anc}(X_4)| = 4$. Since $X_4$ is a leaf node, $\mathrm{TIN}(\mathbf{Z}, \mathbf{Y}) = 4 - 1 = 3$.

By above, we obtain the ordered group decomposition $\{\{X_1\}, \{X_2\}, \{X_3, X_4\}\}$.

## H.4 A Concrete Example of Using TIN on the Chain Structure

To align with Appendix H.3, here we consider a chain structure with 4 variables $\{X_1, X_2, X_3, X_4\}^{\mathsf{T}}$. The edges parameters are:

$$\mathbf{A} = \begin{bmatrix} 0 & 0 & 0 & 0 \\ a & 0 & 0 & 0 \\ 0 & b & 0 & 0 \\ 0 & 0 & c & 0 \end{bmatrix} ; \mathbf{B} = \begin{bmatrix} 1 & 0 & 0 & 0 \\ a & 1 & 0 & 0 \\ ab & b & 1 & 0 \\ abc & bc & c & 1 \end{bmatrix} \tag{H.10}$$

**1)** Let $\mathbf{Z} := \{X_1\}, \mathbf{Y} := \{X_2, X_3, X_4\}$, we have the independent subspace:

$$\Omega_{\mathbf{Z};\mathbf{Y}} = \mathrm{null}(\mathbf{B}_{\mathbf{Y},\mathrm{Anc}(Z)}^{\mathsf{T}}) = \mathrm{null}\left(\begin{bmatrix} a & ab & abc \end{bmatrix}\right), \text{ dimension=2.} \tag{H.11}$$

Two basis of $\Omega_{\mathbf{Z};\mathbf{Y}}$ are:

$$\omega_1 = \begin{bmatrix} b & -1 & 0 \end{bmatrix}^{\mathsf{T}}, \tag{H.12}$$
$$\omega_2 = \begin{bmatrix} bc & 0 & 1 \end{bmatrix}^{\mathsf{T}}$$

We have $\mathrm{TIN}(\mathbf{Z}, \mathbf{Y}) = |\mathbf{Y}| - \dim(\Omega_{\mathbf{Z};\mathbf{Y}}) = 3 - 2 = 1$. Analysis is similar to that of Appendix H.3.

**2)** Let $\mathbf{Z} := \{X_2\}, \mathbf{Y} := \{X_1, X_3, X_4\}$, we have the independence subspace:

$$\Omega_{\mathbf{Z};\mathbf{Y}} = \mathrm{null}(\mathbf{B}_{\mathbf{Y},\mathrm{Anc}(Z)}^{\mathsf{T}}) = \mathrm{null}\left(\begin{bmatrix} 1 & ab & abc \\ 0 & b & bc \end{bmatrix}\right), \text{ dimension=1.} \tag{H.13}$$

One basis of $\Omega_{\mathbf{Z};\mathbf{Y}}$ is:

$$\omega_1 = \begin{bmatrix} 0 & c & -1 \end{bmatrix}^{\mathsf{T}} \tag{H.14}$$

We have $\mathrm{TIN}(\mathbf{Z}, \mathbf{Y}) = |\mathbf{Y}| - \dim(\Omega_{\mathbf{Z};\mathbf{Y}}) = 3 - 1 = 2$. Analysis is similar to that of Appendix H.3.

**3)** Let $\mathbf{Z} := \{X_3\}, \mathbf{Y} := \{X_1, X_2, X_4\}$, we have the independence subspace:

$$\Omega_{\mathbf{Z};\mathbf{Y}} = \mathrm{null}(\mathbf{B}_{\mathbf{Y},\mathrm{Anc}(Z)}^{\mathsf{T}}) = \mathrm{null}\left(\begin{bmatrix} 1 & a & abc \\ 0 & 1 & bc \\ 0 & 0 & c \end{bmatrix}\right), \text{ dimension=0.} \tag{H.15}$$

We have $\mathrm{TIN}(\mathbf{Z}, \mathbf{Y}) = |\mathbf{Y}| - \dim(\Omega_{\mathbf{Z};\mathbf{Y}}) = 3 - 0 = 3$. Analysis is similar to that of Appendix H.3.

**4)** Let $\mathbf{Z} := \{X_4\}, \mathbf{Y} := \{X_1, X_2, X_3\}$, we have the independence subspace:

$$\Omega_{\mathbf{Z};\mathbf{Y}} = \mathrm{null}(\mathbf{B}_{\mathbf{Y},\mathrm{Anc}(Z)}^{\mathsf{T}}) = \mathrm{null}\left(\begin{bmatrix} 1 & a & ab \\ 0 & 1 & b \\ 0 & 0 & 1 \\ 0 & 0 & 0 \end{bmatrix}\right), \text{ dimension=0.} \tag{H.16}$$

We have $\mathrm{TIN}(\mathbf{Z}, \mathbf{Y}) = |\mathbf{Y}| - \dim(\Omega_{\mathbf{Z};\mathbf{Y}}) = 3 - 0 = 3$. Analysis is similar to that of Appendix H.3.

By above, we get the group ordering $\{\{X_1\}, \{X_2\}, \{X_3, X_4\}\}$. Recall the chain structure with triangular head example, we could distinguish it from the chain structure with ordered group decomposition $\{\{X_1\}, \{X_2, X_3, X_4\}\}$.

## H.5 Experiments on another Real-world Dataset: Teacher Burnout

Except for Sach's dataset discussed in §6.2, we also conduct experiments on another real-world dataset, Teacher Burnout [4]. It is from a sociology survey conducted by Barbara Byrne to investigate the influence on the three facets (emotional exhaustion, depersonalization, and personal accomplishment) of full-time elementary teachers' burnout from factors including: organizational (role ambiguity, role conflict, work overload, classroom climate, decision making, superior support, peer support) and personality (self-esteem, external locus of control) variables. Please see chapter six of [4] for more details about the dataset (Page 161), and the structure (Page 191).

While in the raw dataset, each (latent/target) variable has more than one measurements/indicators, in this experiment we pick only one measurement for each to demonstrate the measurement error situation. Specifically, we pick ten variables (according to the ten latent variables in Figure 6.10 of [4]): $RA_1$, $RC_1$, $CC_1$, $DM_1$, $SS_1$, $SE_1$, $ELC_1$, $EE_1$, $DP_1$, and $PA_1$. Though for a thorough study of the dataset, one could try other combinations of measurements, e.g., $RA_2$, $RC_3$, ..., in this experiment we only study one combination as above for illustration.

According to Figure 6.10 of [4], the ordered group decomposition of the ground-truth underlying causal graph is {RA, RC, CC, DM, SS} → {SE, ELC} → {EE} → {DP, PA}. Result given by TIN-subsets is {DM, SE} → {CC, SS} → {RA, ELC} → {RC, EE, DP} → {PA} (with the one-over-others TIN being $5, 6, 7, 8, 9$ respectively). This is similar to Byrne's conclusion (the true ordering) according to the domain knowledge. For example, 1) the three facets of burnout (emotional exhaustion, depersonalization, and personal accomplishment) are caused by other factors and are at the end of the ordered groups, 2) decision making, classroom climates and superior support are root causes (in the first two groups), and 3) self-esteem and role conflict influences external locus of control. Interestingly, some of the ordering inconsistent with the ground-truth might also be reasonable to some extent. For example, 1) self-esteem is among the first group (though should be in the second), maybe because it is "root-like": it is only caused by two root causes and causes another four variables, 2) decision making and superior support are in the first and second groups respectively (though should both be in the first, as two root causes for self-esteem), maybe because there exists difference in their impact on others, and 3) role ambiguity is in the third group (though should be in the first), maybe because that though it is a root, it has only one child, personal accomplishment, which is a leaf node in the graph; the same may applies to role conflict: though being a root, it is even among the second to last group, which is also echoed by other methods.

Here is an overview of the distance scores and ordered groups returned by all methods:

1. TIN-2steps: 0.49, {CC, SE, ELC} → {RA, DM, SS} → {EE, PA} → {RC, DP}.
2. TIN-subsets: 0.47, {DM, SE} → {CC, SS} → {RA, ELC} → {RC, EE, DP} → {PA}.
3. TIN-ISA: 0.56, {RA, DM, SE} → {SS, ELC} → {RC, CC, EE, DP, PA}.
4. TIN-rank: 0.60, {CC, SE} → {RA, RC, DM, ELC, DP} → {SS, EE, PA}.
5. ICA-LiNGAM: 0.56, {CC} → {SE} → {ELC} → {RA} → {SS} → {EE} → {RC, PA} → {DM, DP}.
6. Direct-LiNGAM: 0.73, {DP} → {SE} → {SS} → {RA} → {RC, PA} → {CC} → {EE} → {DM, ELC}.
7. CAMME-OICA: 0.78, {CC, SS, EE, DP, PA} → {RA, RC, DM, SE, ELC}.
8. PC: 0.73, {RA, RC, CC, DM, SS, SE, ELC, EE, DP, PA}.
9. GES: 0.73, {CC, DM, EE, DP} → {RA} → {SS, SE, PA} → {RC, ELC}.
10. NOTEARS: 0.82, {CC, SE, ELC, PA} → {RA, EE, DP} → {RC} → {DM, SS}.
11. SCORE: 0.76, {SE} → {SS} → {RA, CC, DM, ELC, EE, DP, PA} → {RC}.