# OpenReview forum: "Independence Testing-Based Approach to Causal Discovery under Measurement Error and Linear Non-Gaussian Models"
_NeurIPS.cc/2022/Conference — NeurIPS 2022 Accept_

### Official Review · Reviewer_Pi4t · 2022-07-11

**Rating:** 6
**Confidence:** 4
**Soundness:** 3 good
**Presentation:** 3 good
**Contribution:** 3 good

**Summary:**

The authors propose a generalization of the existing causal discovery approaches for learning the structure between latent variables in a parametric linear additive non-Gaussian structural causal model - under the assumption that each latent node has exactly one measurement node that is a child.

**Questions:**

Thank you for your submission. I will be happy to increase my score after clarifications of the below.

"necessarily being one, all results in this paper still hold"
I guess this depends on the downstreat objective. This might not always be a valid assumption.

An example/insight for Prop. 1 would be useful.

"the GIN condition can be readily used to fully identify the structure of G, which is already a breakthrough over existing methods"
Can you elaborate on this? This was not identified by the previous authors? They only learned the partial graph? This statement is missing some context.

Could you elaborate a bit on the recursion idea that is being hinted at for the GIN paper and that I believe will be used here as well? If the source node is discovered, one can get rid of its effect by conditioning on it. But here we only have access to its descendant. So we cannot condition. How can one recurse after the source is found?

line172: "while this is impossible on \tilde{G}"
This and other impossibility claims seem hard to check. Could you give some intuition on which mathematical tool can be used to verify this impossibility claim?

Ancestors in Theorem 1 probably means exogenous ancestors.

Assumption 1 seems reasonable as an alternative formulation of faithfulness.

The method requires precise measurement model, which is a disadvantage, correct? If an observed node is caused by two latent variables, this will not work. Please verify.

Could you please explain the proof of Theorem 2? I am a little concerned with the mapping to the max-flow min-cut formulation. Proof connects through B matrix, which is weighted - each edge has a weight potentially different from 1. Max-flow min-cut connection is valid for weighted or non-weighted. But using B matrix tells me that you are using the weighted version. But then we are making judgements about the rank of matrices which is typically not about the amount of max-flow determined by weighted edges.

Specifically the statement:
"the maximum amount of non-intersecting paths from source to sink is equal to the size of the minimum vertex cut from source to sink."
Maybe I am not familiar with this version because if flow is on an edge weighted graph, I thought cut should also be counted as edge weighted.

Theorem 3 should require some sort of guarantee that the measurement does not lose information. Can you formalize this?

My main concern is about finding the correct value \omega for conducting the transformed independence tests. I also don't see how this statement follows from Theorem 4:
"This transforms the task of estimating the dimension of \Omega(Z;Y) to a simpler one: counting size of the subsets Y0."
One still needs to search for existence of a w in these subsets.

Section 5 in general is too short and is very hard to follow without any explanation. How do we find w_YY in (11)?

line 367:" then check whether all solution"
How can we efficiently check all solutions if there are uncountably many? Please explain.


**Limitations:**

More like future work than the limitations and critique. Could be improved.

**Strengths And Weaknesses:**

+ The exposition is very clear. Authors closely follow several examples and do it well.

+ An important problem with some recent results. This work seems to generalize these for another special case of measurement graphs.

- Presentation gets overly succinct in Section 5.

- Some concerns with the theory.

- Some parts of the algorithmic approach is unclear.

- Emoji-based writing in the intro makes it hard to parse the text/the story.

---

> ### Author Response · Authors · 2022-08-02
> **Response to Reviewer Pi4t**
>
> We appreciate the reviewer's feedback. Thanks for your careful reading!
>
> We roughly divide your questions into three categories. Due to the limit on the number of characters, we will put three separate comments in this thread. Please see below for our response.

---

> > ### Author Response · Authors · 2022-08-02
> > **Response to Reviewer Pi4t - Part 3/3 - Details**
> >
> > **(Q3s)** The reviewer wonders for more clarifications to examples/details. Specifically,
> >
> > ---
> >
> > > "An example/insight for Prop. 1 would be useful."
> >
> > **R:** Thanks for the suggestion!  Consider the underlying graph $\tilde{X}\rightarrow \tilde{A} \rightarrow \tilde{B}\leftarrow \tilde{Y}$:
> > + $\tilde{X}\perp\tilde{Y}|\{\tilde{A} ,\tilde{B}\}$, but $X\not\perp Y | \{A, B\}$: generally, d-separation on $\tilde{G}$ is lost on $G$, since we only have descendants.
> > + $\tilde{X}\not\perp\tilde{Y}|\{\tilde{B}\}$, and also $X\not\perp Y | \{B\}$: if d-connected on $\tilde{G}$, then must also d-connected on $G$.
> > + $\tilde{X}\perp\tilde{Y}|\{\tilde{A}\}$, and also $X\perp Y | \{A\}$ (and $\tilde{X}\perp\tilde{Y}$, $X\perp Y$): _rare_ d-separation. The only way to preserve d-separation on $\tilde{G}$ is by marginally $\perp$. See Appendix A.1 for the proof.
> >
> > > "L172: how to verify 'this is impossible on $\tilde{G}$'?"
> >
> > **R:** Please refer to Example 5 (line 239) and the orange block in the left matrix, where we tried to provide an explanation. In short, Theorem 1 characterizes the independence subspace $\Omega_{\mathbf{Z;Y}}$ as the nullspace of some $\mathbf{B}$ block. Then, to verify such impossibility claims (i.e., $\Omega_{\mathbf{Z;Y}}=\mathbf{0}$), it suffices to show the (algebraic) full row rank of the respective $\mathbf{B}$ block.
> >
> > > "Ancestors in Theorem 1 probably means exogenous ancestors."
> >
> > **R:** No. It is defined in line 194. We do not additionally index exogenous noises for each variable. In Theorem 1, it just denotes the (indices to) ancestral variables, which equals the nonzero column indices of the $\mathbf{B}$ rows block.
> >
> > > "Assumption 1 seems reasonable as an alternative formulation of faithfulness."
> >
> > **R:** Yes, in the sense of "no parameters coupling".
> >
> > > "How to find $\mathbf{W}_{\mathbf{YY}}$ in (11)?"
> >
> > **R:** As mentioned in Section 5.2, conduct the Independent Subspace Analysis where the de-mixing matrix is masked to only update the lower-right $\mathbf{Y}\times\mathbf{Y}$ block $\mathbf{W}_{\mathbf{YY}}$, with the upper-left $\mathbf{Z}\times\mathbf{Z}$ fixed as the identity and elsewhere fixed as zero. For the implementation details of ISA, we directly follow the original paper [6] (see Appendix G.1 and [our code](https://anonymous.4open.science/r/TIN/utils/ISA.py)).  This has been included in the revision (Appendix E.2).
> >
> > ---
> >
> > [1] Silva, Ricardo, et al. "Learning the Structure of Linear Latent Variable Models." _Journal of Machine Learning Research_ 7.2 (2006).
> >
> > [2] Spirtes, Peter L. "Calculation of entailed rank constraints in partially non-linear and cyclic models." _arXiv preprint arXiv:1309.7004_ (2013).
> >
> > [3] Kummerfeld, Erich, and Joseph Ramsey. "Causal clustering for 1-factor measurement models." _Proceedings of the 22nd ACM SIGKDD international conference on knowledge discovery and data mining_. 2016.
> >
> > [4] Shafer, Glenn, Alexander Kogan, and Peter Spirtes. "Generalization of the tetrad representation theorem." Rutgers University. Rutgers Center for Operations Research [RUTCOR], 1993.
> >
> > [5] Xie, Feng, et al. "Generalized independent noise condition for estimating latent variable causal graphs." _Advances in Neural Information Processing Systems_ 33 (2020): 14891-14902.
> >
> > [6] Theis, Fabian. "Towards a general independent subspace analysis." _Advances in Neural Information Processing Systems_ 19 (2006).

---

> > > ### Comment · Reviewer_Pi4t · 2022-08-08
> > > **response to rebuttal 2**
> > >
> > > Thank you for the explanations. I think it would be very helpful to add these/move some of these from appendix to the main paper for clarity.

---

> > > > ### Author Response · Authors · 2022-08-08
> > > > **We appreciate your suggestion.**
> > > >
> > > > We are working on improving the presentation of the paper accordingly (it involved iterations because the paper is a bit dense).  Thanks for your timely feedback!

---

> > ### Author Response · Authors · 2022-08-02
> > **Response to Reviewer Pi4t - Part 2/3 - Methodology**
> >
> > **(Q2s)** The reviewer wonders for more insights about the methodology part. Specifically,
> >
> > ---
> >
> > > "'If each latent variable has two measurements, GIN can fully identify the structure of $\tilde{G}$, which is already a breakthrough' - Can you elaborate on this? This was not identified by the previous authors?"
> >
> > **R:** Yes, previous methods only learned a partial graph (see e.g., BPC (Section 5 of [1]), extended-t-separation (Page 2 left of [2]), and FOFC (Section 4 of [3])). This is generally the case because these methods are based on the Tetrad constraint [4], and Tetrad makes use of only second-order statistics. However, GIN further exploited higher-order information (independence in the non-Gaussian case) and thus identifies more. We have incorporated an illustrating example to show how GIN differs from earlier methods. Please refer to Appendix B in the revision for details.
> >
> > > "Elaborate on the recursion idea for GIN and here as well? How can one recurse when the source cannot be conditioned on?"
> >
> > **R:** First, let us mention that the recursion is not used in our proposed TIN-based method. Instead, we just estimate _one-over-others-TIN_ over every single variable to identify the ordered group decomposition (Lemma 1). Nevertheless, as future work, it may be exploited to further recover the structure of $\tilde{G}$ (see Appendix F.2).
> >
> > As for the recursion idea in GIN (either in Section 4 of [5] or the intro Examples 1-3 of this paper), we agree that one cannot directly condition on or regress on the source node, since we only have access to its descendant. Instead, one can _"condition"_ on the source node by incorporating the corresponding measured variables into $\mathbf{Z}$ sets (termed as _"reference variables"_) to realize independent _"pseudo-residuals"_ (line 117) - just _"as if"_ the effect of the source node is removed for recursion.
> >
> > > "How does '...a simpler one...' statement follows from Theorem 4? One still needs to search for the existence of an $\omega$ in these subsets."
> >
> > **R:** First, we would like to mention that the two methods are theoretically equivalent. The only difference lies on the empirical level. Under finite samples, subsets tackling down allows a smaller combination set, a more flexible thresholding (in judging all/any), and probably less error by more times of independence tests.
> >
> > Consider this example: to identify the root $\tilde{X}_1$ from an $n$-nodes chain structure, i.e., $\operatorname{TIN}(X_1, \mathbf{X}\backslash X_1)=1$.
> >
> > + By directly estimating the dimension of $\Omega_{\mathbf{Z;Y}}$, we need to linearly combine all the rest $n-1$ variables - which is a large set and may already contain lots of random errors, and checks for independence for $n-2$ times (i.e., $n-2$ bases of $\Omega_{\mathbf{Z;Y}}$) - which is a small times of tests, and yields a high degree of freedom for each $\omega$. The TIN result may get error-prone, e.g., 0, or 2, 3, ...
> > + Equivalently, by tackling down to subsets of $\mathbf{Y}$, we need to try over all the 2-sized subsets of the rest $n-1$ variables - i) they are relatively small sets, and ii) we only need to check the _existence_ of the independent linear transformations of 2 variables on a singleton variable - both yielding easier (and more accurate) independence results. Moreover, such tests are needed for ${n-1 \choose 2}$ times over _all_ the 2-sized subsets. Though slower, this large number of tests reduces random error and enables a more flexible thresholding to judge quantifiers all/any (see Appendix G.1).
> >
> > > "How can we efficiently check all solutions if there are uncountably many?"
> >
> > **R:** We only need to check up to $|\mathbf{Y}|$ solutions. Here is a justification:
> >
> > Though the search space for $\omega$ is uncountably infinite, by the property of independence we have that 1) if $\omega^\intercal \mathbf{Y} \perp \mathbf{Z}$, then $(c\omega)^\intercal \mathbf{Y} \perp \mathbf{Z}$  for any $c\in\mathbb{R}$ (closed under scalar multiplication), and 2) if $\omega_1^\intercal \mathbf{Y} \perp \mathbf{Z}$ and $\omega_2^\intercal \mathbf{Y} \perp \mathbf{Z}$, then $(\omega_1+\omega_2)^\intercal \mathbf{Y} \perp \mathbf{Z}$ (closed under addition). Indeed, as we show in Definition 3, $\Omega_{\mathbf{Z;Y}}$ is a subspace. Thus we only need to find the $k$ orthogonal bases of $\Omega_{\mathbf{Z;Y}}$, where $k$ is the subspace dimension.
> >
> > Then, how to find such $k$ bases?  We are not searching directly from the whole $\mathbb{R}^{|\mathbf{Y}|}$. Instead, we greatly reduce the searching space to e.g., 1) $|\mathbf{Y}|$ orthogonal row vectors (TIN-ISA, see Section 5.2 and Appendix G.1 for explanation), and 2) the bases of the nullspace induced by zero-covariances/cumulants equations (TIN-2steps and TIN-rank, see Appendix E.1 and specifically equation (E.10) for explanation).

---

> > ### Author Response · Authors · 2022-08-02
> > **Response to Reviewer Pi4t - Part 1/3 - Theory**
> >
> > **(Q1s)** The reviewer wonders for more intuition about the theoretical part. Specifically,
> >
> > ---
> >
> > > "'Necessarily being one, all results in this paper still hold' - this might not always be a valid assumption?"
> >
> > **R:** For all results in this paper, it is indeed a valid statement, since we do not care about the scales of the latent variables $\tilde{X}_i$ - we may just always view $c_i\tilde{X}_i$ as the latent variables. You may view $\tilde{\mathbf{X}}$ and $\mathbf{X}$ altogether as variables generated by a single graph (where $\tilde{X}_i \rightarrow X_i$ are normal linear causal influences), and this statement follows from the graphical criteria (Theorem 2).
> >
> > Moreover, in general, yes, we agree with you that "it depends on the downstream objective". For instance, if someone just cares about the scales of $\tilde{X}_i$, then this statement is no longer valid. In this case, regarding the identification of $\tilde{G}$, the estimated graph structure of $\tilde{G}$ will be the same, and what changes is the estimated edge coefficients in $\tilde{G}$ (what we do not care about in this paper).
> >
> > > "The method requires a precise measurement model, which is a disadvantage, correct? If an observed node is caused by two latent variables, this will not work. Please verify."
> >
> > **R:** This is a great point, and we agree. Here we consider the measurement error model, where each measurement is caused by one latent variable. For GIN, it can generally handle the cases where measurements are caused by multiple latent variables, as long as each latent variable has enough pure indicators (Definition 1 of [5]). Interestingly, however, we found that this may also be relaxed for our case (where there are not enough pure indicators), and our TIN-based method may still work. See Appendix F.6 (may also F.5) for a detailed discussion.
> >
> > > "Could you please explain the proof of Theorem 2? How is max-flow min-cut formulated on an edge-weighted graph?"
> >
> > **R:** We are afraid that this might be a misunderstanding. The weighted $\mathbf{B}$ matrix is not directly mapped to the minimum vertex cut by the max-flow min-cut theorem. Instead, the proof sketch is formed in two steps: the first step is an algebraic combinatorics one, from the weighted $\mathbf{B}$ matrix to the non-intersecting paths (already unweighted), by the Lindström-Gessel-Viennot theorem; and the second step is a pure graph theory one, from the non-intersecting paths to the minimum vertex cut, by the max-flow min-cut theorem (vertex version, known as Menger's theorem). Please refer to Appendix A.3 and Theorem 6 for details.
> >
> > > "Theorem 3 should require some sort of guarantee that the measurement does not lose information?"
> >
> > **R:** Interestingly, the answer is no. We did not require additional guarantees about the information loss.
> >
> > It is worth mentioning that TIN is just one particular aspect of the statistical information of the variables. Clearly, some information is lost because of the contamination of measurement noise (e.g., the aspects of conditional independence, IN, and SEM are lost, as we show in lines 57, 92, and 105). However, the aspect of TIN remains the same among true and contaminated variables (exactly the fun point of TIN). See Appendix A.4 for the proof (either graphically or mathematically).

---

> > > ### Comment · Reviewer_Pi4t · 2022-08-08
> > > **Rebuttal responses**
> > >
> > > Thank you for the detailed explanations and for clarifying!
> > >
> > > Regarding the last point. I think there can be corner cases without assumptions to avoid them: For example if the measurement model always outputs a constant; or if it introduces some form of unfaithfulness the d-separation statements among the observed variables would not imply d-separation among the latent variables. That's what I had meant by the measurement should preserve information.

---

> > > > ### Author Response · Authors · 2022-08-08
> > > > **Thanks for sharing your intuition and your suggestion**
> > > >
> > > > Many thanks for sharing your insight!
> > > >
> > > > We completely agree with the statement that there can be corner cases without assumptions to avoid them and the intuition that the measurement should preserve information. In our study, the assumed "random measurement error model" (Eq. 1) and linear, non-Gaussian, acrylic model for the measurement-error-free variables (line 49) make this theorem hold true.  As you mentioned, in cases where our assumptions are violated, Theorem 3 might be false. We will add two sentences to emphasize this point.

---

> > > > > ### Comment · Reviewer_Pi4t · 2022-08-09
> > > > > **response to response**
> > > > >
> > > > > Yes, I think making the theorem statements self-contained by explicitly cross-referencing the assumptions they rely on will be very helpful. Thank you for your submission!

---

> ### Author Response · Authors · 2022-08-09
> **Could you please consider updating your score?**
>
> Dear Reviewer Pi4t,
>
> Once again, thanks a lot for reviewing our submission and for the further iterations!  We hope we have properly addressed your major concerns.  If that is the case, could you please consider increasing your score? (As you kindly mentioned in your comments, you would be happy to increase your score after those clarifications.)
>
> If you have any other concerns, could you let us know at your earliest convenience?  (The discussion involving authors will end in 4.5 hours.)  We will immediately respond to them.
>
> With best wishes,
> Authors of submission 202

---

### Official Review · Reviewer_G5xe · 2022-07-12

**Rating:** 7
**Confidence:** 3
**Soundness:** 3 good
**Presentation:** 3 good
**Contribution:** 3 good

**Summary:**

In this paper, the authors study causal graph recovery among unobserved target variables from observations made with   measurement error. The authors propose  the Transformed Independent  Noise (TIN) condition, which checks for independence between a specific linear   transformation of some measured variables and certain other measured variables. By  utilizing TIN, the ordered group decomposition of the causal model is identifiable.  Experimental results on both synthetic and real-world data demonstrate the effectiveness and reliability of the method.

**Questions:**


1. Is it possible to apply the method directly to causal structure recovery?

**Limitations:**

The paper can be strengthened with additional real-world examples and experimental results.


**Strengths And Weaknesses:**

Strengths:

 1. The paper is well written with good structures. Detailed illustrations and annotation make it easy to read.

  2. The methodology proposed by this paper and technical sections look solid and sound with technical details.

Weaknesses:

1. Experiments on more real-world datasets with measurement errors could make the paper stronger.

2. Experimental comparison with other methods such as [1] and [2] on ordered group decomposition should be added.

[1] Zheng, Xun, et al. "Dags with no tears: Continuous optimization for structure learning." Advances in Neural Information Processing Systems 31 (2018).

[2] Rolland, Paul, et al. "Score matching enables causal discovery of nonlinear additive noise models." arXiv preprint arXiv:2203.04413 (2022).

---

> ### Author Response · Authors · 2022-08-02
> **Response to Reviewer G5xe**
>
> We appreciate the reviewer's encouragement and helpful feedback. Please see below for our response.
>
> ---
>
> **(Q1)** The reviewer suggests experiments on more real-world datasets with measurement error.
>
> **R:** Thanks for the suggestion! We have conducted experiments on another real-world dataset, Teacher Burnout data [1]. In short, the result produced by our method is similar to the domain knowledge, and achieves the lowest (best) Kendall-tau distance to the ground-truth. Please see Appendix H.5 in the revision for a detailed analysis.
>
> ---
>
> **(Q2)** The reviewer suggests a comparison with other methods such as NOTEARS [2] and SCORE [3].
>
> **R:** In light of your suggestion, we have also compared with NOTEARS and SCORE. The performance of NOEARS is rather poor in this case. Interestingly, we found that SCORE is among the two strongest competitors (the other is ICA-LiNGAM). Please see Section 6 and Figure 5 in the revision for details.
>
> ---
>
> **(Q3)** The reviewer wonders whether it is possible to apply the method directly to causal structure recovery.
>
> **R:** In this paper, the answer is no. By directly applying the _one-over-others-TIN_ method, generally speaking, we cannot recover the causal structure, but only the ordered group decomposition - although it is already very informative.
>
> However, if we apply the proposed TIN condition over more general pairs of variables (not only one-over-others), the result may be more informative than just the ordered group decomposition. For example, though with the same ordered group decomposition, the chain structure and the fully connected DAG can actually be distinguished by TIN. Please see Section 7 and Appendix F.2 for details.
>
> Then, is the answer "yes" achievable? In the general case, no. The exact structure of $\tilde{G}$ cannot be recovered, since 1) as mentioned in line 310 and Definition 6, some variables are naturally unidentifiable under measurement error, and 2) different DAGs may produce completely the same TIN results, and thus TIN can identify them up to an "equivalence class" (please see Appendix F.2 for examples). As mentioned in line 1177, an alternative is to further apply O-ICA over the search space that is already greatly reduced by TIN.
>
> Thanks for this exciting question!
>
> ---
>
> [1] Byrne, Barbara M. "Structural equation modeling with Mplus: Basic concepts, applications, and programming." _routledge_, 2013.
>
> [2] Zheng, Xun, et al. "Dags with no tears: Continuous optimization for structure learning." _Advances in Neural Information Processing Systems_ 31 (2018).
>
> [3] Rolland, Paul, et al. "Score matching enables causal discovery of nonlinear additive noise models." _International Conference on Machine Learning_. PMLR, 2022.

---

### Official Review · Reviewer_ZM5r · 2022-07-24

**Rating:** 5
**Confidence:** 2
**Soundness:** 2 fair
**Presentation:** 3 good
**Contribution:** 3 good

**Summary:**

This paper proposes an algorithm to learn an ordered group decomposition of the causal graph over variables $\mathbf{X}$ which are not measured directly. In other words, each measured variable $\tilde{X}$ is a proxy of a target variables $X_i$ such that $X_i = \tilde{X}_i + E_i$, where $E_i$ is the measurement error.

The approach assumes the LiNGAM’s setting, I.e.:
1) acyclic model
2) causal sufficiency (no unmeasured confounders between the measured and also between the target variables),
3) linear functions, and
4) non-Gaussian error terms.

Further, it assumes a random measurement error model, where $X_i = \tilde{X}_i + E_i$  and  $E_i$  are additive errors assumed to be mutually independent and independent of $X_i$.

Specifically, the authors propose the Transformed Independent Noise (TIN) condition, which checks independence between a particular linear combination of some variables and others. This condition generalizes the current approaches IN and GIN and provides information about the graphical structure over the measured variables, even when there are measurement errors. The authors show that the ordered group decomposition of the causal model is identifiable in this setting.

The proposed techniques were evaluated both through simulations and an application to real data.



**Questions:**

No further questions.

**Limitations:**

See sections above.



**Strengths And Weaknesses:**

The paper tackles an important and relevant problem. The considered assumptions are a bit restrictive. The parametric assumptions were well justified and motivated in the text and the theoretical contributions were illustrated through examples and seem solid. However, most of the proofs and intuitions were moved to the appendix, making the main part of the paper hard to read.

A critical issue of the paper is that the assumption of causal sufficiency is not even mentioned in the paper.  When describing LinNGAM in line 49, the authors say that

“the generating process for $\mathbf{\tilde{X}}$ is linear, non-Gaussian, acyclic model (LiNGAM)”.

Then, in line 249, the authors say:

“ Assumption 1 is the only one we make besides LiNGAM throughout  the paper, where violation of Assumption 1 is of Lebesgue measure 0, and LiNGAM is testable.”

The causal sufficiency assumption is not mentioned neither for LiNGAN nor for the proposed method. This is the strongest assumption and is NOT testable. Although it may be okay to make this assumption in this context, an appropriate justification must be provided. Further, it would be really appreciated if some simple examples illustrating what would happen if the causal assumption is violated.

Disclaimer:  Apart from the discussion on causal sufficiency, the text is well-written and the problem is well-motivated. Examples also helped to understand the contributions. However, I didn't check the proofs in the appendix and I am not familiar with the previous methods (IN and GIN), so I cannot judge the novelty and soundness of the methods.

---

> ### Author Response · Authors · 2022-08-02
> **Response to Reviewer ZM5r**
>
> We are grateful for the reviewer's insightful comments and constructive suggestions. Please see below for our response.
>
> ---
>
> **(Q1)** The reviewer has questions regarding the causal sufficiency assumption. Specifically,
>
> > "The assumption of causal sufficiency is not even mentioned in the paper."
>
> **R:** Many thanks for your concern. In our presentation, we considered causal sufficiency as implied by the formal definition of LiNGAM (please refer to the original paper [1], p.2005, paragraph 3). In light of your concern, for clarity, we have explicitly made the causal sufficiency assumption (in addition to LiNGAM) in the revision (line 48: "assume causal sufficiency relative to $\tilde{\mathbf{X}}$").
>
> > Further, "what would happen if the causal assumption is violated?"
>
> **R:** This is indeed an important practical issue. In short, if directly using the _one-over-others-TIN_ method in this paper, the output causal ordering may be incorrect. Interestingly, however, we found that this may depend on the specific structural patterns - we provide two illustrating examples: one where the group ordering is still (partially) identifiable, and one in the contrast. Please see Appendix F.5 in the revision for our detailed discussion.
>
> Overall, we really appreciate the reviewer for this insightful question. Though as mentioned by the reviewer, "it is okay to make this assumption in this context" (which, to the best of our knowledge, is indeed a common assumption in the current literature on handling measurement error), the assumption itself, after all, is strong and not testable. It would be useful (and fun!) to investigate the case where causal sufficiency is violated (in a sense of "latent of latent") systematically. With the TIN condition, if we can characterize the "specific patterns" mentioned above, we may further construct correction rules or algorithm relaxations so that the identifiability is still (partially) preserved. We leave the systematic investigation as a line of our future research.
>
> > "The authors say that Assumption 1 is the only one made besides LiNGAM."
>
> **R:** Because of your comment and concern, we have updated the paper to further explicitly include the causal sufficiency assumption (as discussed above) and the random measurement error model (which is a standard model to deal with measurement error in the current literature).  Please see lines 49 and 249 of the updated manuscript.
>
> It might be helpful to mention the difference between our assumptions and those required by other methods: existing methods for causal discovery with latent variables usually either make
>  - additional structural assumptions (e.g., "require at least two measurements for each latent variable"[2, 3]; "require the DAG to be a polytree or some other specific families"[4]), or
> - additional parametric assumptions (e.g., "require that the conditional probabilities $P(X_i | \tilde{X}_i)$ can be estimated from data, by assuming e.g., mutually irreducible distributions"[5]).
>
> Our problem setting and the assumptions are identical to that in [6], which is quite general compared to others. While comparing to [6], we achieve the same identifiability results by escaping from O-ICA but only conducting independence tests.
>
> ---
>
> **(Q2)** "The proofs and intuitions are moved to the appendix."
>
> **R:** Although we had to move the proofs to the appendix due to the page limit, we have tried to give interpretations of Theorems 1 and 2's proofs in lines 220 and 260 of the main paper. In addition, we included illustrative examples 1, 2, 3, and 4 also for that purpose; we tried to connect them to our basic idea: "_create independence_, by leveraging the parametric assumption and benefit from non-Gaussianity" (line 95), and "asymmetry actually exists beyond GIN, by $\omega$ characterized from higher-order statistics" (line 173).
>
> Thanks again for your interest in our proofs and intuitions!
>
> ---
>
> [1] Shimizu, Shohei, et al. "A linear non-Gaussian acyclic model for causal discovery." _Journal of Machine Learning Research_ 7.10 (2006).
>
> [2] Kummerfeld, Erich, and Joseph Ramsey. "Causal clustering for 1-factor measurement models." _Proceedings of the 22nd ACM SIGKDD international conference on knowledge discovery and data mining_. 2016.
>
> [3] Salehkaleybar, Saber, et al. "Learning Linear Non-Gaussian Causal Models in the Presence of Latent Variables." _Journal of Machine Learning Research_ 21 (2020): 39-1.
>
> [4] Anandkumar, Animashree, et al. "Learning linear bayesian networks with latent variables." _International Conference on Machine Learning_. PMLR, 2013.
>
> [5] Halpern, Yoni, Steven Horng, and David Sontag. "Anchored discrete factor analysis." _arXiv preprint arXiv:1511.03299_(2015).
>
> [6] Zhang, Kun, et al. "Causal Discovery with Linear Non-Gaussian Models under Measurement Error: Structural Identifiability Results." _UAI_. 2018.

---

> > ### Author Response · Authors · 2022-08-07
> > **Could you please let us know whether our responses and updated submission properly addressed your concern?**
> >
> > Dear Reviewer ZM5r,
> >
> > Thank you very much for your time spent on our submission. We have tried to address your concerns in the response and updated submission--any feedback from you would be appreciated. If you have further comments, please kindly let us know--we hope for the opportunity to respond to them.
> >
> > Best wishes,
> > Authors of paper 202

---

> > ### Author Response · Authors · 2022-08-09
> > **Could you please provide your feedback on our response?**
> >
> > Dear Reviewer ZM5r,
> >
> > Thanks for providing the Author Rebuttal Acknowledgement. Given that the discussion involving authors will end in 5.5 hours, could you please let us know whether your main concerns were addressed by our response and updated submission?  If there are any other concerns, please let us know, and we will immediately respond to them.
> >
> > If your main concerns are addressed, could you please update your recommendation to reflect it?  We understand you are very busy and appreciate your time.  Your feedback is valuable to us--we will be waiting for it.  Thank you.
> >
> > Best wishes,
> > Authors of submission 202

---

### Author Response · Authors · 2022-08-09
**Online demo of the TIN condition for you at http://tincondition.xyz**

Dear program committee members,

Many thanks for your insightful feedback! We have tried to address your questions in the response and updated manuscript.

In addition, we have also made available an online demo for the Transformed Independent Noise (TIN) condition, a key contribution of this paper. By playing with TIN on specific structure examples, users may gain a better intuition about TIN. You are welcome to try it out at [http://tincondition.xyz](http://tincondition.xyz) .

Please kindly let us know if you have any further comments. We are looking forward to the opportunity to respond to them. Thanks again for your time spent on our submission.

Best wishes,
Authors of Paper 202

---

### Meta-Review · Area_Chair_TpjK · 2022-08-26

**Recommendation:** Accept
**Confidence:** Certain

**Metareview:**

The paper considers structure recovery when there is a causal DAG on variables X where causal mechanisms are linear but exogenous noise variables are non-Gaussian (similar setting to the one in the standard prior work LiNGAM). However, each variable is not directly observed but through measurement independent noise. Authors show that by using independence tests between transformations of variables, one can recover the order group decomposition of the graph.

I think the identifiability result is novel. Reviewers are overall positive.
Main concerns were:
a) Main concern of the reviewer with the lowest score is simply that causal sufficiency has not been spelled out
b) Another concerns is missing references to assumptions when deriving theoretical results clearly.

These two are not very major and I suggest the authors to pay attention to these and comprehensively list all assumptions clearly upfront in their camera ready.



**Award:**

No

---

### Decision · Program_Chairs · 2022-09-14

Accept